

# Plectronoceratids (Cephalopoda) from the latest Cambrian at Black Mountain, Queensland, reveal complex three-dimensional siphuncle morphology, with major taxonomic implications

Alexander Pohle[1,2], Peter Jell[3] and Christian Klug[1]

[1] Palaeontological Institute and Museum, University of Zurich, Zürich, Switzerland
[2] Institute of Geology, Mineralogy and Geophysics, Ruhr University Bochum, Bochum, Germany
[3] School of Earth Sciences, University of Queensland, St. Lucia, Queensland, Australia

Corresponding author
Alexander Pohle,
alexander.pohle@rub.de

## ABSTRACT

The Plectronoceratida includes the earliest known cephalopod fossils and is thus fundamental to a better understanding of the origin and early evolution of this group of molluscs. The bulk of described material comes from the late Cambrian Fengshan Formation in North China with isolated occurrences in South China, Laurentia, Kazakhstan and Siberia. Knowledge of their morphology and taxonomy is limited in that most specimens were only studied as longitudinal sections, which are prone to misinterpretations due to variations in the plane of section. We describe more than 200 new specimens, which exceeds the entire hitherto published record of plectronoceratids. The material was collected by Mary Wade and colleagues during the 1970s and 1980s, from the lower Ninmaroo Formation at Black Mountain (Mount Unbunmaroo), Queensland, Australia. Despite the collecting effort, diverse notes and early incomplete drafts, Mary Wade never published this material before her death in 2005. The specimens provide novel insights into the three-dimensional morphology of the siphuncle based on abundant material, prompting a general revision of the order Plectronoceratida. We describe *Sinoeremoceras marywadeae* sp. nov. from numerous, well-preserved specimens, allowing investigation of ontogenetic trajectories and intraspecific variability, which in turn enables improved interpretations of the three-dimensional siphuncle morphology. The siphuncle of *S. marywadeae* sp. nov. and other plectronoceratids is characterised by highly oblique segments, an elongated middorsal portion of the septal neck (= septal flap) and laterally expanded segments that extend dorsally relative to the septal flap (= siphuncular bulbs). We show that this complex siphuncular structure has caused problems of interpretation because it was studied mainly from longitudinal sections, leading to the impression that there were large differences between specimens and supposed species. We revise the order Protactinoceratida and the families Protactinoceratidae and Balkoceratidae as junior synonyms of the Plectronoceratida and Plectronoceratidae, respectively. We reduce the number of valid genera from eighteen (including one genus formerly classified as an ellesmeroceratid) to three: *Palaeoceras* Flower, 1954, *Plectronoceras* Kobayashi, 1935 and *Sinoeremoceras* Kobayashi, 1933. We accept 10 valid species to which the 68 previously established

species may be assigned. *Sinoeremoceras* contains 8 of the 10 plus the new species. Two species, previously referred to ellesmeroceratid genera, are transferred to *Sinoeremoceras*. This revised scheme groups plectronoceratids into distinct geographically and stratigraphically separated species, which better reflects biological realities and removes bias caused by preparation techniques. North China remains important containing the highest known diversity and was likely a centre of cephalopod diversification.

## INTRODUCTION

Cephalopods date back to the late Cambrian (*Teichert, 1967*, *1988*; *Holland, 1987*; *Wade, 1988*; *Kröger, Vinther & Fuchs, 2011*; *Pohle et al., 2022*), but the earliest unequivocal cephalopods are comparatively poorly known, despite more than 160 species in about 40 genera having been described from this time. Most descriptions are from the late Cambrian Fengshan Formation of North China (*Kobayashi, 1933*, *1935*; *Chen et al., 1979a*[1], *1979b*; *Chen et al., 1980*; *Chen & Qi, 1982*; *Chen & Teichert, 1983*; *Lu, Zhou & Zhou, 1984*), but a few occur in South China (*Chen & Qi, 1981*; *Li, 1984*), Laurentia (*Flower, 1954*, *1964*; *Landing & Kröger, 2009*; *Landing et al., 2023*), Kazakhstan (*Malinovskaya, 1964*) and Siberia (*Korde, 1949*; *Balashov, 1959*; *Dzik, 2020*). The number of described specimens per species is typically very low and while subsequent studies have questioned the distinctness of at least some of these taxa (*Holland, 1987*; *Hewitt, 1989*; *Wade & Stait, 1998*; *Mutvei, Zhang & Dunca, 2007*; *King & Evans, 2019*; *Mutvei, 2020*; *Pohle et al., 2022*), no comprehensive revision of Cambrian cephalopods has been undertaken for several decades. Cambrian taxa are morphologically very similar in that they are relatively small and have a slight endogastric conch curvature with a marginally positioned siphuncle, a compressed cross-section, and closely spaced septa. Nevertheless, they can be divided into two distinct groups, one with an expanded siphuncle (Plectronoceratida and Protactinoceratida) and a second group with a tubular or slightly concave siphuncle (Ellesmeroceratida and Yanheceratida) (*Chen & Teichert, 1983*). This study addresses the first group, documenting new representatives and revising the taxonomy.

The first report of a Cambrian cephalopod remains the oldest known unequivocal cephalopod, originally described as *Cyrtoceras cambria Walcott, 1905*, a tiny cyrtoconic shell with a siphuncle on the concave side of the conch (subsequently referred to as ventral). Almost 30 years later, this species was designated the type of *Plectronoceras Ulrich & Foerste, 1933* and *Kobayashi (1935)* erected the Plectronoceratidae. *Kobayashi (1933*, *1935*) was the first to document the peculiar siphuncular structure in members of this family, which consisted, in his view, of frequent ontogenetic changes in the shape and length of the septal necks and the presence of (using Kobayashi's terminology) "siphuncular bulbs", which represent parts of the siphuncle that are strongly swollen.

[1] In the English abstract of *Chen et al. (1979a)*, the author 邹西平 is incorrectly transliterated as Tsou Si-Ping, while *Chen et al. (1979b)* and later publications list the same author as Zou Xi-Ping. We consistently use the latter spelling; note that this affects the spelling of the author attribution of some species.

Lastly, he reported structures crossing the siphuncle, "tabulae" and "pseudodiaphragms". He erected the Plectronoceratidae to include *Plectronoceras Ulrich & Foerste, 1933*, *Sinoeremoceras Kobayashi, 1933*, *Multicameroceras Kobayashi, 1933* and *Wanwanoceras Kobayashi, 1933* and assigned them to the Ellesmeroceratida. The latter three genera were distinguished by small differences in conch outline and siphuncle structure and at that time still considered to be Early Ordovician in age (*Kobayashi, 1931*, *1933*). The only known Siberian plectronoceratid, *Multicameroceras sibiriense Balashov, 1959*, has been largely overlooked by later authors. Kobayashi's classification remained and was adopted by the American *Treatise on Invertebrate Palaeontology* (*Furnish & Glenister, 1964*) and its Russian equivalent, the *Osnovy Paleontologii* (*Balashov, 1962a*). *Flower (1964)* introduced the suborder Plectronoceratina, to include the Plectronoceratidae with *Plectronoceras* and *Palaeoceras Flower, 1954*, and a new family, the Balkoceratidae with the new *Balkoceras Flower, 1964*, and tentatively *Shelbyoceras* Ulrich & Foerste *in Bridge (1931)*, although the latter was subsequently identified as a monoplacophoran (*Stinchcomb & Echols, 1966*; *Stinchcomb, 1980*). *Palaeoceras* and *Balkoceras* shared a siphuncular bulb with previously described taxa, but differed by their conch curvature, *Palaeoceras* being orthoconic, while *Balkoceras* was described as slightly exogastric.

The largest addition of new plectronoceratid taxa resulted from a series of articles documenting Cambrian cephalopods from China (*Chen et al., 1979a*, *1979b*, *1980*; *Chen & Qi, 1981*, *1982*; *Chen & Teichert, 1983*; *Li, 1984*; *Lu, Zhou & Zhou, 1984*). The Plectronoceratina became the order Plectronoceratida and the order Protactinoceratida was erected with only the Protactinoceratidae, which was distinguished from the Plectronoceratidae mainly by their larger, more strongly expanded siphuncles and well developed diaphragms with calcareous deposits between them (*Chen et al., 1979a*; *Chen & Teichert, 1983*). Several dozen species were introduced to the Plectronoceratida and assigned to *Eodiaphragmoceras* Chen & Qi *in Chen et al., 1979a*; *Jiagouceras*, Chen & Zou *in Chen et al., 1979a*; *Lunanoceras* Chen & Qi *in Chen et al., 1979a*; *Paraplectronoceras* Chen, Qi & Chen *in Chen et al., 1979a*; *Rectseptoceras* Zou & Chen *in Chen et al., 1979a*; *Theskeloceras Chen & Teichert, 1983* and *Parapalaeoceras Li, 1984*. Species of *Protactinoceras* Chen & Qi *in Chen et al., 1979a*; *Physalactinoceras* Chen & Qi *in Chen et al., 1979a*; *Benxioceras Chen & Teichert, 1983* and *Mastoceras Chen & Teichert, 1983* were attributed to the Protactinoceratida. The Cambrian age of *Kobayashi (1931*, *1933)*'s specimens was clarified (*Chen et al., 1979a*, *1979b*) and *Multicameroceras Kobayashi, 1933* was synonymised with *Wanwanoceras Kobayashi, 1933* by *Chen & Teichert (1983)*. Thus, the number of genera had increased from six to sixteen and species numbers had similarly increased.

A different opinion was presented by *Dzik (1984)*, who regarded *Plectronoceras* and *Multicameroceras* as the only valid members of the Plectronoceratidae; he accepted *Palaeoceras* but transferred it to the Ellesmeroceratidae. In his concept, both families belonged to the suborder Ellesmeroceratina of the order Endoceratida. Although the general concept of *Dzik (1984)* was criticized by *Turek & Marek (1986)* and *Wade (1988)*, there were further critical opinions on the high number of Chinese species, citing a "tendency to split rather than to lump" (*Holland, 1987*, p. 3) or that the authors "seem to
imply that an average of 1.6 specimens are sufficient to judge the intraspecific variation of 30 new nautiloid species and that one in three of these species has the status of a new genus" (*Hewitt, 1989*, p. 284). It is worth noting here that the vast majority of Chinese species were based on longitudinal (thin) sections only, and neither three-dimensional outline of the conch nor suture lines or shell ornamentation are known.

An intense debate on the origin of cephalopods that emerged during the 1970s and continued through the 1980s revolved almost exclusively around comparisons with *Plectronoceras cambria*, stating that it was poorly known from very few specimens, ignoring the rich material of slightly younger Cambrian cephalopods from China and elsewhere (*e.g.*, *Yochelson, Flower & Webers, 1973*; *Jell, 1976*; *Runnegar & Jell, 1976*; *Dzik, 1981*, *1984*; *Bandel, 1982*; *Chen & Teichert, 1983*; *Runnegar & Pojeta, 1985*; *Teichert, 1988*; *Wade, 1988*; *Webers & Yochelson, 1989*; *Webers, Yochelson & Kase, 1991*)—although to be fair the Cambrian age of the Chinese material was only clarified in the year 1979. The debate focussed mostly on the origin of the siphuncle and the order of character acquisition of (multiple?) septa, perforation of the septa and connecting ring, and is still unresolved today. We do not add much to this topic, but by showing that plectronoceratids are less diverse than previously thought, we suggest consideration of a broader spectrum of plectronoceratids instead of focussing on *Plectronoceras* alone when identifying cephalopod origins; as we show, this genus is generally very similar to later plectronoceratids.

*Mutvei, Zhang & Dunca (2007)* was the next study focussing on plectronoceratids, and it made detailed investigations of the siphuncular structure of *Protactinoceras* and *Theskeloceras*, concluding that the Protactinoceratida had to be synonymised with the Plectronoceratida, because they considered that differences in siphuncle morphology were caused by some sections being misaligned with the median plane. Nevertheless, they retained Plectronoceratidae and Protactinoceratidae as valid, without identifying distinguishing characteristics. *Mutvei (2020)* added another hypothesis on the origin of the cephalopod siphuncle and acknowledged a very large, strongly expanded siphuncle in protactinoceratids as diagnostic character but cited *Protactinoceras* as its only member, transferring all other genera to the Plectronoceratida. *Pohle et al. (2022)* synonymised the Plectronoceratida and Protactinoceratida based on siphuncle morphology but provided no detailed descriptions or illustrations, which is one purpose of this article.

This historical review of Cambrian Plectronoceratida demonstrates the poor level of understanding that remains and highlights the desperate need for better preserved specimens to address this issue. Such material has been available in museum collections for 40 years but remained unpublished. Abundant ellesmeroceratid cephalopods were reported in the Ninmaroo Formation at Black Mountain, western Queensland, Australia (margin of East Gondwana during the late Cambrian) and dated as Tremadocian, considered as part of the Lower Ozarkian by *Whitehouse (1936*; efforts to trace these specimens have failed). The question of where to draw the Cambrian–Ordovician boundary was uppermost in Whitehouse's mind, and although he placed the Ozarkian, with his *Ellesmereoceras* [*sic.*] Stage near its base, in the Cambrian, the Ninmaroo Formation came to be recognised as spanning the lower boundary of the Tremadocian so

that the bulk of that formation is Ordovician (*Öpik, 1960*). Although *Öpik (1960)* listed the fauna of the Ninmaroo Formation in depth, he did not mention ellesmeroceratids. However, *Teichert & Glenister (1952)* did list Whitehouse's ellesmeroceratids in their survey of Australian nautiloids. *Öpik (1967)* listed "nautiloids" at two localities in the Mindyallan Mungerebar Limestone, but this report was discarded by *Chen & Teichert (1983)*, as none of the original material could be tracked down. During the 1970s and 1980s, Mary Wade and Queensland Museum field parties collected many cephalopods from the upper Cambrian part of the Ninmaroo Formation at Black Mountain and from Early Ordovician faunas in the Toko Range. Unfortunately, apart from a few publications (*e.g., Wade, 1977a, 1977b*), much of her extensive collection of early Palaeozoic cephalopods remained undescribed at the time of her death in 2005 (see *Turner, 2007* for a summary of her scientific achievements). Remarkably, despite frequent exchanges in writing between Mary Wade and other contemporaneous experts on Palaeozoic cephalopods such as Rousseau H. Flower, Chen Jun-Yuan and Curt Teichert, some of them specifically mentioning the Cambrian material from Queensland, its existence had largely been forgotten in the newer literature. This is even more astounding considering that cephalopods in the Ninmaroo Formation had been mentioned on multiple occasions before (*Whitehouse, 1936*; *Teichert & Glenister, 1952*; *Druce, Shergold & Radke, 1982*; *Grégoire, 1988*; *Wade, 1988*; *Kobayashi, 1989*; *Nicoll & Shergold, 1991*; *Shergold et al., 1991*; *Wade & Stait, 1998*). We continue Mary Wade's work here and use it as a basis for a general revision of the Plectronoceratida. Note that the material also includes ellesmeroceratids, which will be covered in a separate study.

## MATERIALS AND METHODS

### Geological background

The material described here comes from Black Mountain (Mount Unbunmaroo), which is situated 55 km northeast of the nearest town, Boulia, in the Burke River Structural Belt within the southeastern part of the Georgina basin in western Queensland (Fig. 1). Black Mountain, Mount Ninmaroo, Mount Datson and Dribbling Bore make up a faulted and folded belt trending south southeast. The section at Black Mountain has been the subject of numerous faunal studies around the Cambrian–Ordovician boundary interval and has played an important role in regional and global biostratigraphic correlations (*Druce & Jones, 1971*; *Jones, Shergold & Druce, 1971*; *Shergold, 1975*; *Druce, Shergold & Radke, 1982*; *Shergold et al., 1982, 1991*; *Nicoll & Shergold, 1991*; *Ripperdan et al., 1992*; *Shergold & Nicoll, 1992*; *Zhen, Percival & Webby, 2017*).

The earliest occurrence of cephalopods at Black Mountain is within the Ninmaroo Formation, which overlies the Chatsworth Limestone (Fig. 2). The origin of the specimens is indicated by horizon numbers (BMT1, BMT 2, *etc.*,) given by Mary Wade and written on nearly all specimens as noted in the Queensland Museum collection database. Furthermore, she plotted the numbered horizons on the stratigraphic chart from *Druce, Shergold & Radke (1982)*. The plectronoceratids come exclusively from layer BMT 1, which is situated at around 500 m in *Druce, Shergold & Radke*'s *(1982)* section, within the middle part of the Unbunmaroo Member of the Ninmaroo Formation. In their chart, this horizon
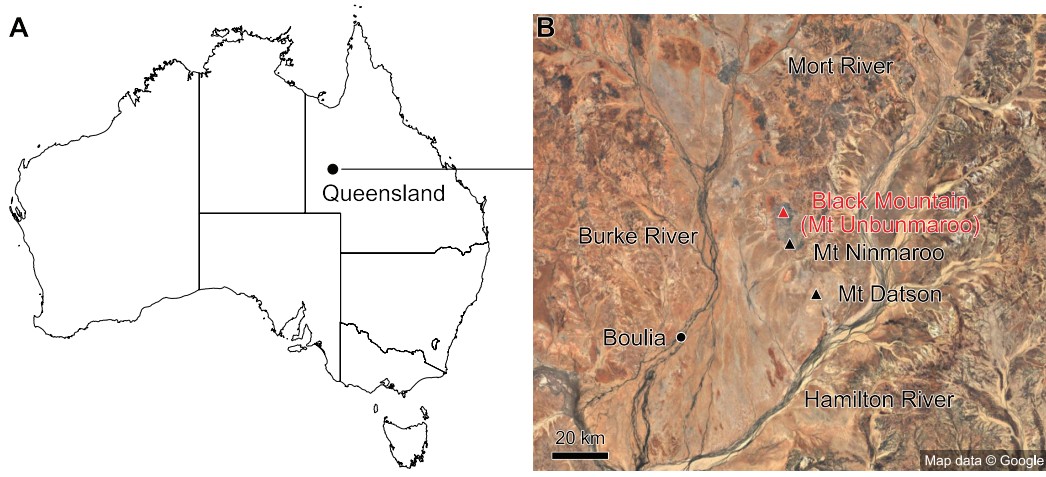

**Figure 1** **Map showing the location of Black Mountain.** (A) Location within Australia and Queensland. (B) Satellite image of the southern part of the Burke River Structural Belt (map © 2023 Google).

corresponds to the lower Datsonian regional stage and the *Mictosaukia perplexa* trilobite zone and *Cordylodus proavus* conodont zone. Following the revised conodont zonation of *Nicoll & Shergold (1991)*, BMT 1 lies within their Assemblage Zone 3 (*Hirsutodontus appressus*), as there is no evidence of *C. proavus* lower than the upper Unbunmaroo Member. This zone lies within the upper part of the *Eoconodontus* Zone and can thus be correlated with the *Cambrooistodus minutus* Subzone in Laurentia (*Miller, 2020*). The base of the Datsonian is defined by the lowest occurrence (= LO; see *Landing et al. (2013)* for problems of the FAD concept) of *C. proavus*, so horizon BMT 1 is late Payntonian rather than Datsonian in age. *Iapetognathus fluctivagus*, the primary marker for the base of the Ordovician is absent from the Black Mountain section, complicating the global correlation of the Cambrian–Ordovician boundary at this locality, although the currently available evidence places it within the *Cordylodus lindstromi* Zone that also marks the base of the Warendian (*Nicoll & Shergold, 1991*; *Shergold et al., 1991*; *Ripperdan et al., 1992*) or slightly higher (*Zhen, Percival & Webby, 2017*). In any case, the horizon BMT 1 is dated well into the Cambrian Stage 10, but potentially slightly younger than the cephalopod occurrences from North China, where the coeval *Mictosaukia* Zone has been found to be essentially devoid of cephlaoopds (*Chen & Teichert, 1983*; *Fang et al., 2019*). Further evidence for the late Cambrian age of BMT 1 is the fact that *Eoconodontus notchpeakensis*, its LO below the onset of the HERB carbon isotope excursion being a potential marker species for the GSSP of the Cambrian Stage 10 (*Landing, Westrop & Miller, 2010*; *Landing, Westrop & Adrain, 2011*; *Miller et al., 2015*), occurs between the assemblage zones 2–4 of *Nicoll & Shergold (1991)*. In comparison, the cephalopod-rich Wanwankou Member of the Fengshan Formation in North China correlates with the upper *Proconodontus muelleri* Zone and lower *Eoconodontus notchpeakensis* Subzone (*Chen & Teichert, 1983*). In their global correlation of the Cambrian–Ordovician boundary, *Geyer (2019)* and *Miller (2020)* placed the boundary within or at the base of the *C. lindstromi* Zone, respectively. Plectronoceratids appear to be slightly more common than ellesmeroceratids in BMT1, although we do not

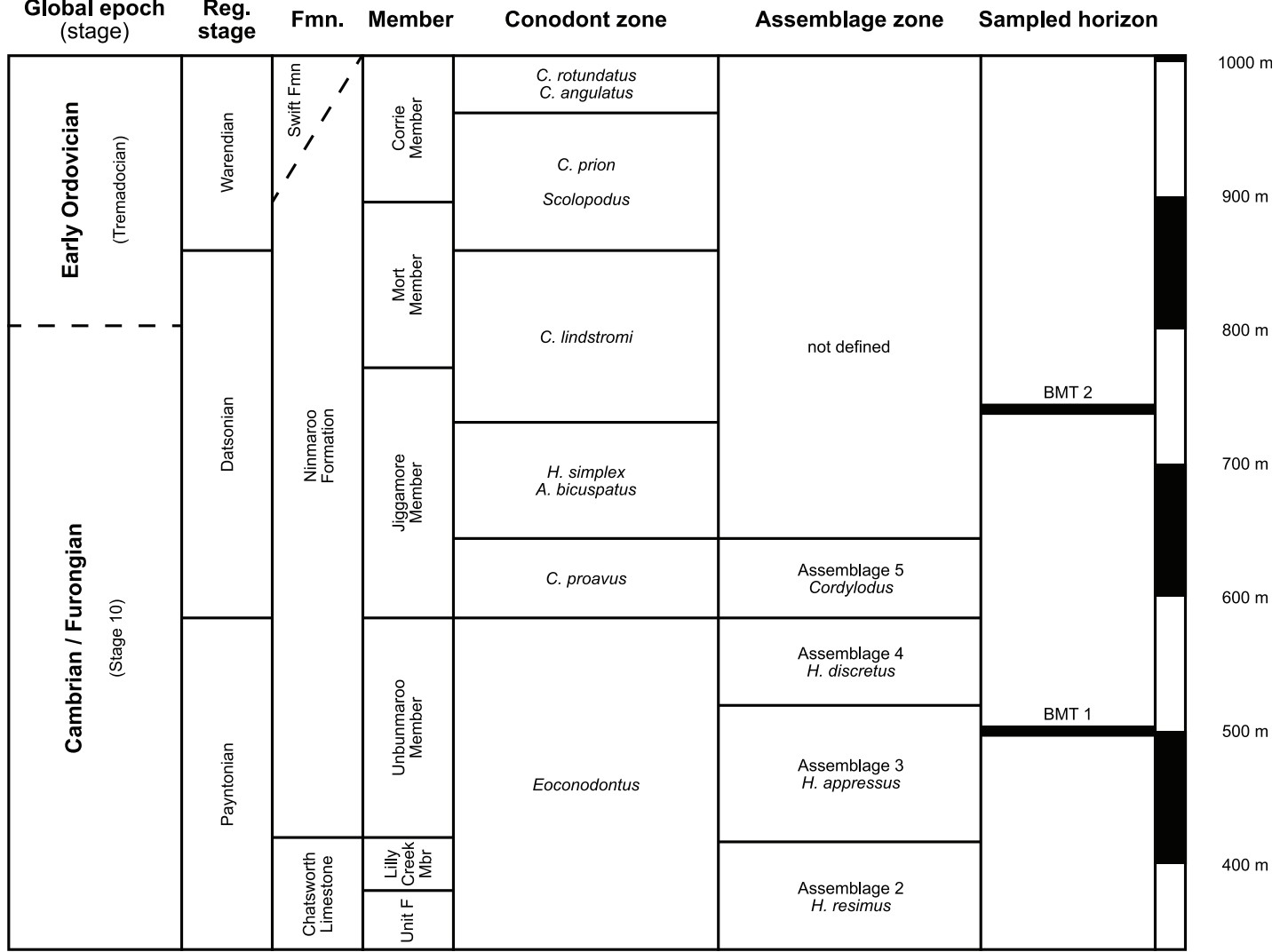

**Figure 2 Stratigraphic overview and correlation of the Ninmaroo Formation.** Data from *Druce, Shergold & Radke (1982)* and *Nicoll & Shergold (1991)*. BMT horizons represent Mary Wade's "nautiloid bands". All material studied here comes from horizon BMT 1. Only the oldest two cephalopod horizons are shown, both are Cambrian in age. Further abundant cephalopods occur in the early Tremadocian.

know whether this faunal composition is genuine or if it represents a difference in sampling effort. BMT 2 is situated in the lowermost *C. lindstromi* Zone, *i.e.*, probably very close to the Cambrian–Ordovician boundary but contains no plectronoceratids and is thus not investigated here.

## Morphology

Terminology and abbreviations of conch parameters largely follow *Pohle et al. (2022*, supplementary information) and are listed in Table 1. Measurements of the Black Mountain material were taken with digital callipers. To compare species from China, North America, and Russia, we took the published measurements and ratios from the original species descriptions where available, thus reflecting intraspecific variation as

**Table 1 Measurements and conch parameters used in this study, including corresponding abbreviations.**

| Abbreviation | Parameter | Calculation |
|---|---|---|
| ch | Conch height | Measured |
| cw | Conch width | Measured |
| cl | Cameral length | Measured |
| sd | Siphuncular diameter | Measured |
| l | Length of fragment | Measured |
| CWI | Conch width index | cw/ch |
| RCL | Relative cameral length | cl/ch |
| RSD | Relative siphuncular diameter | sd/ch |
| $ER_h$ | Height expansion rate | $2*\tan^{-1}(0.5*(ch_n-ch_{n-1})/l)$ |
| $ER_w$ | Width expansion rate | $2*\tan^{-1}(0.5*(cw_n-cw_{n-1})/l)$ |

conceived by the authors. We measured additional data points from our own photos or directly from the original illustrations using ImageJ (*Rueden et al., 2017*). Note that in the Chinese literature, expansion rate is usually given as a ratio, *e.g.*, an expansion rate of 1:5 corresponds to a conch that expands by 1 mm within a length of 5 mm. We calculated angular expansion rates from these values, using the following formula for an expansion ratio of a:b (after *Pohle & Klug, 2018*):

$$ER = 2 \times \tan^{-1}\left(\frac{0.5 \times a}{b}\right).$$

As expansion differs in lateral and dorsoventral directions in conchs with constant non-circular cross-sections (which is always the case in the material studied here), we differentiate between height expansion rate $ER_h$ and width expansion rate $ER_w$. For most previously described species, which are only available as longitudinal thin sections assumed to represent the median plane, we use only $ER_h$.

Despite the previous description of more than 60 species, ontogenetic trajectories of important conch parameters have never been plotted for Cambrian cephalopods; we provide them in this article for the first time, using conch height (ch) as proxy for the ontogenetic stage. Besides expansion rate ($ER_h$ and $ER_w$), we compared conch width index (CWI), relative cameral length (RCL) and relative siphuncular diameter (RSD). We did not distinguish between the diameter at the septal foramen and the siphuncular segment, because the difference is usually less than 1 mm, making it difficult to obtain accurate measurements, particularly when sections are potentially misaligned, or when the siphuncle has been subject to significant weathering, as is often the case in the Australian material. Ontogenetic trends were tested by using simple linear regression models, taking conch height as the independent variable. Where significant trends could be observed, the linear regression models were incorporated into species diagnoses, providing expected values of conch parameters at certain conch heights. Thus, in the case of significant *p*-values, intercept and slope describe the direction and position of the linear trendline,

while $R^2$ represents the proportion of variation that can be explained by ontogenetic variation and σ is the standard error of the regression, which can be thought of as the average spread of values from the trendline. From published material, we usually recorded only a single ontogenetic stage, to weight each specimen equally. The only exception to this approach was made where only very few specimens of a species were available; in those cases, we incorporated multiple measurements per specimen into the regression models. To test if the slopes and intercepts of the ontogenetic trajectories are significantly different from each other, we included the species as interaction terms into the regression models and used analyses of variance (ANOVA) to determine *p*-values. We exclusively used this approach where conch parameters showed trends throughout ontogeny. Intercepts were only compared where no significant difference in slope was detected, as the intercept (value of conch parameter at conch height of 0 mm) is only meaningful when the slopes are parallel, *i.e.*, the difference is constant throughout ontogeny. Because significant and non-significant *p*-values of the regression coefficients and intercepts may also result from preparation and preservation biases, sample size or scatter of the data, this approach requires careful interpretation of the results and should not be used as single discriminator between species.

We compare the distribution of maximum phragmocone size and body chamber size (where known) to investigate differences in conch size between populations. The specimens were grouped according to their geographic and stratigraphic occurrences, to show temporal and spatial variation between populations or species.

## Species delimitation

As the great majority of Cambrian cephalopod species have been described from China, comparison with these is essential. Unfortunately, most Chinese specimens are only available as longitudinal thin-sections (the original blocks are probably lost) and in most cases, it is difficult to reconstruct the exact orientation of the plane of section relative to the median plane. Examples of the result of misaligned sections have already been shown by *Wade & Stait (1998)*, *Mutvei, Zhang & Dunca (2007)* and *Mutvei (2020)*. Consequently, the diagnostic value of many characters is questionable, since other species would have to be sectioned exactly in the same plane (which is usually impossible to do) to be comparable. Unfortunately, this renders many Chinese taxa unrecognisable, until more material is studied to clarify the extent of (three-dimensional) variation. Nevertheless, declaring nearly all Chinese species as *nomina dubia* is undesirable because it would render most Chinese Cambrian cephalopods unusable for taxonomic purposes, thus ignoring a large part of the hitherto known morphological variation. Instead, we adopt a pragmatic approach, identifying which features may vary due to misaligned sections, based on comparisons with the three-dimensionally preserved siphuncles of the Australian plectronoceratids. If it appears likely that diagnostic characters of genera and species merely represent differences in the plane of section, we identify synonymy with other taxa. Additionally, we compare the ontogenetic trajectories to assess inter- and intraspecific variation. This approach shows the variability between species as conceived by their original authors, revealing whether they form distinct groups that can be separated based

on conch parameters. While some errors in conch parameters due to non-aligned plane of section are to be expected, we assume that these affect all specimens with a similar probability. Thus, the expected errors are more likely to be random than systematic, although the exact impacts on conch parameters are difficult to predict. As plectronoceratids have been reported from localities widely dispersed across China and other parts of the world, we compare material from different geographical regions to investigate whether there are differences between populations. We make similar comparisons of material from different stratigraphic levels. Additionally, we compare body size distributions from different localities and stratigraphic levels, with particular reference to preserved body chambers and their diameters.

When considering variation due to the plane of section, we refer to a Cartesian coordinate system (Fig. 3). Palaeozoic cephalopods are most commonly investigated using median sections, which corresponds to the yz-plane in our coordinate system. However, this ideal case is, in practice, difficult to achieve, especially for small specimens that are still embedded in matrix. The true plane of section can thus differ from the ideal median plane in a combination of three principal ways:

- x-translation: parallel shift of yz-plane along x-axis to off-centre position (may result in apparent differences in siphuncle size and position).
- y-rotation: rotation of yz-plane around y-axis (may result in apparent ontogenetic changes).
- z-rotation: rotation of yz-plane around z-axis (may result in apparent differences in siphuncle size, septal concavity, or expansion rate).

Cambrian cephalopods have also been investigated based on cross-sections (*e.g.*, *Xiaoshanoceras subcirculare* Chen & Teichert, 1983, pl. 17, fig. 5) or coronal sections (*e.g.*, *Balkoceras gracile* Flower, 1964, pl. 3, fig. 15). These two planes correspond to the xy-plane (possible modifications: z-translation, x-rotation and y-rotation; may result in apparent differences in cross-section shape) and xz-plane (possible modifications: y-translation, x-rotation and z-rotation; may result in apparent differences in siphuncle size, shape or ontogenetic changes), respectively.

Based on these considerations, we evaluated which of the characters commonly used for generic and specific diagnoses are likely to be influenced by misaligning the plane of section. Only characters that are unlikely to be affected by sectioning can be used with certainty for species identifications. Those characters that were identified by us to be susceptible to variations in alignment of the plane of section, are only cautiously used for taxonomic purposes. Accordingly, established diagnoses were subjected to the following questions, which guided us in our revised taxonomic classification of plectronoceratids:

1) Are there any discrete characters that allow for the distinction between species other than those features that can be explained by a variation in alignment of the plane of section?

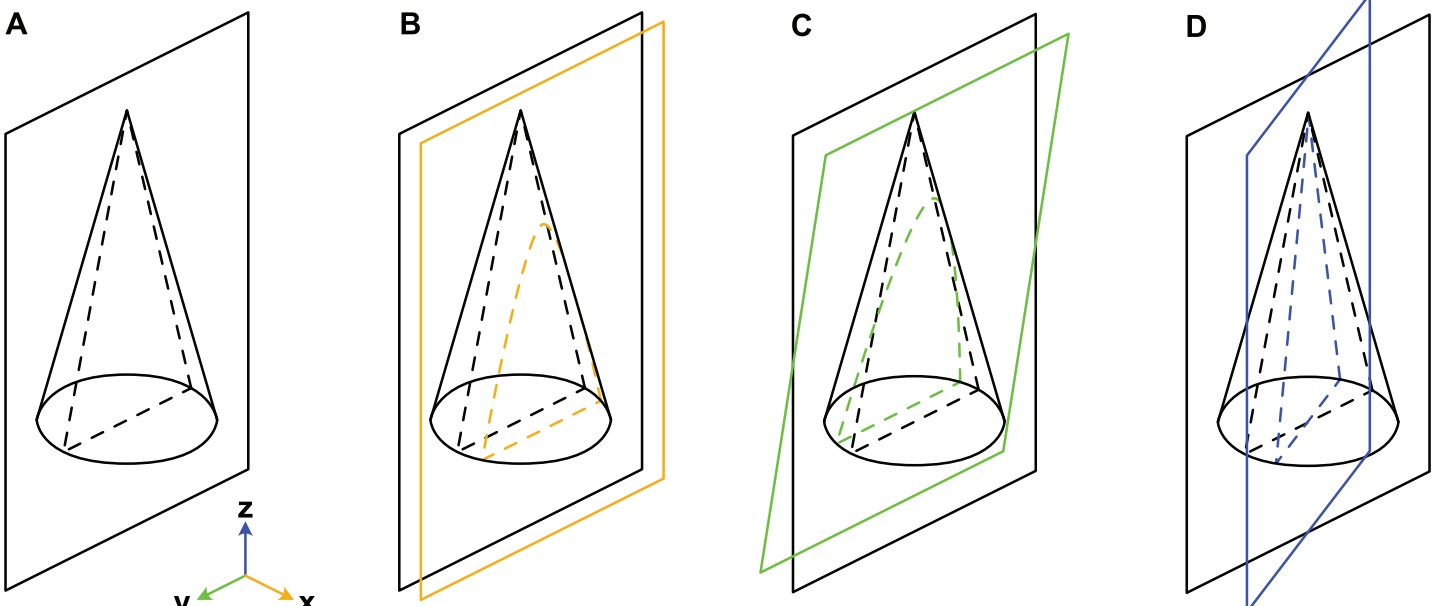

**Figure 3 Misalignment of the plane of section with the median plane.** The conch is here represented by a simple cone. (A) Plane of section identical to median plane. (B) Section displaced by x-translation. (C) Section displaced by y-rotation. (D) Section displaced by z-rotation. Note that although the cone has a radial symmetry in this example, the conchs of Cambrian cephalopods are bilaterally symmetrical because of their compressed cross-section, ventral siphuncle and in most cases slight endogastric curvature.

2) Are there distinct groupings of conch parameter distributions or ontogenetic distributions that could be used to distinguish between species?

3) Are the distributions of conch parameters and ontogenetic trajectories different between regions and/or stratigraphic level?

Synonymies proposed herein are intended as a useful reconsideration of previously available data with a view to better understanding of biological speciation. An improved taxonomy of plectronoceratids must include 3D-reconstructions through either μCT scans or serial grinding tomography, ideally of many specimens from the original localities to accurately assess variation of the siphuncle in three-dimensional space.

The electronic version of this article in Portable Document Format (PDF) will represent a published work according to the International Commission on Zoological Nomenclature (ICZN), and hence the new names contained in the electronic version are effectively published under that Code from the electronic edition alone. This published work and the nomenclatural acts it contains have been registered in ZooBank, the online registration system for the ICZN. The ZooBank LSIDs (Life Science Identifiers) can be resolved and the associated information viewed through any standard web browser by appending the LSID to the prefix http://zoobank.org/. The LSID for this publication is: urn:lsid:zoobank.org: pub:F1C67134-9A19-4D18-AB74-B984B5555D40. The online version of this work is archived and available from the following digital repositories: PeerJ, PubMed Central SCIE and CLOCKSS.

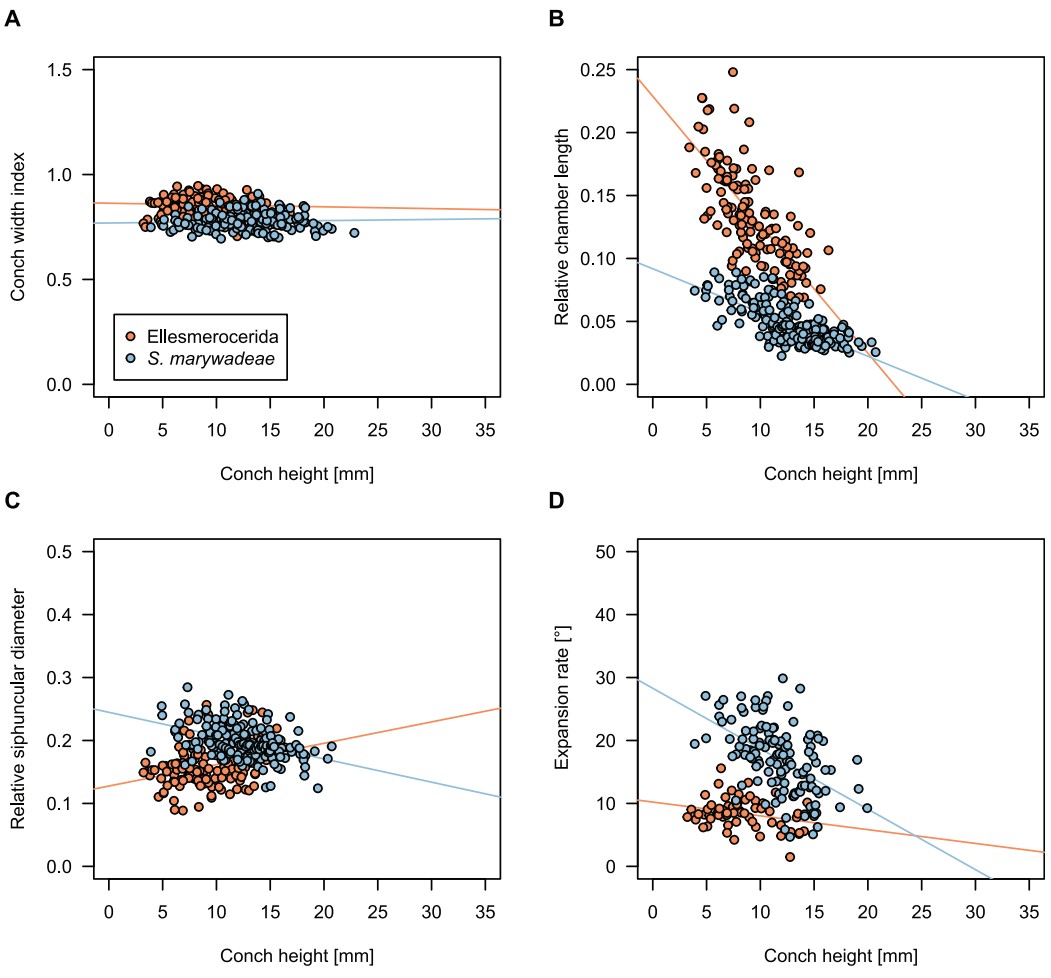

**Figure 4 Conch parameters through ontogeny of cephalopods in the Unbunmaroo Member (BMT 1) of the lower Ninmaroo Formation at Black Mountain, Queensland, Australia.** The cephalopods are represented by *Sinoeremoceras marywadeae* sp. nov. and undescribed ellesmeroceratids. Ontogeny is represented by conch height. (A) Conch width index (CWI). (B) Relative cameral length (RCL). (C) Relative siphuncular diameter. (D) Height expansion rate ($ER_h$).

# RESULTS

While plectronoceratids and ellesmeroceratids from the lower Ninmaroo Formation can be distinguished relatively easily by the more distant cameral spacing (Fig. 4B) and tubular siphuncle in ellesmeroceratids, our quantitative ontogenetic trajectories of the conch parameters did not reveal any distinct distributions within plectronoceratids, although there is considerable (but continuous) variability in conch width index, relative cameral length, relative siphuncle diameter and expansion rate (Fig. 4). Moreover, there are no purely qualitative characters that would allow for the distinction between multiple species. We thus assign all plectronoceratids from the Unbunmaroo Member at Black Mountain to a single species, *Sinoeremoceras marywadeae* sp. nov. (Figs. 5–7). In contrast to previously described plectronoceratids, the structure of the siphuncle is visible in three-dimensions, in numerous specimens (Fig. 7). This structure was described briefly by *Wade (1988)* and

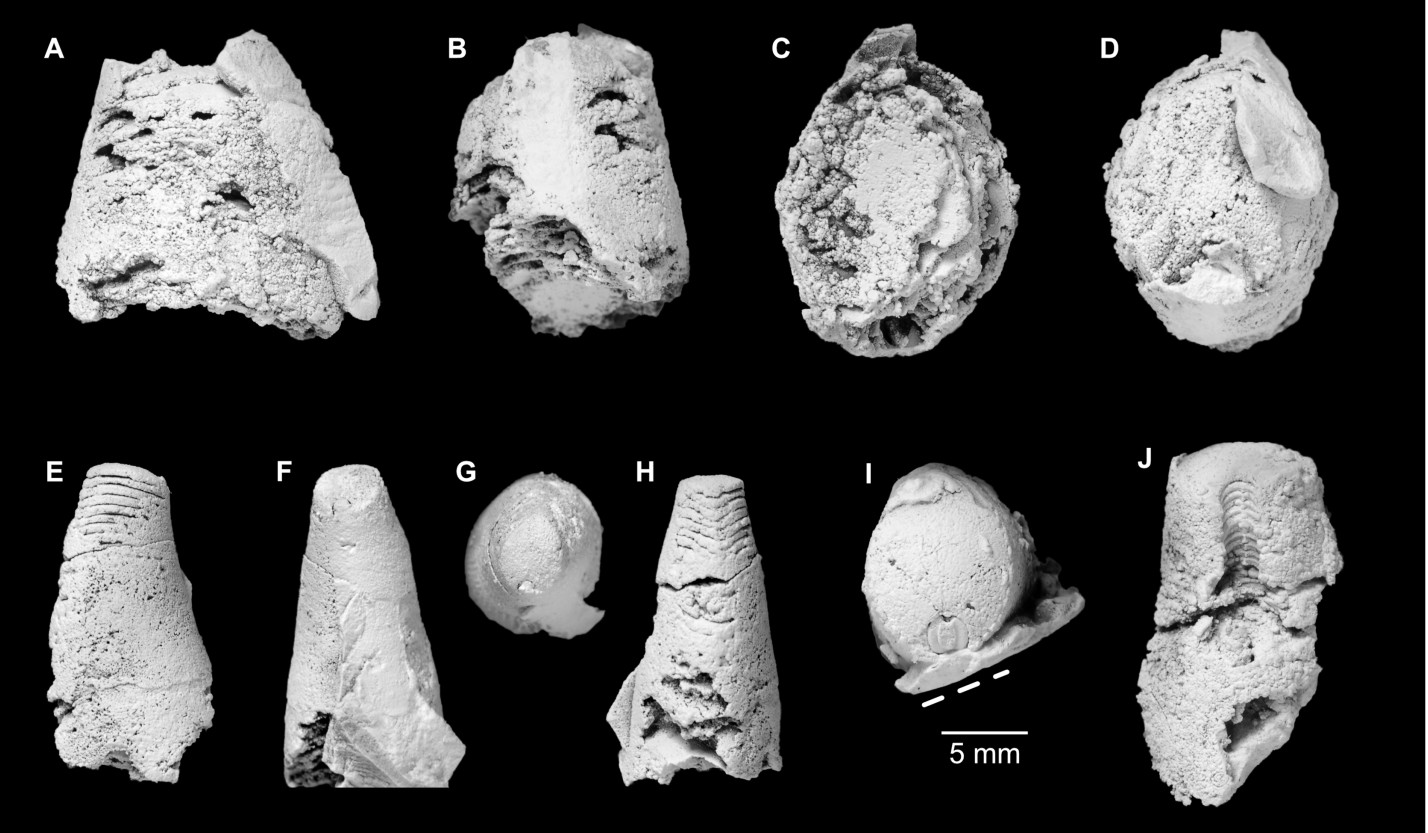

**Figure 5 *Sinoeremoceras marywadeae* sp. nov. from the lower Ninmaroo Formation, Unbunmaroo Member (BMT 1), Black Mountain, near Boulia, Queensland, Australia.** All specimens whitened with $NH_4Cl$. (A–D) QMF 39529, holotype. (A) Lateral view, siphuncle on the left side. (B) Ventral view. (C) Apertural view. (D) Apical view. (E–H) QMF 39533, paratype. (E) Lateral view, siphuncle on right side. (F) Ventral view. (G) Apical view. (H) Dorsal view. (I) QMF 13332, paratype, apical view. Dashed line indicates position of polished surface (Fig. 7L). (J) QMF 39542, paratype, ventral view.

figured in *Wade & Stait (1998)*. Without having seen the Australian material and based only on the interpretation of sectioned specimens from China, *Mutvei, Zhang & Dunca (2007)* and *Mutvei (2020)* more or less accurately reconstructed the three-dimensional structure of the siphuncle and suggested that there is no difference between plectronoceratids and protactinoceratids. The Australian specimens confirm the hypotheses on the synonymy of two orders and allow a more detailed description and reconstruction (Fig. 8).

All measurements and data taken from the literature are reported in Data S1–S3. Tables S1–S4 contains *p*-values for pairwise comparisons of the linear regression models for all species accepted here.

## Siphuncle morphology

The siphuncular segments are strongly oblique (Fig. 7D). In combination with the very short chamber length, this means that individual segments are strongly elongated dorsoventrally despite the cross-section of the siphuncle being roughly circular. In transverse section, three or four segments may be visible at the same time (Figs. 7G, 7J).

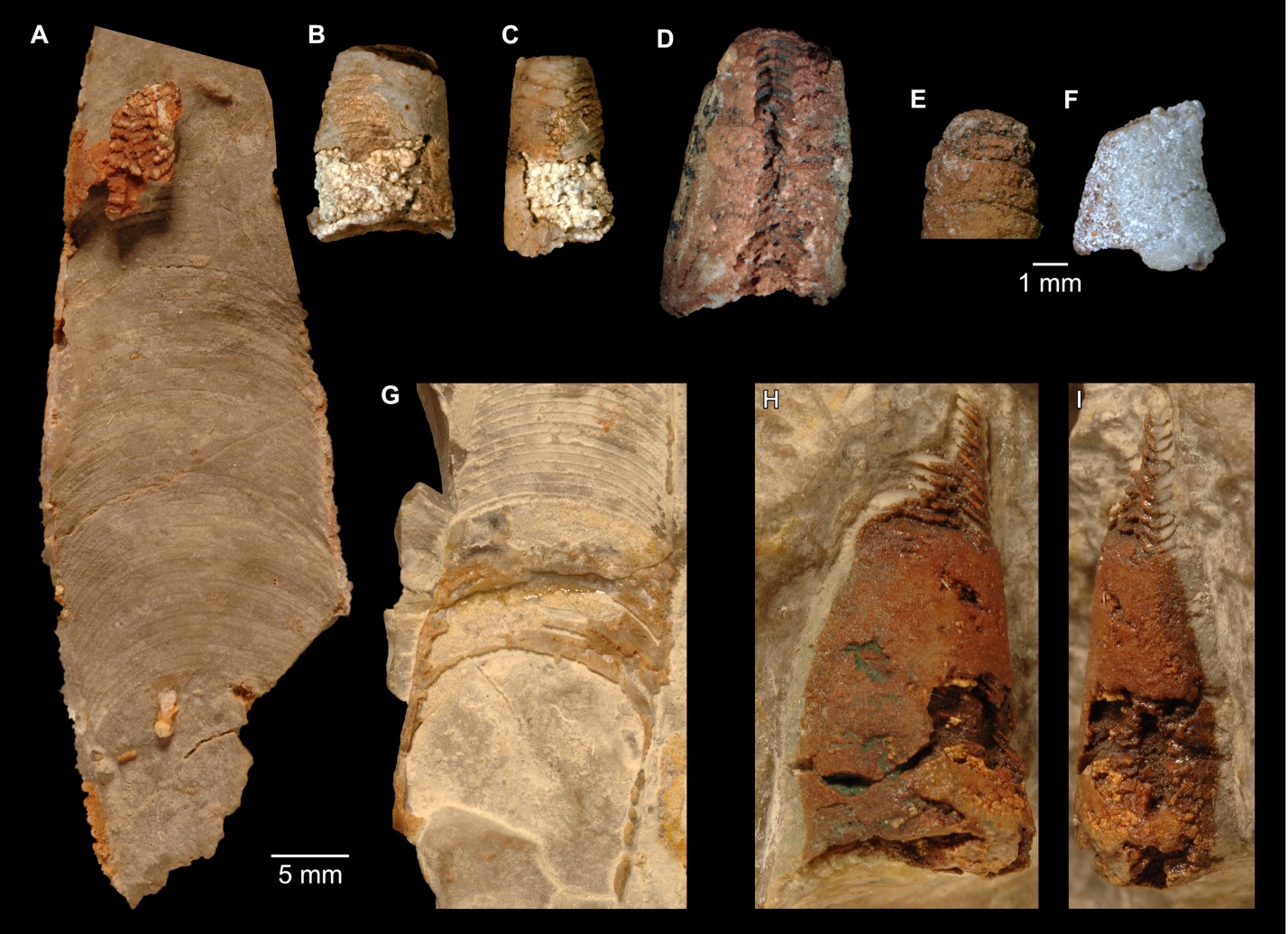

**Figure 6 *Sinoeremoceras marywadeae* sp. nov. from the lower Ninmaroo Formation, Unbunmaroo Member (BMT 1), Black Mountain, near Boulia, Queensland, Australia.** (A) QMF 39524, ventrolateral view of natural section. Mould of siphuncle at apical end of specimen, apertural end preserves part of body chamber. (B and C) QMF 61634, paratype. (B) Lateral view, siphuncle on right side. (C) Dorsal view. (D) QMF 61277, paratype, ventral view. Note the preservation of the siphuncle at apical and apertural end. (E) QMF 61372, small apical fragment, lateral view, siphuncle on right side. (F) QMF 40857, small apical fragment, lateral view, siphuncle on right side. (G) QMF 40846, natural section of specimen preserving at least part of body chamber. (H and I) QMF 40856, paratype. (H) Lateral view, siphuncle on right side. (I) Ventral view.

The segments are expanded, resulting in cyrtochoanitic septal necks (Figs. 7K–7M). It is not quite clear how the septal necks join the shell wall ventrally, but from longitudinal and cross-sections they appear to become straight (Figs. 7G, 7J, 7K). It is likely that the septal necks adnate to the shell wall midventrally, so that the foramen is open towards the shell for a short distance. This would explain why most specimens are preserved with the siphuncle exposed along the venter as an external mould, though the question remains whether the connecting ring would be closed ventrally (thus directly overgrowing the shell wall). In internal moulds of the siphuncle, the segments end in a relatively acute angle

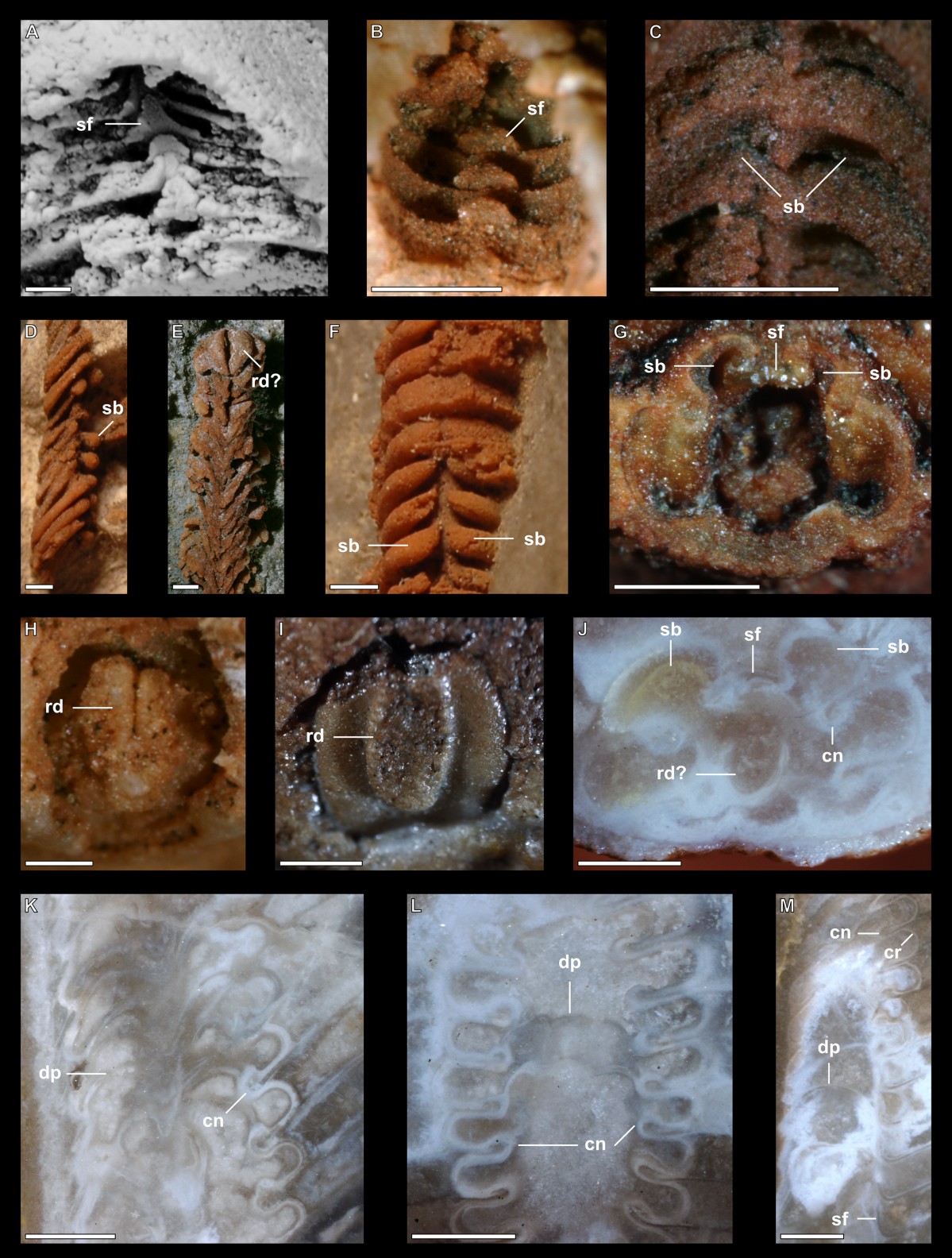

**Figure 7 Siphuncle details of *Sinoeremoceras marywadeae* sp. nov. from the lower Ninmaroo Formation, Unbunmaroo Member (BMT 1), Black Mountain, near Boulia, Queensland, Australia.** (A) QMF 39529, holotype, ventrapertural view of external mould. Note the middorsally

**Figure 7 (continued)**
elongated, partly overlapping septal necks (septal flap). (B) QMF 61634, paratype, ventrapertural view of external mould. Ontogenetically younger stage of septal flaps, where they are shorter and non-overlapping. (C) QMF 61468, ventral view of external mould. The septal flaps are missing due to erosion, exposing the imprint of the siphuncular bulges. (D) QMF 39528, lateral view of isolated internal mould. (E) QMF 13998, ventral view of isolated internal mould. Note the bilaterally symmetrical apical part, corresponding to a diaphragm. (F) QMF 61293, dorsal view of isolated internal mould. Note the paired discontinuous siphuncular bulges. (G) QMF 39523, apical view of natural cross-section, exposing two consecutive segments and the septal flap. (H) QMF 61634, paratype, apical view. (I) QMF 13332, apical view. (J) QMF 13991, cross-section. (K) QMF 13992, median section, slightly off centre. (L) QMF 13332, coronal section, close to ventral shell wall, reproducing a "*Protactinoceras*"-like morphology. (M) QMF 14006, median section. Note transformation from seemingly cyrtochoanitic to orthochoanitic and nearly holochoanitic septal necks, resulting from a slightly misplaced plane of section. Abbreviations: cn = cyrtochoanitic septal neck, cr = connecting ring, dp = diaphragm, rd = median ridge of diaphragm, sb = siphuncular bulb, sf = septal flap.

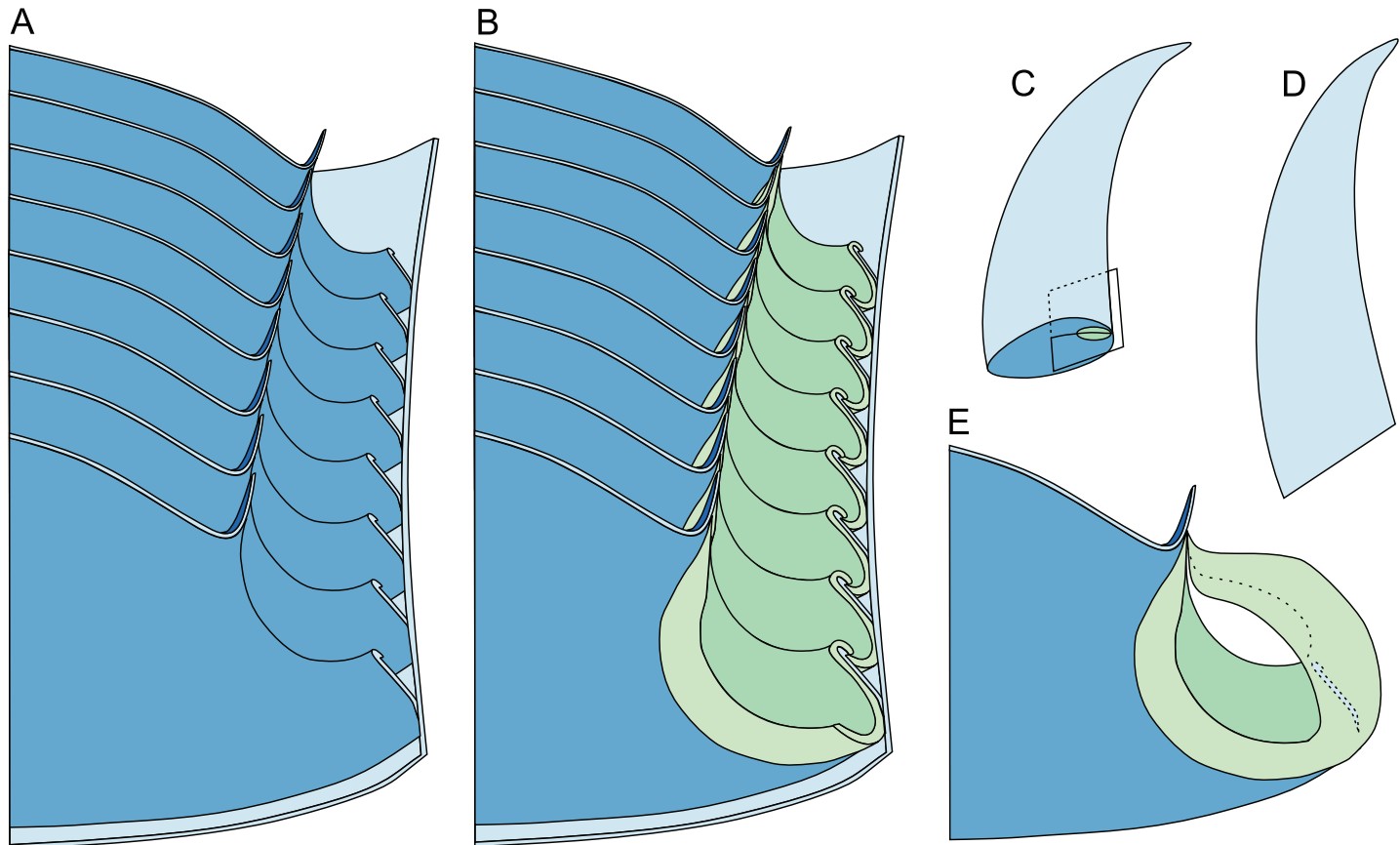

**Figure 8 Reconstruction of the three-dimensional shape of the siphuncle of *Sinoeremoceras marywadeae*.** Redrawn and modified after *Wade & Stait (1998*, fig. 12.8). (A) Median section through phragmocone, without connecting rings. (B) Median section through phragmocone, with connecting rings. (C) Apertural-lateral view of conch, position of siphuncle and section indicated. (D) Lateral view of conch. (E) Single septum with single siphuncular segment, including complete connecting ring.

ventrally (Fig. 7E), which is another indication that the septal necks are unlikely to be cyrtochoanitic throughout.

The deviation from typical cyrtochoanitic septal necks is more obvious in the dorsal area of the siphuncle. Towards the dorsal side of the foramen, the septal necks are elongate adapically, resulting in a triangular shape (Figs. 7A, 7B). Mary Wade referred to this structure as a "septal flap", a term which is followed here. There is some overlap between

successive septal flaps (Fig. 7A), creating a holochoanitic or even macrochoanitic appearance of the septal necks. However, smaller specimens appear to have shorter septal flaps (Fig. 7B), suggesting an ontogenetic trend towards an elongation of the septal flap, although this is currently difficult to assess due to the limited amount of material preserving septal flaps and different states of preservation. We found no indication of clear growth-independent morphological groups of septal flaps and as there are no other distinguishing characters, we regard those specimens as growth stages of one species. Note that in terms of shape, the septal flaps do not represent a transition from cyrtochoanitic to orthochoanitic septal necks, but rather consist of tilted cyrtochoanitic septal necks directed laterally rather than adapically as evident from natural and artificial cross-sections of the siphuncle (Figs. 7G, 7J). Correspondingly, the segments expand laterally, creating disconnected siphuncular bulges dorsolaterally of the septal flap (Figs. 7F, 7G, 7J).

Most specimens are preserved with at least part of the siphuncle exposed, due to its position on the ventral shell margin. In specimens, where the siphuncle is exfoliated (*i.e.*, where an external mould is evident), there are essentially two types of preservation. In the first, the septal flaps are preserved on the dorsal side of the siphuncle and may overlap each other (Figs. 7A, 7B), while the siphuncular segments are visibly expanded into the chambers. In the second type, a small ridge is present middorsally, with traces of the laterally expanded siphuncle but no visible septal flap (Fig. 7C). These different appearances might lead one to conclude that the original siphuncles were structurally different and thus could be used for species discrimination. However, the ridges in the second type likely represent the imprints of the ends of the siphuncular bulges, as they fit very closely to internal moulds of the siphuncle as seen from the dorsal side (Fig. 7F). Here, the converging siphuncular bulges leave a small gap middorsally, which would result in exactly this ridge. This can also be seen in a heavily corroded specimen, where both preservation types occur in the same individual (Fig. 6D).

Imprints of diaphragms can be seen at the adapical end of several phragmocone fragments (Figs. 7H, 7I), as well as in sectioned specimens (Figs. 7J–7M). The diaphragms are convex towards the apex, with a central ridge that is about as wide as the septal flap, but not extending to the venter. While the lateral parts of the diaphragm appear to be parallel with the septum, the central ridge is more perpendicular to the growth axis, thus traversing at least one or two further adapical segments. Consequently, the lateral parts of the diaphragm appear to slope ventraperturally in longitudinal sections (Fig. 7K) while the ridge is directly transverse (Fig. 7M). In another specimen, the ridge is crossed by a narrow furrow along the median plane, passing from the dorsum to about the middle of the diaphragm, thus splitting the dorsal part of the diaphragm essentially in half (Fig. 7H). At the dorsal end, the two halves of the ridge diverge, which leaves a triangular space likely corresponding to the mould of the septal flap. In some specimens, the central furrow reaches across the entire ridge, to the ventral side of the siphuncle (Fig. 7E). The influence of taphonomy is not entirely clear in this case, both specimens are somewhat corroded, allowing for the possibility that the rest of the furrow is simply not preserved, nor is the influence of growth clear, with the two specimens representing different ontogenetic stages. The diaphragms reported here are structurally complex, and their common

characterisation as "concave", "conical", "directly transverse" or other simple terms are probably not enough to capture their variability, as different structures may appear from differently oriented sections of the same species or even specimen.

## Ontogenetic trajectories

We assign 210 specimens from horizon BMT1 to *Sinoeremoceras marywadeae* sp. nov. The ontogenetic trajectories of conch parameters show continuous variation among the material, making it impossible to use them for species delimitation. The conch width index (CWI) is constant throughout ontogeny, with some variability, which may partly be due to suboptimal preservation or weathering (Fig. 4A). Cameral length is usually less than 1 mm throughout ontogeny so promoting a decreasing relative conch width (RCL) (Fig. 4B). The relative size of the siphuncle (RSD) apparently slightly decreases during ontogeny (Fig. 4C); however, this could be a taphonomic artefact, as larger specimens tend to be more heavily corroded, making it difficult to measure their siphuncle. Weathering and the associated uncertainties in measurements are also the likely causes for the relatively high variation in siphuncle size. Expansion rate decreases with a relatively high variation (Fig. 4D), very likely another consequence of weathering, notwithstanding the difficulties of measuring expansion rate in cyrtoconic specimens and the relatively high error potential when calculating conch angles (*Pohle & Klug, 2018*).

When compared to co-occurring ellesmeroceratids, the trajectories differ significantly ($p$-value for two-sample t-test of (constant) CWI < 0.001 and $p$-values of ANOVA of the slope of the linear regressions of RCL, RSD and $ER_h$ < 0.001), although they are partially overlapping (Fig. 4). Ellesmeroceratids from Black Mountain are easily distinguished by their invariably tubular siphuncle. Ellesmeroceratids have a very slightly wider conch cross-section (Fig. 4A), longer cameral lengths (Fig. 4B), a narrower siphuncle (Fig. 4C) and a slower expansion rate that is only slightly decreasing during ontogeny (Fig. 4D).

To investigate the potential impact of small differences in alignment of the plane of section, we compared the ontogenetic trajectories of both halves of a longitudinally sectioned specimen of *Sinoeremoceras marywadeae* sp. nov. (QMF 13992) and two thin sections of previously described protactinoceratids assigned to *Sinoeremoceras foliosum* Chen & Qi in *Chen et al., 1979a* (NIGP 46128) and *Physalactinoceras qiushugouense Chen & Teichert, 1983* (NIGP 73801), respectively (Fig. 9). The sections of QMF 13992 are only about 0.5 mm apart, but the variation seen between them is about as large as the variation seen between the two other species, which supposedly belong to separate genera. While the ontogenetic trajectories of cameral length and siphuncular diameter (Figs. 9B, 9C) are very similar in all four specimens (counting the two halves of QMF 13992 as separate specimens), the expansion rate is higher and decreases more slowly in the Chinese specimens than both halves of the Australian specimen (Fig. 9D). Applying ANOVA to the regression coefficient of the conch parameters reveals that the slopes of RCL and RSD in the Chinese and Australian specimens are significantly different ($p$-value < 0.001), but not for $ER_h$ ($p$-value = 0.64), which is likely caused by the large scatter of $ER_h$ in comparison to RCL and RSD.
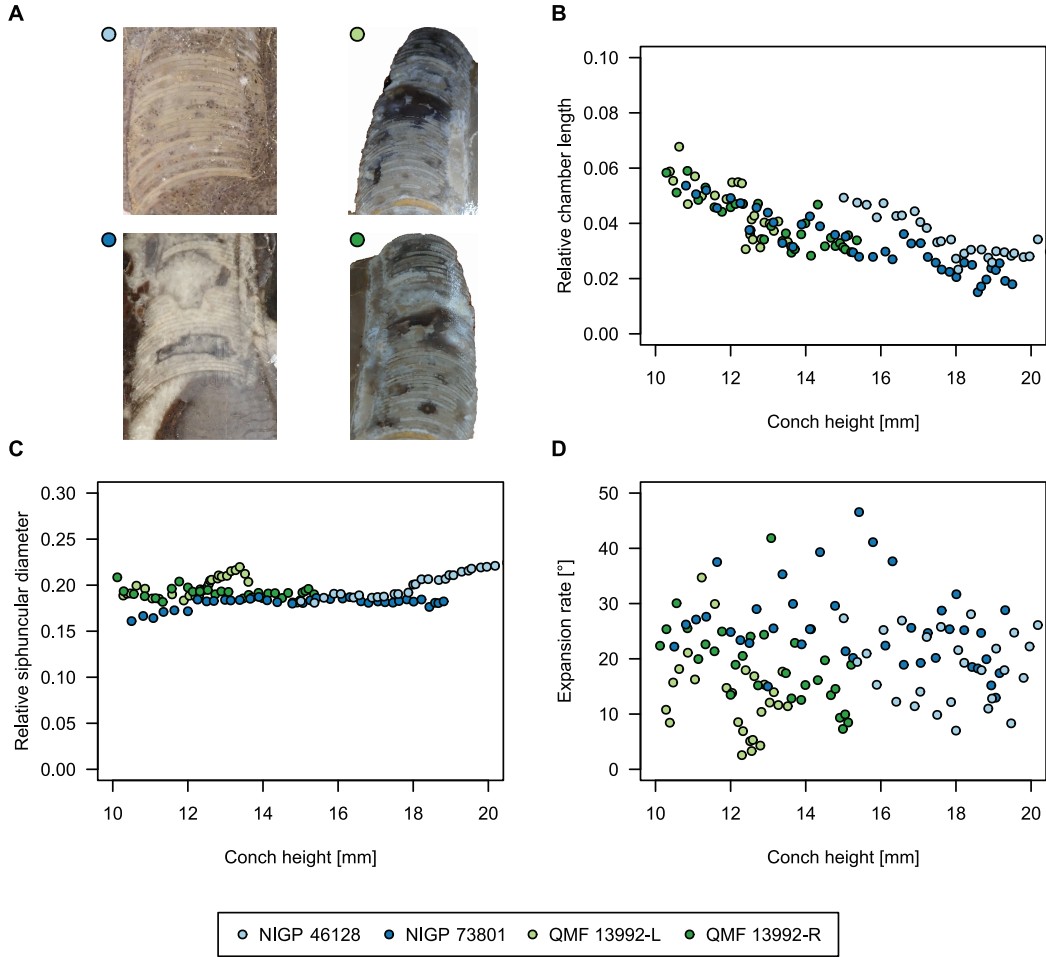

**Figure 9** **Influence of plane of section on conch parameters of sectioned specimens throughout ontogeny (represented by conch height).** (A) Measured specimens: NIGP 46128, *Sinoeremoceras foliosum* Chen & Qi in *Chen et al., 1979a* (= *Sinoeremoceras bullatum* (Chen & Qi in *Chen et al., 1979a*)). NIGP 73801; *Physalactinoceras qiushugouense* *Chen & Teichert, 1983* (= *Sinoeremoceras wanwanense* (*Kobayashi, 1931*)); QMF 13992, *Sinoeremoceras marywadeae* sp. nov., left and right half, respectively, to compare minimal misalignments in the plane of section of the same individual. (B) Relative cameral length (RCL). (C) Relative siphuncular diameter. (D) Height expansion rate (ER_h).

In general, the ontogenetic patterns of Cambrian plectronoceratids from elsewhere in the world are congruent with those seen in *S. marywadeae* sp. nov., *i.e.*, constant CWI, decreasing RCL and ER, and constant RSD, although the latter is somewhat more variable (Fig. 10). Ontogenetic trajectories show some overlap between different localities and stratigraphic ages, but the inferred slopes and intercepts of RCL and ER are distinct, though not always significantly (Tables S1–S4). Stratigraphically older specimens, *e.g.*, from the *Ptychaspis-Tsinania* or the *Quadraticephalus* zones in North China, show a more rapid decrease in RCL and lower ER at comparable conch diameters. However, when including geographic comparisons, the picture becomes more complex. Specimens of *Plectronoceras cambria* (*Walcott, 1905*), the oldest cephalopod, are relatively similar to each other in their conch parameters regardless of their provenance. In fact, some of the

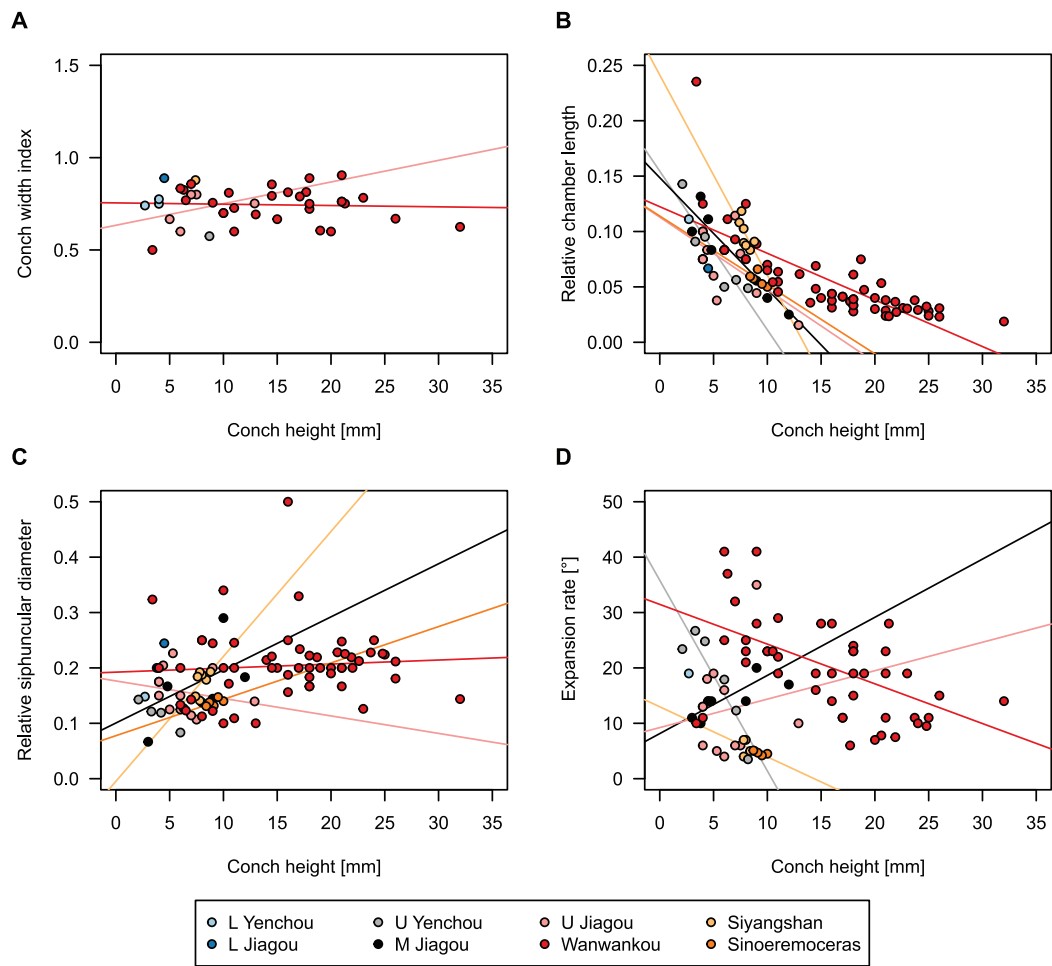

**Figure 10 Conch parameters of previously described plectronoceratids from China throughout ontogeny (represented by conch height) by regional stratigraphic unit.** The specimens from the lower Yenchou and lower Jiagou Members represent *Plectronoceras*, those from the San Saba Member are *Palaeoceras*, while all other specimens are assigned to *Sinoeremoceras*. Note that most points represent individual species or even genera according to previous interpretations. For comparison with *S. marywadeae* sp. nov., see Fig. 4. (A) Conch width index (CWI). (B) Relative cameral length (RCL). (C) Relative siphuncular diameter (RSD). (D) Height expansion rate (ER$_h$).

differences may even be attributable to rounding errors. Comparing between regions in the next youngest *Quadraticephalus* Zone is challenging because only two relatively poorly preserved specimens of one species have been described outside northern Anhui, *S. (?) shanxiense* (*Chen & Teichert, 1983*) from Shanxi Province. In terms of conch parameters, the specimens from Shanxi fall within the variation seen in the material from Anhui, although they are more strongly curved. Regional differences are much more obvious in the Wanwankou Member of the Fengshan Formation and its equivalents, which represents the highest and most fossiliferous (in terms of cephalopods) plectronoceratid-bearing horizon in China. Variability is highest, both between and within regions, including occurrences in Laurentia and Siberia, which are assumed to be roughly equivalent in age (*Flower, 1964*; *Fang et al., 2019*; *Dzik, 2020*). Specimens from Shandong (here regarded as

belonging to *S. bullatum* (Chen & Qi *in Chen et al., 1979a*) and *S. sinense* (Chen & Qi *in Chen et al., 1979a*)) and from Liaoning (interpreted as *S. wanwanense* (*Kobayashi, 1933*)) show very similar trajectories (*p*-values for regression coefficient of RCL = 0.07 and $ER_h$ = 0.66 between *S. bullatum* and *S. wanwanense*), but approximately contemporaneous plectronoceratids from Zhejiang, South China (here attributed to *S. endogastrum* (*Li, 1984*)), have a much lower expansion rate and more rapid ontogenetic reduction in RCL. In this regard, the single specimen described from Inner Mongolia as *S. magicum* Chen *in Lu, Zhou & Zhou, 1984*, is closer to the Zhejiang plectronoceratids than to those from Liaoning and Shandong. The Laurentian plectronoceratids (here all assigned to *Palaeoceras mutabile Flower, 1954*) differ from either of the two previous groups, as they show the lowest expansion rate and steepest ontogenetic trajectory of RCL of all plectronoceratids investigated here. The single species known from the Ust-Kut Formation of Siberia, *S. sibiriense* (*Balashov, 1959*), displays similarities to the plectronoceratids of Liaoning and Shandong, although its expansion rate remains lower throughout ontogeny.

In summary, although the range of variation across all specimens is considerable, they do not fall into distinct groups, making species diagnoses based on small differences in those ratios very doubtful. Importantly, there is a general trend of reduction in RCL and ER, which means that ontogenetic changes must be considered when assessing their diagnostic potential. Differences between regional or temporal populations are usually larger than differences within a single population.

## Body size

Comparing body size reveals temporal and spatial differences (Fig. 11). The largest specimens come from the Wanwankou Member of Shandong (mean = 20.8 mm, *n* = 20), followed by those from contemporaneous beds of Liaoning (mean = 15.5 mm, *n* = 36). In both provinces, many specimens are documented with preserved body chambers (Shandong mean body chamber diameter = 23.1 mm, *n* = 7; Liaoning mean body chamber diameter = 18.3 mm, *n* = 16) and thus, these size distributions are representative for their time of death, even if they may include juvenile specimens. The single known plectronoceratid from the Ust-Kut Formation of Siberia falls within the upper half of the Wanwankou size spectrum but does not have its body chamber preserved (maximum diameter = 24 mm). Plectronoceratids from the Fengshan Formation of northern Anhui are distinctly smaller, reaching average diameters of only 6.5 mm, with specimens from the lower *Quadraticephalus* Zone (6.7 mm, *n* = 5; mean body chamber diameter = 7.2 mm, *n* = 4) and upper *Quadraticephalus Zone* (7.3 mm, *n* = 3; all with body chamber) of the middle Jiagou Member reaching slightly larger sizes than those from the upper Jiagou Member (*Acaroceras-Eburoceras* Zone, approximately equivalent to the Wanwankou Member, 6.0 mm, *n* = 13; mean body chamber diameter = 5.8 mm, *n* = 8). Comparable sizes are reached by the few specimens known from the Siyangshan Formation of Zhejiang (mean = 8.4 mm, *n* = 2) and the *Sinoeremoceras* Zone of Inner Mongolia (10 mm, *n* = 1), both of which have been correlated with the Wanwankou Member, although the Siyangshan plectronoceratids are likely slightly older, corresponding to the *Lotagnostus americanus* Zone (*Peng et al., 2012*). The plectronoceratids from Texas are similarly small

A

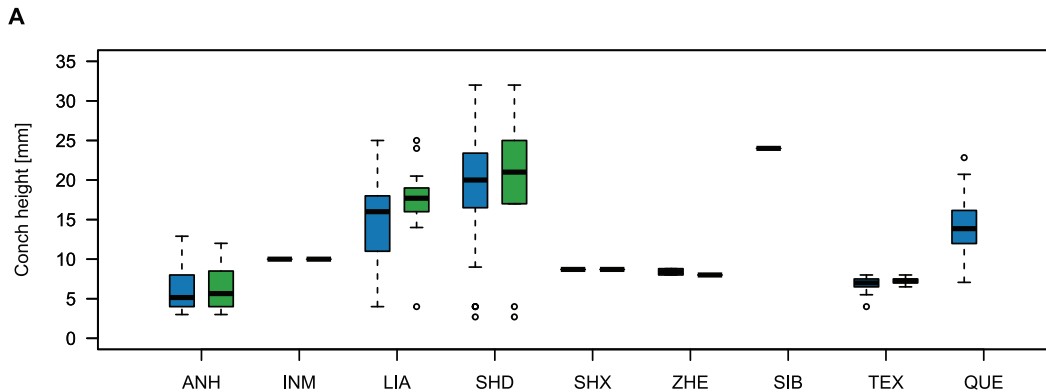

B

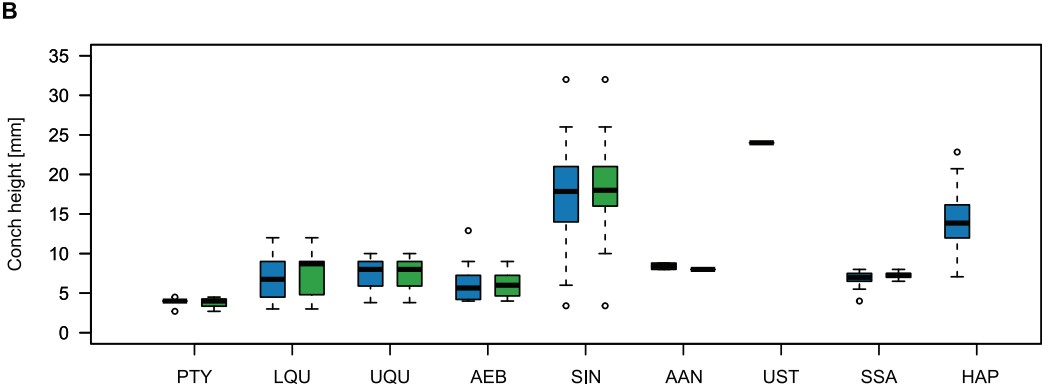

**Figure 11 Size comparison between Cambrian plectronoceratids from China, Siberia and Australia.**
(A) Maximum conch height (blue) and height of the body chamber (green) in comparison with geography. (B) The same, but in comparison with stratigraphic horizon. Abbreviations: ANH, Anhui, North China; INM, Inner Mongolia, North China; LIA, Liaoning, North China; SHD, Shandong, North China; SHX, Shanxi, North China; ZHE, Zhejiang, South China; SIB, Siberia, Russia; TEX, Texas, USA; QUE, Queensland, Australia; PTY, *Ptychaspis-Tsinania* Zone; LQU, lower *Quadraticephalus* Zone; UQU, upper *Quadraticephalus* Zone; AEB, *Acaroceras-Sinoeremoceras* Zone; SIN, *Sinoeremoceras* Zone; AAN, *Acaroceras-Antacaroceras* Zone; UST, Ust-Kut Formation; SSA, San Saba Member; HAP, *Hirsutodontus appressus* Zone.

(6.8 mm, $n = 12$; mean body chamber diameter = 7.3 mm, $n = 6$), although the known size range is narrower (all body chambers with diameters 6.5–8.0 mm). Specimens of *Plectronoceras* are the smallest of all plectronoceratids, reaching average diameters of only 3.8 mm ($n = 5$; all except one with body chamber). They are restricted to the *Ptychaspis-Tsinania* Zone of North China, but their size does not vary between regions; specimens from Anhui, Liaoning and Shandong are of almost identical size. The Queensland material is intermediate between the small Anhui plectronoceratids and the large Wanwankou plectronoceratids, with an average conch diameter of 14.1 mm. However, in contrast to other regions, specimens with preserved body chambers are rare, and thus the average adult size is probably slightly larger. It is not clear whether the scarcity of body chambers in the Black Mountain material is a taphonomic effect or represents a collection bias, although the sheer amount of material collected by Mary Wade and colleagues indicates that they did not discriminate based on preservation status. In any case, the few preserved

body chambers in the material at hand suggest that our body size sample does not represent a gross underestimation. Furthermore, the trajectories of RCL are steeper than in the material from Liaoning and Shandong, but shallower than in specimens from Anhui, suggesting that this parameter approaches similar values in late ontogenetic stages and thus may serve as a tentative indication of adulthood in plectronoceratids.

## DISCUSSION

Although *Kobayashi (1933*, *1935)* already noticed the peculiarity of the plectronoceratid siphuncle, he and subsequent generations of cephalopod workers never fully anticipated the highly unusual 3D structure (*Flower, 1954*, *1964*; *Chen et al., 1979a*; *Chen & Teichert, 1983*). Originally, the key character defining the Plectronoceratidae was the so-called siphuncular bulb (*Kobayashi, 1933*, *1935*) in *Sinoeremoceras* and *Multicameroceras*, and notably also in a single siphuncular segment of *Plectronoceras liaotungense Kobayashi, 1935*. This single fragment has caused much debate. Some workers doubted its biological origin or regarded it as a taphonomic artefact (*Ulrich & Foerste, 1933*; *Miller, 1943*; *Webers & Yochelson, 1989*; *Webers, Yochelson & Kase, 1991*), while others accepted it after initial doubts and used it to derive the Discosorida directly from the Plectronoceratida because *Ruedemannoceras Flower, 1940* seemingly had a similar structure (*Flower, 1954*, *1964*; *Flower & Teichert, 1957*). Yet another interpretation of the siphuncular bulb was that the connecting rings were flexible and were post-mortem sucked into the chambers by invading sediment (*Flower, 1954*; *Yochelson, Flower & Webers, 1973*; *Dzik, 2020*). In the debate on the nature of the connecting rings, there were also different opinions regarding the shape of the septal necks, which were described as cyrtochoanitic (*Ulrich & Foerste, 1933*; *Kobayashi, 1935*), orthochoanitic (*Miller, 1943*; *Ulrich et al., 1944*; *Chen & Teichert, 1983*), simply "short" (*Flower, 1954*, *1964*) or "variable, from orthochoanitic to hemichoanitic or cyrtochoanitic" (*Furnish & Glenister, 1964*). This interesting detail did not receive much attention, although it represents a strong link between *Plectronoceras* and many younger plectronoceratids. An elegant solution to the apparent disagreement on the shape of the septal necks in *Plectronoceras* is that they are similar to those seen in *Sinoeremoceras*, having a septal flap that may be seen as hemichoanitic, orthochoanitic or cyrtochoanitic depending on the plane of section. This similarity may also suggest that the contested connecting ring of *Plectronoceras* is real, as it strongly resembles those expected in *Sinoeremoceras* in a similar plane. Since *Sinoeremoceras* is traditionally assigned to the Protactinoceratida, this also confirms that the order is synonymous with the Plectronoceratida (*Mutvei, Zhang & Dunca, 2007*; *Pohle et al., 2022*).

First doubts on the validity of the Protactinoceratida were raised by *Dzik (1984)*, who suggested that of the 54 named species *in Chen et al. (1979a*, *1979b*) from the Fengshan Formation only three were recognisable, namely *Multicameroceras zaozhuangense* (interpreted as including all protactinoceratids and most plectronoceratids), *Ellesmeroceras elongatum* (interpreted as including most ellesmeroceratids and some plectronoceratids) and *Euroceras jiagouense* (interpreted as referring to strongly curved ellesmeroceratids). However, he did not consider the detailed structure of the siphuncle and accepted the seemingly strong ontogenetic changes in the siphuncle of
plectronoceratids (*Dzik, 1984*, p. 15). His taxonomic treatment of the Chinese Cambrian cephalopods was somewhat superficial, as he assigned specimens to *Multicameroceras* solely because of the "swollen" connecting ring. In his classification, the Ellesmeroceratina containing the Plectronoceratidae and Ellesmeroceratidae was regarded as a suborder of the Endoceratida. We demonstrate considerable differences between plectronoceratids from the Fengshan Formation in different regions of China, which contradicts *Dzik*'s *(1984)* hypothesis that most of them belong to *Multicameroceras zaozhuangense* (Chen & Qi *in Chen et al., 1979a*). *Dzik (1984)* overlooked the oldest available name for the species in question, *Sinoeremoceras wanwanense* (*Kobayashi, 1931*) and *Multicameroceras Kobayashi, 1933* has been made a subjective junior synonym of *Sinoeremoceras Kobayashi, 1933* by *Chen & Teichert (1983)*. *Dzik (2020)* raised the possibility that the connecting rings of plectronoceratids were poorly calcified and elastic. He hypothesised that the expanded segments were caused by lowered pressures that sucked the connecting ring into the chambers. The regular, bilaterally symmetrical morphology of subsequent segments in the Australian material suggests that the connecting rings were calcified and not flexible in plectronoceratids, as was already shown by the ultrastructure of the connecting ring (*Mutvei, Zhang & Dunca, 2007*).

Understanding the siphuncle in three dimensions in the Australian material, *Wade (1988)* and *Wade & Stait (1998)* suggested that seemingly different genera of protactinoceratids were based on misaligned sections, although this material was never formally described nor figured. Mary Wade mentioned the similarity between plectronoceratids and protactinoceratids but did not consider them synonymous. However, from her letters and notes, it is evident that she considered the possibility that *Plectronoceras* is in fact a juvenile "*Protactinoceras*", implying synonymy.

*Mutvei, Zhang & Dunca (2007)* and *Mutvei (2020)* concluded that the Plectronoceratida are identical with the Protactinoceratida, without having three-dimensionally preserved material at hand. Our new material confirms their conclusion, and clarifies some of the earlier misconceptions that were presented in the absence of well-preserved material. *Mutvei, Zhang & Dunca (2007)* synonymised the orders, but retained the Plectronoceratidae and the Protactinoceratidae. However, the original diagnoses of Protactinoceratida and Protactinoceratidae were identical (*Chen et al., 1979a*), so synonymy of the two families was a certain consequence. Although *Mutvei (2020)* partially reversed his earlier opinion, regarding *Protactinoceras* as the only valid genus within the Protactinoceratidae and Protactinoceratida, ventrally polished specimens from Black Mountain demonstrate that it is possible to recreate the "*Protactinoceras*" outline by a very strong z-rotation (Fig. 7L). In fact, the supposedly "central" siphuncle of *Protactinoceras* is not demonstrated in any cross-section of a Cambrian cephalopod from China or Australia, in all of which the siphuncle touches the shell wall ventrally. "*Protactinoceras*" specimens show other indications that the plane of section is not in the median plane, such as the seemingly strong adapertural decrease in siphuncle size (Fig. 12A). Thus, we consider it probable that sections attributed to "*Protactinoceras*" display a considerable degree of z-rotation and the only way to unequivocally demonstrate that "*Protactinoceras*" is not a

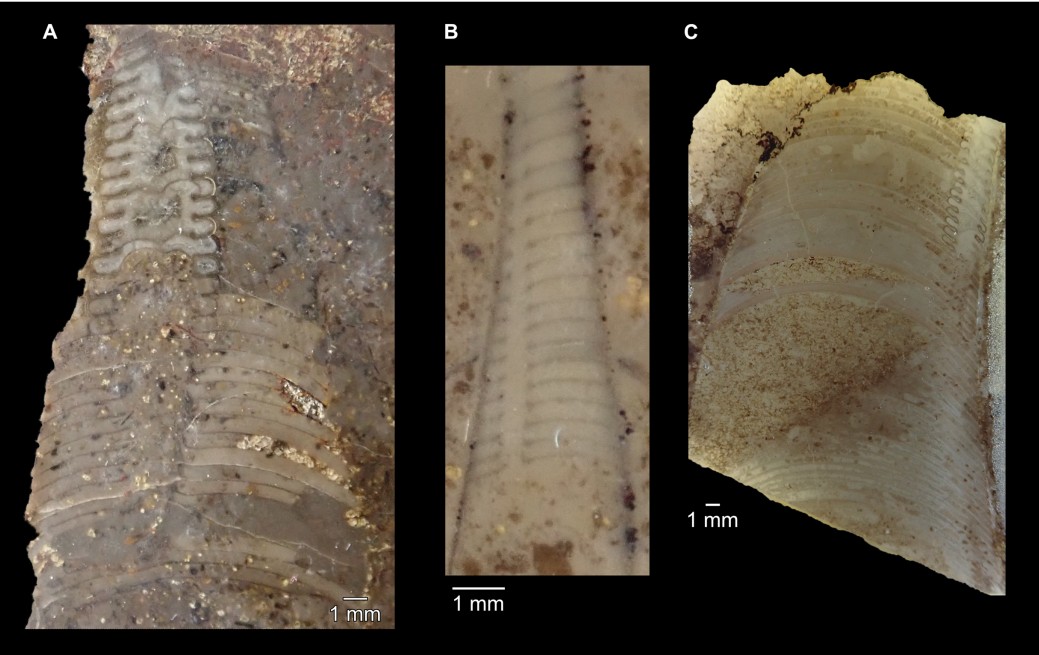

**Figure 12 Indications for misaligned sections of plectronoceratids from the late Cambrian Fengshan Formation of North China.** (A) NIGP 46133, seemingly strong ontogenetic decrease in siphuncle size eventually leading to complete disappearance (y-rotation) and apparent central siphuncle position (strong z-rotation). Originally designated as holotype of *Protactinoceras magnitubulum* Chen & Qi *in Chen et al., 1979a*, likely junior synonym of *Sinoeremoceras bullatum* (Chen & Qi *in Chen et al., 1979a*). (B) NIGP 46184, extremely flat septa, initially negative expansion rate and strong ontogenetic shift in siphuncle position. Originally designated as holotype of *Rectseptoceras eccentricum* Zou & Chen *in Chen et al., 1979a*, likely junior synonym of *S. inflatum* (Chen & Zou *in Chen et al., 1979a*). (C) NIGP 73860, Disappearance of siphuncle (y-rotation) and septal flap not visible (x-translation). Originally designated as holotype of *Physalactinoceras compressum Chen & Teichert, 1983*, junior synonym of *S. bullatum* (Chen & Qi *in Chen et al., 1979a*).

result of a misaligned section would be to present a cross-section of a plectronoceratid with a siphuncle that is demonstrably removed from the shell wall (subventral).

As can be seen from the above summary, the distinction between plectronoceratids and protactinoceratids (both the orders and the families) has been questioned for some time, but ultimate proof in the form of three-dimensionally preserved material has been lacking. In addition, the wide variation in longitudinal sections of protactinoceratids prevented recognition of the complex three-dimensional morphology of the siphuncle. In revising the taxonomy of these groups, it is thus necessary to go back to the original definition of the Protactinoceratida and consider whether their diagnostic characters represent biological variation or whether these differences can be explained by misalignment of the plane of section. According to *Chen & Teichert (1983,* p. 74), the Protactinoceratida "resembles the Plectronoceratida in most features, differing, however, from the latter in its much larger siphuncle with much more strongly expanded segments and its more advanced diaphragms and development of calcite fillings in the spaces between diaphragms." Considering each of these characters separately, we conclude:

- *Siphuncle size*: No sharp boundary exists between smaller and larger siphuncles (Fig. 10C), but rather a continuous distribution between the two extremes. Siphuncle size is easily influenced by misaligned sections. Thus, the distinction between plectronoceratids and protactinoceratids cannot be based on siphuncle size.

- *Expanded segments*: As with siphuncle size, the expansion of segments can be misleading if the section is misaligned. For example, a section that is exactly aligned with the median plane will have very little expansion, as it passes through both the septal flap and the midventral position where the septal foramen meets the shell wall. Likewise, a section with a strong z-rotation will go through the lateral parts of the siphuncle, which are more strongly expanded. This includes sections that go through the septal flap but also through the siphuncular bulge immediately dorsolaterally. Thus, strongly expanded siphuncular segments may also be excluded from the list of diagnostic characters.

- *More advanced diaphragms*: Besides the very vague term "advanced", which additionally implies directionality in the evolution of the diaphragms, the structure of the diaphragms in the Australian material suggests that variation in alignment of the plane of section can result in very different shapes. For example, a section through the median plane would cross the central ridge, resulting in a "simple" concave diaphragm, while an x-translated section would pass through the lateral parts of the diaphragm, thus seemingly sloping more steeply ventraperturally. Any section that passes over the ridge of the diaphragm will appear "complex" and the shape of the ridge with the central furrow closely corresponds to the ω-shape of "*Protactinoceras*". Consequently, no unambiguous difference in the shape of the diaphragms exists between plectronoceratids and protactinoceratids that could not be produced by misalignment of the plane of section.

- *Calcite fillings between the diaphragms*: Even *Chen & Teichert (1983)* were doubtful about the biogenic origin of these structures. We agree with *Wade (1988)* that these are taphonomic artefacts. Furthermore, *Mutvei, Zhang & Dunca (2007)* found no difference in the calcite fillings between diaphragms of protactinoceratids and plectronoceratids. An organic origin of these structures would have to be demonstrated first, *e.g.*, by growth lines or geochemical indicators. Even if the fillings are of organic origin, *Mutvei, Zhang & Dunca (2007)* showed that they cannot serve to distinguish Protactinoceratida from Plectronoceratida.

Distinction between the two orders (and families) is thus impossible based on any of these characters. A section through the median plane or parallel to it with slight x-translation will in most cases result in a typical "plectronoceratid-shape". In contrast, a section with considerable z-rotation will more closely resemble a typical "protactinoceratid-shape".

Identifying generic and specific discriminators requires assessment of which characters are considered as diagnostic, and which characters potentially represent only variation in alignment of the plane of section. The following criteria can be used to identify whether the plane of section differs markedly from the median plane and at the same time provide guidance as to which characters should be treated with caution:

- *Exposure of the siphuncle*: In the longitudinal median section, the siphuncle has to be well exposed over the entire length of the specimen. Changes in the size of the siphuncle or the shape and length of the septal necks are expected to be minimal within just a few chambers. However, deviations of the plane of section from the median plane can result in apparent rapid changes. Good examples are *Protactinoceras magnitubulum* Chen & Qi *in Chen et al., 1979a* (text-fig. 10, pl. 1, fig. 3, pl. 2, fig. 5, pl. 3, figs. 12, 13) or *Physalactinoceras breviconum* Chen & Qi *in Chen et al., 1979a* (text-fig. 13, pl. 1, fig. 7). In both examples, the siphuncle "disappears" adaperturally, an indication that the plane of section is misaligned with the median plane. Although a slight decrease in the relative size of the siphuncle of the Australian species is apparent, the absolute size of the siphuncle constantly increases, and the decrease in RSD may at least partially be related to preservation (see systematic description of *S. marywadeae* sp. nov.).

- *Siphuncle position*: A seemingly central siphuncular position in a section of a phragmocone with a strictly marginal siphuncle can be achieved by a plane of section with a strong z-rotation, with additional slight y-rotation and/or x-translation so that the plane of section is more or less parallel to the siphuncle. Although a few species of Cambrian cephalopods have been described with siphuncles that seem to be removed from the venter, there is no known cross-section of a Cambrian cephalopod that shows a siphuncle demonstrably and undoubtedly removed from the shell wall. Apparent submarginal siphuncles described in early Cambrian elongate conical shells suggested to be cephalopods by *Hildenbrand et al. (2021)* are not accepted. These are more likely hyoliths "containing invaginated *Coleoloides* tubes" (*Landing et al., 2023*, p. 3). Furthermore, the Australian Cambrian cephalopods (plectronoceratids and ellesmeroceratids) known from hundreds of specimens with exposed siphuncles have exclusively marginal siphuncles. The same is true for earliest ordovician cephalopods (*Ulrich et al., 1944*; *Unklesbay, 1954*; *Unklesbay & Young, 1956*; *Flower, 1964*; *Kröger & Landing, 2007*; *Cichowolski et al., 2023*), thus suggesting that this is a highly conserved plesiomorphic character. Migration of the siphuncle away from the venter probably evolved later, but perhaps independently in multiple cephalopod lineages. Species with apparently submarginal siphuncles need to be confirmed by cross-sections, which is not the case for any of the Chinese species in question. The most obvious examples are all species of *Protactinoceras* (see below) but compare also *Jiagouceras cordatum* Chen & Zou *in Chen et al., 1979a*, text-fig. 5, pl. 4, fig. 17 (note the almost straight septa) or *Recteseptoceras eccentricum* Zou & Qi *in Chen et al., 1979a*, text-fig. 9, pl. 3, fig. 1 (septa almost straight, siphuncle absent in adapical part).

- *Depth of the septal concavity*: Even if a specimen has been cut perfectly through the siphuncle, it is still possible that the plane of section lies in a more ventro-lateral plane instead of dorso-ventral. In this case, very flat septa can give an indication for such misaligned planes of section. A good example is again *Jiagouceras cordatum*.

- *Expansion rate*: This is more difficult to assess because expansion rate may also change naturally during ontogeny, and the ontogenetic trajectory of the Australian material demonstrates a decrease in expansion rate (even though variation is generally high).

However, very rapid ontogenetic changes and especially an adapertural decrease in absolute conch diameter may indicate an x-translated or y-rotated plane of section. If the apical or adapertural end of the section is distinctly rounded and maybe even shows traces of shell wall, these are strong indications of a misaligned plane of section. In *Rectseptoceras eccentricum* Zou & Chen *in Chen et al., 1979a* (text-fig. 9, pl. 3, fig. 1), the conch diameter seemingly decreases before it increases again. This shape can be Explained by the endogastric curvature, where the apical and adapertural parts of the specimens are further removed from the shell wall than the middle part.

- *Conch curvature*: All described protactinoceratid and plectronoceratid genera and species are either straight or slightly curved. The amount of curvature has often been used to distinguish taxa at species level. The Balkoceratidae *Flower (1954)* are remarkable in that they are exogastric. However, the exogastric curvature is so slight that it might just as well be considered as straight. Specimens known only from thin sections might appear exogastric if the plane of section was misaligned, creating an outline that is seemingly more convex on the "ventral" than on the "dorsal" side. Thus, only if the dorsal side is distinctly shown as concave can exogastric curvature be confirmed from thin sections. Furthermore, curvature may be underestimated due to z-rotation: in the most extreme case, a 90° z-rotation could result in a seemingly orthoconic conch, even if the true conch shape is distinctly cyrtoconic.

We base our synonymies on the above considerations, also considering distributions of conch parameters, ontogenetic trajectories and body size distributions when compared between different regions and stratigraphic horizons. We retain *Plectronoceras*, *Palaeoceras* and *Sinoeremoceras*, as most of the characters used to diagnose other genera are likely severely influenced by misalignment of planes of section in specimens with a siphuncle identical with that in *S. marywadeae* sp. nov. While we cannot absolutely rule out the existence of specimens that deviate from the pattern of a cyrtochoanitic septal neck with an elongated middorsal septal flap, such a structure has yet to be demonstrated in any specimen using either three-dimensionally preserved or prepared specimens or reconstructions using imaging techniques such as CT-scanning or serial grinding tomography. However, until a distinctly different structure is demonstrated, and none have been found yet that unequivocally demonstrate a lack of the septal flap, the siphuncle of *S. marywadeae* sp. nov. must be regarded as a null model of an expanded siphuncle in a Cambrian cephalopod. Many previously established genera are thus considered to have been based on highly suspect criteria that do not allow generic discrimination and are here treated as subjective junior synonyms. We keep *Plectronoceras* separate but mainly because of its historical importance in research on fossil cephalopods and because its detailed three-dimensional siphuncular structure is not known beyond doubt. However, since the main character that distinguishes *Plectronoceras* from *Sinoeremoceras* is size, it makes assignment of some of the smaller species (such as *S. inflatum*) to *Sinoeremoceras* somewhat arbitrary. We thus consider this revision as a first step, to consolidate current knowledge on plectronoceratids and strongly advocate careful search for specimens in which three-dimensional structure of the siphuncle can be ascertained to determine

**Table 2 Estimated deviations from the median plane of previously described plectronoceratids and "protactinoceratids".**

| Genus | x-translation | y-rotation | z-rotation |
|---|---|---|---|
| *Eodiaphragmoceras* | 0 | 0 | 0–1 |
| *Benxioceras* | 0–1 | 0 | 1 |
| *Mastoceras* | 0–1 | 1 | 0–1 |
| *Paraplectronoceras* | 0–1 | 1 | 0–1 |
| *Theskeloceras* | 0–1 | 1 | 0–1 |
| *Parapalaeoceras* | 1 | 0 | 0–1 |
| *Physalactinoceras* | 1 | 0–1 | 0–1 |
| *Multicameroceras* | 1 | 1 | 0–1 |
| *Sinoeremoceras* | 1 | 1 | 0–1 |
| *Wanwanoceras* | 1 | 1 | 0–1 |
| *Protactinoceras* | 1 | 1 | 2 |
| *Lunanoceras* | 1–2 | 0 | 0–1 |
| *Jiagouceras* | 1–2 | 0 | 1–2 |
| *Rectseptoceras* | 1–2 | 1 | 2 |

**Note:**
*Plectronoceras, Balkoceras* and *Palaeoceras* are not included because they were not based on longitudinal sections.
Codings: 0, no deviation; 1, small to moderate deviation; 2, strong deviation.

variation and taxonomic potential. *Palaeoceras Flower, 1954* is maintained because it is difficult to assess the accuracy of the three-dimensional reconstructions of the siphuncle (*Flower, 1954*, *1964*). It lacks cyrtochoanitic septal necks but has similarly expanded siphuncular segments, thus probably representing a transitional form between plectronoceratids and ellesmeroceratids. Variation between *Palaeoceras* and *Balkoceras* is much less than that described within *S. marywadeae* sp. nov., so we consider their generic, and very likely even their specific separation unwarranted. The necessary deviations of the plane of section from the median plane to produce the synonymised taxa are listed in Table 2.

Revision at species level is more challenging because three-dimensional structures of the siphuncles are unknown and most species have been erected based on a few longitudinal sections, many of them being monotypic. Across the entire sample of known plectronoceratids and "protactinoceratids", the variability in ontogenetic trajectories of conch parameters does not fall into distinct groups (Fig. 10). However, when comparing specimens from different geographical regions and stratigraphic intervals, they produce overlapping but distinct distributions (Figs. 10, 11). Body size differs, with specimens from Liaoning and Shandong reaching distinctly larger sizes than those from Anhui, Zhejiang and elsewhere. The specimens from Queensland are in between those two size classes, while the Laurentian plectronoceratids are invariably small. Likewise, the *Plectronoceras* specimens from the stratigraphically oldest *Ptychaspsis-Tsinania* Zone are the smallest, while the largest plectronoceratids can be found in the younger Wanwankou Member. Thus, specimens from different regions can be regarded as having distinct size distributions. Admittedly, we do not know whether these distributions represent their

adult size, as many specimens are missing body chambers, particularly in the material from Black Mountain. Nevertheless, the same pattern remains even when taking only body chamber diameters into account, suggesting that the pattern is not driven by size sorting. The remaining question is whether these different populations represent different species. Lacking conclusive evidence on the detailed three-dimensional structure of the siphuncle in most specimens, we propose a pragmatic approach that accepts these populations as separate species, even though it is possible that some populations may contain more than one, similar sized species, or that some populations from separate regions with similar size distribution in reality represent one and the same species. Conveniently, most species consist of only few specimens from the same locality, which requires little reassignment of specimens to different species apart from the synonymies. We keep separate some species that have been proposed on limited material from isolated regions, because further material is needed to demonstrate their taxonomic positions. A few species that display characteristics that cannot be fully explained by inferring misaligned planes of section are also kept separate. This arrangement is not intended as a final verdict on plectronoceratid taxonomy, but as a more meaningful assessment of natural variation between these many specimens and will serve as a baseline for future understanding of this group.

The three-dimensional structure of the siphuncle in plectronoceratids from all previously sampled regions will be required for future refinement. All previously established species and their assignment based on our revised concept are reported in Table 3.

The stratigraphic and geographic distribution of the revised species is shown in Fig. 13. *Fang et al. (2019)* reported that most Cambrian cephalopods from China occur in Stage 10, while *Plectronoceras* is restricted to the late Jiangshanian. However, this depends on the definition of the GSSP for Stage 10. Currently, there are two candidates being discussed, either the LO of *Eoconodontus notchpeakensis* (*e.g.*, *Landing, Westrop & Adrain, 2011*; *Miller et al., 2011*; *Miller et al., 2015*) or the LO of *Lotagnostus americanus* (*e.g.*, *Peng et al., 2014*; *Bagnoli et al., 2017*). The upper Yenchou and the Wanwankou members of the Fengshan Formation in North China, from which the majority of Cambrian plectronoceratids have been reported, correlate with the interval between these two candidates (*i.e.*, approximately between the *Proconodontus posterocostatus* Zone to *Eoconodontus notchpeakensis* Subzone; compare, *e.g.*, *Chen & Teichert, 1983*; *Bagnoli et al., 2017*; *Geyer, 2019*) and would become mostly Jiangshanian if the proposal for *Eoconodontus notchpeakensis* is accepted. The slightly younger *Sinoeremoceras marywadeae* sp. nov. and *Palaeoceras mutabile* are undoubtedly from Stage 10, as they occur in the upper *Eoconodontus* Zone. The age of *S. sibiriense* from the Ust-Kut Formation on the Chunya River is difficult to constrain, but if the Cambrian cephalopods from the same Formation on the Angara River are contemporaneous, their age probably lies somewhere between the *C. proavus* and *I. fluctivagus* zones (*Dzik, 2020*).

The stratigraphic range of plectronoceratids is therefore very short and limited to the transition between the Jiangshanian and Stage 10.

Available evidence of body size distributions suggests that plectronoceratid evolution began with very small specimens, approximately contemporaneously appearing in different regions within North China, but soon thereafter split into independent

**Table 3 List of all previously described members of Plectronoceratida and Protactinoceratida including their origin, type horizon and synonymy status according to this study.**

| Original binomen | Revised species | Reference | Region | Type horizon |
|---|---|---|---|---|
| *Palaeoceras mutabile* | *Palaeoceras mutabile* | *Flower (1954)* | **Texas** | **San Saba** |
| *Balkoceras gracile* | *Pa. mutabile* | *Flower (1964)* | Texas | San Saba |
| *Pa. undulatum* | *Pa. mutabile* | *Flower (1964)* | Texas | San Saba |
| *Plectronoceras exile* | *Pa. mutabile* | *Flower (1964)* | Texas | San Saba |
| *Cyrtoceras cambria* | *Plectronoceras cambria* | *Walcott (1905)* | **Shandong** | **Lower Yenchou (PT)** |
| *Pl. huaibeiense* | *Pl. cambria* | *Chen et al. (1979a)* | Anhui | Lower Jiagou (PT) |
| *Pl. liaotungense* | *Pl. cambria* | *Kobayashi (1935)* | Liaoning | Lower Yenchou (PT) |
| *Physalactinoceras bullatum* | *Sinoeremoceras bullatum* | *Chen et al. (1979a)* | **Shandong** | **Wanwankou** |
| *Lunanoceras changshanense* | *S. bullatum* | *Chen et al. (1979a)* | Shandong | Wanwankou |
| *L. precordium* | *S. bullatum* | *Chen et al. (1979a)* | Shandong | Wanwankou |
| *Ph. breviconicum* | *S. bullatum* | *Chen et al. (1979a)* | Shandong | Wanwankou |
| *Ph. changshanense* | *S. bullatum* | *Chen et al. (1979a)* | Shandong | Wanwankou |
| *Ph. compressum* | *S. bullatum* | *Chen & Teichert (1983)* | Shandong | Wanwankou |
| *Ph. globosum* | *S. bullatum* | *Chen et al. (1979a)* | Shandong | Wanwankou |
| *Ph. longiconum* | *S. bullatum* | *Chen & Teichert (1983)* | Shandong | Wanwankou |
| *Ph. subcirculum* | *S. bullatum* | *Chen et al. (1979a)* | Shandong | Wanwankou |
| *Protactinoceras magnitubulum* | *S. bullatum* | *Chen et al. (1979a)* | Shandong | Wanwankou |
| *Pr. lunanense* | *S. bullatum* | *Chen et al. (1979a)* | Shandong | Wanwankou |
| *Sinoeremoceras foliosum* | *S. bullatum* | *Chen et al. (1979a)* | Shandong | Wanwankou |
| *S. zaozhuangense* | *S. wanwanense* | *Chen et al. (1979a)* | Shandong | Wanwankou |
| *W. lunanense* | *S. bullatum* | *Chen et al. (1979a)* | Shandong | Wanwankou |
| *Parapalaeoceras endogastrum* | *Sinoeremoceras endogastrum* | *Li (1984)* | **Zhejiang** | **Siyangshan** |
| *Ppa. sinense* | *S. endogastrum* | *Li (1984)* | Zhejiang | Siyangshan |
| *Paraplectronoceras inflatum* | *Sinoeremoceras inflatum* | *Chen et al. (1979a)* | **Anhui** | **Upper Jiagou** |
| *Acaroceras primordium* | *S. inflatum* | *Chen & Qi (1982)* | Anhui | Middle Jiagou (LQ) |
| *Jiagouceras cordatum* | *S. inflatum* | *Chen et al. (1979a)* | Anhui | Upper Jiagou |
| *L. compressum* | *S. inflatum* | *Chen et al. (1979a)* | Anhui | Upper Jiagou |
| *L. densum* | *S. inflatum* | *Chen & Qi (1982)* | Anhui | Middle Jiagou (LQ) |
| *L. longatum* | *S. inflatum* | *Chen & Qi (1982)* | Anhui | Middle Jiagou (LQ) |
| *Ppl. abruptum* | *S. inflatum* | *Chen et al. (1980)* | Anhui | Upper Jiagou |
| *Ppl. curvatum* | *S. inflatum* | *Chen et al. (1980)* | Anhui | Upper Jiagou |
| *Ppl. impromptum* | *S. inflatum* | *Chen et al. (1979a)* | Anhui | Upper Jiagou |
| *Ppl. longicollum* | *S. inflatum* | *Chen & Qi (1982)* | Anhui | Upper Jiagou |
| *Ppl. pandum* | *S. inflatum* | *Chen & Qi (1982)* | Anhui | Upper Jiagou |
| *Ppl. parvum* | *S. inflatum* | *Chen et al. (1980)* | Anhui | Middle Jiagou (UQ) |
| *Ppl. pyriforme* | *S. inflatum* | *Chen et al. (1979a)* | Anhui | Upper Jiagou |
| *Ppl. suxianense* | *S. inflatum* | *Chen et al. (1979a)* | Anhui | Middle Jiagou (UQ) |
| *Ppl. vescum* | *S. inflatum* | *Chen et al. (1980)* | Anhui | Upper Jiagou |
| *Rectseptoceras eccentricum* | *S. inflatum* | *Chen et al. (1979a)* | Anhui | Upper Jiagou |
| *S. anhuiense* | *S. inflatum* | *Chen et al. (1979b)* | Anhui | Upper Jiagou |
| *S. pisinum* | *S. inflatum* | *Chen et al. (1980)* | Anhui | Upper Jiagou |

(Continued)

| Original binomen | Revised species | Reference | Region | Type horizon |
|---|---|---|---|---|
| *Wanwanoceras exiguum* | *S. inflatum* | *Chen & Qi (1982)* | Anhui | Middle Jiagou (LQ) |
| *W. multiseptum* | *S. inflatum* | *Chen et al., 1979b* | Anhui | Upper Jiagou |
| ***S. magicum*** | ***Sinoeremoceras magicum*** | ***Lu, Zhou & Zhou (1984)*** | **Inner Mongolia** | ***Sinoeremoceras* Zone** |
| ***Multicameroceras sibiriense*** | ***Sinoeremoceras sibiriense*** | ***Balashov (1959)*** | **Southern Krasnojarsk** | **Ust-Kut** |
| ***Eodiaphragmoceras sinense*** | ***Sinoeremoceras sinense*** | ***Chen et al. (1979a)*** | **Shandong** | **Wanwankou** |
| ***Hunyuanoceras shanxiense*** | ***Sinoeremoceras (?) shanxiense*** | ***Chen & Teichert (1983)*** | **Shanxi** | **Upper Yenchou (LQ)** |
| ***S. wanwanense*** | ***Sinoeremoceras wanwanense*** | ***Kobayashi (1931)*** | **Liaoning** | **Wanwankou** |
| *Benxioceras rapidum* | *S. bullatum* | *Chen & Teichert (1983)* | Liaoning | Wanwankou |
| *Mastoceras obliquum* | *S. wanwanense* | *Chen & Teichert (1983)* | Liaoning | Wanwankou |
| *Ma. qiushugouense* | *S. bullatum* | *Chen & Teichert (1983)* | Liaoning | Wanwankou |
| *Mu. cylindricum* | *S. wanwanense* | *Kobayashi (1933)* | Liaoning | Wanwankou |
| *Mu. multicameratum* | *S. wanwanense* | *Kobayashi (1931)* | Liaoning | Wanwankou |
| *Ph. benxiense* | *S. wanwanense* | *Chen & Teichert (1983)* | Liaoning | Wanwankou |
| *Ph. confusum* | *S. wanwanense* | *Chen & Teichert (1983)* | Liaoning | Wanwankou |
| *Ph. cornutum* | *S. wanwanense* | *Chen & Teichert (1983)* | Liaoning | Wanwankou |
| *Ph. niuxintaiense* | *S. wanwanense* | *Chen & Teichert (1983)* | Liaoning | Wanwankou |
| *Ph. papilla* | *S. wanwanense* | *Chen & Teichert (1983)* | Liaoning | Wanwankou |
| *Ph. planoconvexum* | *S. wanwanense* | *Chen & Teichert (1983)* | Liaoning | Wanwankou |
| *Ph. rarum* | *S. wanwanense* | *Chen & Teichert (1983)* | Liaoning | Wanwankou |
| *Ph. qiushugouense* | *S. wanwanense* | *Chen & Teichert (1983)* | Liaoning | Wanwankou |
| *Ph. speciosum* | *S. wanwanense* | *Chen & Teichert (1983)* | Liaoning | Wanwankou |
| *S. magnum* | *S. wanwanense* | *Chen & Teichert (1983)* | Liaoning | Wanwankou |
| *S. taiziheense* | *S. wanwanense* | *Chen & Teichert (1983)* | Liaoning | Wanwankou |
| *Theskeloceras benxiense* | *S. wanwanense* | *Chen & Teichert (1983)* | Liaoning | Wanwankou |
| *T. subrectum* | *S. wanwanense* | *Chen & Teichert (1983)* | Liaoning | Wanwankou |
| *W. peculiare* | *S. wanwanense* | *Kobayashi (1933)* | Liaoning | Wanwankou |
| *W. peculiare curtum* | *S. wanwanense* | *Chen & Teichert (1983)* | Liaoning | Wanwankou |

**Note:**
The list is alphabetically sorted after revised genus first, followed by accepted species (highlighted in bold). Junior synonyms are listed alphabetically after their respective senior synonyms. Abbreviations: PT, *Ptychaspis-Tsinania* Zone; *LQ*, lower *Quadraticephalus* Zone; UQ, upper *Quadraticephalus* Zone.

populations. While plectronoceratids from Anhui, Zhejiang, Inner Mongolia and Texas increased only slightly in size, and specimens from Anhui even show a subtle subsequent decrease, the plectronoceratids from Liaoning, Shandong and Siberia became the largest plectronoceratids, reaching between two and three times the size of their relatives in other regions. The younger plectronoceratids from Stage 10 of Queensland display intermediate size between the former two groups, but since the origin of this population is unknown, it is impossible to conclude whether this represents a body size decrease or increase.

The collection site at Black Mountain, Queensland, and all other Cambrian cephalopod sites lie within tropical palaeolatitudes (Fig. 14; *Kröger, 2013*; *Fang et al., 2019*). Notably, *Sinoeremoceras marywadeae* sp. nov. represents the first record of plectronoceratids from Gondwana. The interpretation of Early Cambrian fossils from the higher latitudes of

**Figure 13 Global stratigraphic distribution of revised plectronoceratid species.** Occurrences are grouped after palaeocontinents: northern Gondwana (Queensland), North China (Anhui, Liaoning, Shandong, Shanxi, Inner Mongolia), South China (Zhejiang), Laurentia (Texas), Siberia. Correlations and conodont zones after *Webby et al. (2004)*, *Bergström et al. (2009)*, *Peng et al. (2012)*, *Miller et al. (2015)*, *Bagnoli et al. (2017)*, *Zhen, Percival & Webby (2017)*, *Geyer (2019)* and *Miller (2020)*. Uncertain stratigraphic ranges are shown as white bars. Note that most plectronoceratids are restricted to an interval close to the yet to be defined boundary between the Jiangshanian and Stage 10.

Avalonia (Newfoundland) as cephalopods (*Hildenbrand et al., 2021*) has been rejected (*Pohle et al., 2022*; *Landing et al., 2023*). It is difficult to reconstruct migration pathways based on the scant data, but cephalopods may have originated in North China, based on their widespread occurrence and highest diversity within that region (*Chen & Teichert, 1983*; *Fang et al., 2019*). From there, they could disperse to neighbouring regions, shown by their common occurrence in South China and their abundance in Queensland (*i.e.*, northeastern Gondwana), while being rather rare on distant palaeocontinents such as Laurentia and Siberia. While Cambrian cephalopods are often associated with stromatolitic environments in very shallow marine deposits (*Chen & Teichert, 1983*; *Landing & Kröger, 2009*; *Dzik, 2020*), they must have been able to spread across oceans to other palaeocontinents relatively quickly.

Since cephalopods likely descended from bottom-dwelling monoplacophorans (*e.g.*, *Kröger, Vinther & Fuchs, 2011*), it may be asked whether the plectronoceratids and other early cephalopods were sufficiently buoyant to swim. The basic prerequisite for buoyancy regulation is a phragmocone consisting of septa and siphuncle. Thus, the usual assumption is that this mechanism was already present in the earliest members of the clade (*e.g.*, *Crick, 1988*; *Wade, 1988*; *Mutvei, Zhang & Dunca, 2007*), although some authors have argued that they were bottom dwellers (*e.g.*, *Yochelson, Flower & Webers, 1973*). Based on

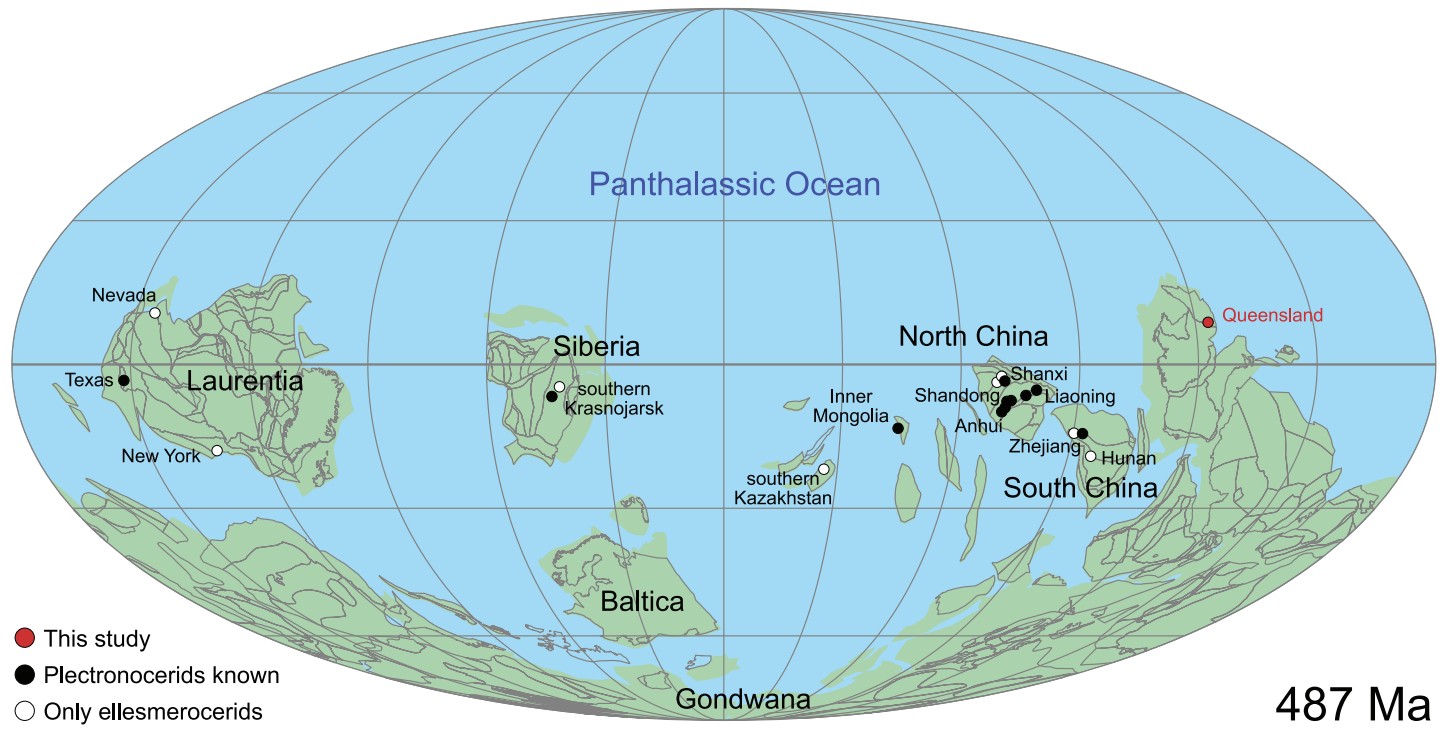

**Figure 14 Global palaeobiogeographic distribution of plectronoceratids and other Cambrian cephalopods during the Furongian (Jiangshanian and Stage 10).** Palaeogeographic reconstruction produced with GPlates (*Müller et al., 2018*), using data from *Merdith et al. (2021)*. Occurrence data on Cambrian cephalopods comes from the literature (see text for references).

hydrostatic models of Palaeozoic cephalopods, *Peterman, Barton & Yacobucci (2019)* concluded that, in order to attain neutral buoyancy, the body chamber would have to occupy at most 35% of the total conch length. Unfortunately, body chambers of plectronoceratids are rare and completeness is commonly difficult to assess. Nevertheless, the few available body chambers reported here (Figs. 6A, 6G), in *Chen et al. (1979a, 1979b)*, and *Chen & Teichert (1983)* barely exceed the corresponding conch diameter, which together with the moderate expansion rate would result in a relatively short body chamber. Using data from *Sinoeremoceras marywadeae* sp. nov. with a maximum chamber height of 22.8 mm (QMF 39527) and a mean expansion rate of 17.0° (Table 4), the total conch length would be around 76.3 mm (if approximated as a strictly conical conch). To attain neutral buoyancy, such a conch would require a body chamber below 26.7 mm, *i.e.*, slightly longer than high. *Peterman, Barton & Yacobucci (2019)* based their estimate of 40% body chamber length on a single specimen of *Plectronoceras cambria* (*Walcott, 1905*) from *Furnish & Glenister (1964*, fig. 81.1b), originally drawn in *Ulrich et al. (1944*, pl. 68, fig. 8). The accuracy of the drawing is unclear but nevertheless, it appears that body chamber ratios that would allow for neutral buoyancy are likely. The relatively long, unwieldy conchs would also make a purely crawling mode of life unlikely.

As the oldest known cephalopods, plectronoceratids are fundamental to better understanding of the origin and early evolution of the clade (*Kobayashi, 1935*; *Miller, 1943*; *Flower, 1954*; *Yochelson, Flower & Webers, 1973*; *Dzik, 1981*; *Wade, 1988*; *Webers &*

**Table 4 Distribution of conch parameters in *Sinoeremoceras marywadeae* sp. nov.**

| | n | Mean | Median | Min | Max | 25% qt | 75% qt | sd |
|---|---|---|---|---|---|---|---|---|
| CWI | 178 | 0.78 | 0.77 | 0.69 | 0.91 | 0.75 | 0.80 | 0.04 |
| CWI* | 74 | 0.78 | 0.77 | 0.69 | 0.87 | 0.75 | 0.80 | 0.04 |
| RCL | 187 | 0.05 | 0.05 | 0.02 | 0.09 | 0.04 | 0.06 | 0.02 |
| RCL* | 72 | 0.05 | 0.05 | 0.02 | 0.09 | 0.04 | 0.07 | 0.02 |
| RSD | 173 | 0.20 | 0.19 | 0.12 | 0.28 | 0.18 | 0.21 | 0.03 |
| RSD* | 73 | 0.20 | 0.20 | 0.13 | 0.28 | 0.18 | 0.22 | 0.03 |
| $ER_h$ | 130 | 17.0 | 17.4 | 4.7 | 29.9 | 12.7 | 20.2 | 5.5 |
| $ER_h^*$ | 64 | 19.4 | 19.9 | 8.0 | 26.4 | 16.8 | 22.5 | 5.1 |
| $ER_w$ | 98 | 13.8 | 13.4 | 3.2 | 27.7 | 9.7 | 17.4 | 5.8 |
| $ER_w^*$ | 61 | 15.7 | 15.0 | 3.5 | 27.7 | 12.1 | 18.6 | 5.5 |

Note:
Measurements were taken from multiple ontogenetic points per specimen, rows with asterisk (*) denote measurements and calculations for a single ontogenetic point per specimen, *i.e.*, the adapicalmost available chamber.

*Yochelson, 1989*; *Mutvei, Zhang & Dunca, 2007*; *Kröger, Vinther & Fuchs, 2011*; *Pohle et al., 2022*). Phylogenetic analysis recovered the Plectronoceratida as sister group to all other cephalopods (*Pohle et al., 2022*), though the question remains whether the very first cephalopod was an ellesmeroceratid or a plectronoceratid (see also *Mutvei, Zhang & Dunca, 2007*; *Dzik, 2020*; *Mutvei, 2020*). Stratigraphically, plectronoceratids appear before ellesmeroceratids but the complexity of the plectronoceratid siphuncle has been taken as an argument that the simpler siphuncles of the ellesmeroceratids represent the ancestral state (*Mutvei, Zhang & Dunca, 2007*; *Mutvei, 2020*). The interpretation that the complexity of the plectronoceratid siphuncle results from the connecting rings being flexible (*Flower, 1964*; *Dzik, 2020*) is refuted here, as the three-dimensional structure of the siphuncles from Black Mountain is very regular (Fig. 7). We favour the hypothesis with the plectronoceratid siphuncle as the ancestral state (*Wade, 1988*; *Pohle et al., 2022*), because complexity does not necessarily predict ancestor-descendant relationships. Instead, the unique bilateral symmetry of the plectronoceratid siphuncle may be a remnant of an originally paired structure, inherited from the cephalopod ancestors (*Wade, 1988*). Thus, a better understanding of the plectronoceratid siphuncle could potentially reveal homology of the siphuncle with another molluscan organ. This has the potential to solve the long-debated but still unresolved question of the origin of the siphuncle (*Yochelson, Flower & Webers, 1973*; *Jell, 1976*; *Dzik, 1981*; *Chen & Teichert, 1983*; *Wade, 1988*; *Webers & Yochelson, 1989*; *Webers, Yochelson & Kase, 1991*; *Mutvei, 2020*).

Although the siphuncle is the single most important cephalopod autapomorphy to distinguish cephalopods from monoplacophorans in the Cambrian (*Yochelson, Flower & Webers, 1973*), crown cephalopods are characterised by several unique traits among molluscs, *e.g.*, arms, hyponome, jaws, and eyes (*Sasaki, Shigeno & Tanabe, 2010*). These synapomorphies must have evolved at some point between the divergence of cephalopods from other molluscs and the origin of the crown group (*e.g.*, *Kröger, Vinther & Fuchs, 2011*; *Vinther, 2015*; *Tanner et al., 2017*). Unfortunately, the timing of the origin of the crown group is still unresolved, as it depends on the position of the ancestral group of the

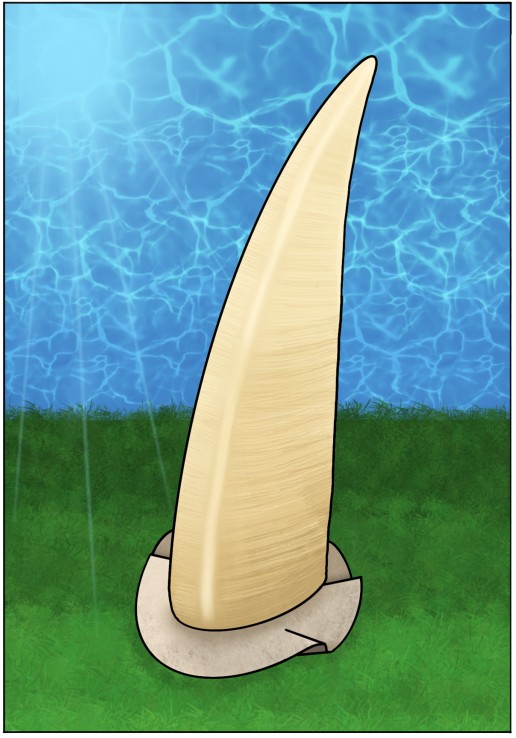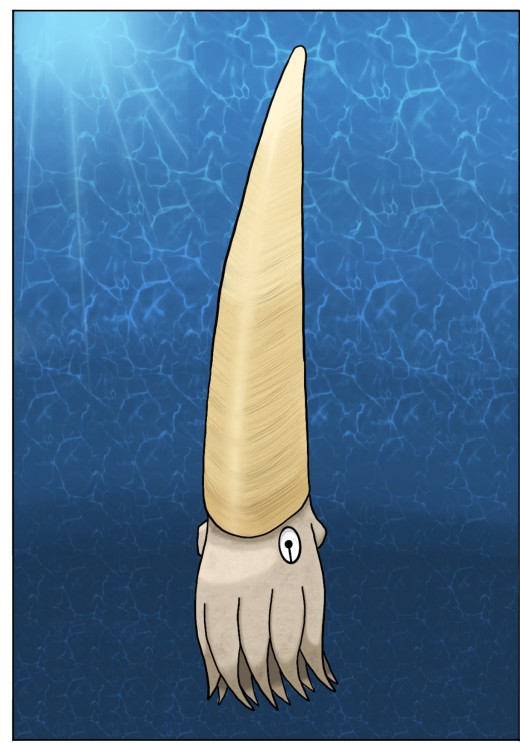

**Figure 15 Possible life reconstructions of *Sinoeremoceras marywadeae* sp. nov.** Credit: Evelyn Friesenbichler. Note that these reconstructions correspond to two extremes to show the possible range of interpretation based on phylogenetic bracketing. The left reconstruction is inspired by a limpet-like soft part anatomy as in living monoplacophorans (*e.g.*, *Wingstrand, 1985*; *Ruthensteiner, Schröpel & Haszprunar, 2010*), without typical cephalopod autapomorphies such as eyes, arms and hyponome. The right reconstruction represents soft part anatomy that would be expected close to the cephalopod crown group (*e.g.*, *Kröger, Vinther & Fuchs, 2011*; *Klug et al., 2015*).

Nautilida (*Pohle et al., 2022*). If its origin is from the Multiceratoidea (*King & Evans, 2019*; *Pohle et al., 2022*), the stem group would consist of only plectronoceratids and certain ellesmeroceratids, making an early origin of these synapomoprhies likely. Conversely, if the Nautilida arose from the Orthoceratoidea (*Kröger, Vinther & Fuchs, 2011*; *Pohle et al., 2022*), all pre-Devonian cephalopods would belong to the stem group, providing a much larger potential time window for the origin of these synapomorphies. This has consequences on how primitive cephalopods can be reconstructed, as despite the common reconstruction of *Plectronoceras* and other early cephalopods with a very typical cephalopod soft body (*e.g.*, *Yochelson, Flower & Webers, 1973*; *Holland, 1987*; *Kröger, 2007*; *Kröger, Vinther & Fuchs, 2011*; *Klug et al., 2015*; *Pohle et al., 2022*), anything between this and a more monoplacophoran, "limpet-like" anatomy is possible (Fig. 15). Lastly, this shows that besides being phylogenetically equally distantly related to crown coleoids and nautiloids (*Pohle et al., 2022*), plectronoceratids also differ significantly from *Nautilus* in terms of anatomy, and the common designation of early Palaeozoic cephalopods as "nautiloids" does not reflect the evolutionary dynamics at the dawn of the cephalopod clade. The term implies a close evolutionary relationship of *Nautilus* with the earliest

cephalopods, even though it is phylogenetically more closely related to ammonoids or even *Octopus* than it is to *Plectronoceras* (*Kröger, Vinther & Fuchs, 2011*). We therefore suggest to replace "nautiloid" whenever possible with alternatives such as early or stem group cephalopods, or directly refer to some of the major clades identified in early Palaeozoic cephalopods (*Pohle et al., 2022*).

## Systematic palaeontology

Synonymy lists are organised in three separate parts: 1) a list of all synonyms referring to their original description, 2) non-original uses of synonyms and 3) erroneous referral to the taxon. Synonymy lists of taxa above species-level consist only of the first part.

Phylum **MOLLUSCA** *Linnaeus, 1758*
Class **CEPHALOPODA** *Cuvier, 1797*
**STEM CEPHALOPODA**
Order **PLECTRONOCERATIDA** *Flower, 1964*

Plectronoceratina *Flower, 1964*: 28.
Protactinocerida Chen & Qi *in Chen et al., 1979a*: 11.
Plectronocerida [nom. transl.]—*Chen et al., 1979a*: 6.
Plectronoceratida [nom. corr.]—*King & Evans, 2019*: 76.
Protactinoceratida [nom. corr.]—*King & Evans, 2019*: 76.

**Included family**
Plectronoceratidae *Kobayashi, 1935*.

**Emended diagnosis**
Relatively small (up to 30 mm in diameter) orthoconic to slightly cyrtoconic conchs, with extremely short chambers and marginal siphuncle on ventral side. If cyrtoconic, the curvature is endogastric. Orthoconic forms may grade into exogastric curvature, but the difference is very subtle. Siphuncle bilaterally symmetrical, expanded ventrally and laterally, mostly with cyrtochoanitic septal necks, while the dorsal side of the neck is tongue-like elongated, forming a holochoanitic septal neck (septal flap). Siphuncular segments form two bulges dorsally that reach behind the septal flap in late ontogenetic stages, without touching each other. Complex diaphragms usually present adapically.

**Remarks**
We consider the Protactinoceratida Chen & Qi *in Chen et al., 1979a* a junior synonym of the Plectronoceratida, because both show a strong bilateral symmetry in their siphuncular segments and septal necks. The segments are expanded laterally and straight dorsally, while representatives of the Ellesmeroceratida can be distinguished by their more or less radially symmetrical siphuncle with straight or concave segments and the usually slightly longer chambers. Many previously established genera have been based on longitudinal sections, which do not show the three-dimensional outline of the siphuncle. For example, a total of seven genera have been established within the Protactinoceratida.
The three-dimensional siphuncular structure seen in the Queensland specimens shows

that these genera can be synonymized, as they are based on sections that are not in the median plane. The different sections, classified as "genera", can be reproduced by sectioning specimens in various planes that differ to some extent from the median plane. Ontogenetic changes from cyrtochoanitic to ortho- to hemi- and holochoanitic septal necks and strongly expanded segments with siphuncular bulbs indicate skewed longitudinal sections. As these characters occur in both orders, it is impossible to assign taxa to one or the other order (see also *Wade & Stait, 1998*; *Mutvei, Zhang & Dunca, 2007*; *Mutvei, 2020*). The detailed three-dimensional structure of the siphuncle is not known in many taxa; thus, some variation from the three-dimensional pattern described here may exist.

Our revised taxonomy presents a significant reduction in the number of taxa. This opens the question whether ranking the Plectronoceratida as an order carries any benefit, or whether its only family should be placed within the Ellesmeroceratida, especially as *Palaeoceras* appears to be transitional between the two groups. As this is mostly a matter of preference, we keep the current classification (see also *King & Evans, 2019*; *Pohle et al., 2022*) but call for further discussions on this topic. Note that we follow the proposal to return to the classical "-ceratida" endings for the revised edition of the *Treatise on Invertebrate Palaeontology, Part K*, but maintain the "-ceratoidea" endings for subclasses, because this matter is more complicated and requires further discussion to achieve congruence across major groups of cephalopods (*King & Evans, 2019*; see also *Hoffmann et al., 2022*).

**Geographic and stratigraphic occurrences**
Anhui, Inner Mongolia, Liaoning, Shandong and Shanxi Provinces of North China, Zhejiang Province of South China, Texas, North America, Siberia, Russia and Queensland, Australia; Jiangshanian to Stage 10, Furongian, late Cambrian.

Family **PLECTRONOCERATIDAE** *Kobayashi, 1935*

Plectronoceratidae *Kobayashi, 1935*: 20.
Balkoceratidae *Flower, 1964*: 33.
Protactinoceratidae Chen & Qi *in Chen et al., 1979a*: 11.

**Type genus**
*Plectronoceras Ulrich & Foerste, 1933*.

**Included genera**
*Plectronoceras Ulrich & Foerste, 1933*; *Palaeoceras Flower, 1954*; *Sinoeremoceras Kobayashi, 1933*.

**Emended diagnosis**
As for the order.

**Remarks**
As outlined above, the supposed differences between protactinoceratids and plectronoceratids result mainly from the orientation of the plane of section relative to the

growth axis. It is therefore impossible to distinguish between the two taxa on familial level. In addition to the Plectronoceratidae, the order was also proposed to contain the Balkoceratidae *Flower (1964)*, which was distinguished by a slight exogastric curvature. However, the presumed balkoceratid *Theskeloceras Chen & Teichert, 1983*, is likely not exogastric and rather a synonym of *Sinoeremoceras* of the Plectronoceratidae. *Teichert (1967)* was initially reluctant to accept the family, but later changed his opinion (*Chen & Teichert, 1983*). In contrast to the Australian and Chinese plectronoceratids, the type specimens of *Balkoceras* appear to be slightly exogastric, which led to its separation into the Balkoceratidae *Flower (1964)*. However, as we regard *Balkoceras* as a synonym of *Palaeoceras*, we consider the Balkoceratidae as a subjective junior synonym of the Plectronoceratidae as well.

Genus **Sinoeremoceras** *Kobayashi, 1933*

*Sinoeremoceras Kobayashi, 1933*: 272.
*Wanwanoceras Kobayashi, 1933*: 271.
*Multicameroceras Kobayashi, 1933*: 273.
*Paraplectronoceras* Chen, Qi & Chen *in Chen et al., 1979a*: 7.
*Jiagouceras* Chen & Zou *in Chen et al., 1979a*: 8.
*Lunanoceras* Chen & Qi *in Chen et al., 1979a*: 9.
*Eodiaphragmoceras* Chen & Qi *in Chen et al., 1979a*: 10.
*? Rectseptoceras* Zou & Chen *in Chen et al., 1979a*: 11.
*Protactinoceras* Chen & Qi *in Chen et al., 1979a*: 12.
*Physalactinoceras* Chen & Qi *in Chen et al., 1979a*: 13.
*Theskeloceras Chen & Teichert, 1983*: 52.
*? Hunyuanoceras Chen & Teichert, 1983*: 59.
*Benxioceras Chen & Teichert, 1983*: 89.
*Mastoceras Chen & Teichert, 1983*: 92.
*Parapalaeoceras Li, 1984*: 229.

**Type species**
*Eremoceras wanwanense Kobayashi, 1931*

**Included species**
*Sinoeremoceras wanwanense* (*Kobayashi, 1931*); *Sinoeremoceras bullatum* (Chen & Qi *in Chen et al., 1979a*) comb. nov.; *Sinoeremoceras endogastrum* (*Li, 1984*) comb. nov.; *Sinoeremoceras inflatum* (Chen & Zou *in Chen et al., 1979a*) comb. nov.; *Sinoeremoceras magicum* Chen *in Lu, Zhou & Zhou, 1984*; *Sinoeremoceras (?) shanxiense* (*Chen & Teichert, 1983*) comb. nov. *Sinoeremoceras sibiriense* (*Balashov, 1959*) comb. nov., *Sinoeremoceras sinense* (Chen & Qi *in Chen et al., 1979a*) comb. nov.; *Sinoeremoceras marywadeae* sp. nov.

**Emended diagnosis**
Endogastrically curved conch with moderate to high expansion rate in early stages, decreasing during ontogeny. Cross-section compressed, dorsum or venter may be more

narrowly rounded. Shell surface poorly known, but apparently smooth or with fine, directly transverse lirae. Cameral length very short, usually <1 mm; RCL decreasing during ontogeny from about 0.15 to 0.05 or less. Siphuncle on ventral side of the conch, laterally expanding between chambers, but straight mid-dorsally, where the otherwise cyrtochoanitic septal necks are elongated and form a triangular, adapically directed septal flap that overlaps the adapically adjacent septal neck. Siphuncular bulges dorsal to septal flap, not joining middorsally. Siphuncular diaphragms present in early stages, with same spacing as septa, with base extending from the septal foramen and adapically convex, parallel to septum, except for central ridge that extends more horizontally towards the shell wall.

**Remarks**
*Kobayashi (1933)* erected *Sinoeremoceras*, *Wanwanoceras* and *Multicameroceras* in the same publication, and despite *Wanwanoceras* having page priority, we, as first revisers under ICZN Article 24(a), give preference to *Sinoeremoceras*, because the type shows the typical siphuncular outline, and *Multicameroceras* was already considered as a synonym by *Chen & Teichert (1983)*. In addition to the "protactinoceratids", most genera previously assigned to the Plectronoceratidae are synonymized with *Sinoeremoceras*. *Plectronoceras Ulrich & Foerste, 1933* differs from *Sinoeremoceras* mainly by its small adult size. *Palaeoceras Flower, 1954* has a lower expansion rate and is essentially straight or only very slightly cyrtoconic; it also apparently lacks cyrtochoanitic septal necks.

*Multicameroceras Kobayashi, 1933* (type species: *Ellesmeroceras*? *multicameratum Kobayashi, 1931*, original designation) was established alongside *Sinoeremoceras Kobayashi, 1933*, citing a "cylindrical to conical elongate" conch compared to a "somewhat fusiform" conch as the major difference between the two (*Kobayashi, 1933*, p. 273). However, besides the very challenging task of drawing an objective boundary between these two states, we find specimens, which may be considered either one of these two and any transitional forms in between, among the Australian material and we attribute these differences to intraspecific or ontogenetic variation. We thus agree with *Chen & Teichert (1983)*, who synonymised the two genera.

*Wanwanoceras Kobayashi, 1933* (type species: *W. peculiare Kobayashi, 1933*, original designation) was provided without differential diagnosis. *Kobayashi (1933)* regarded it as distinct from *Sinoeremoceras* and *Multicameroceras* based on the shape of the septal necks, which seem to be more strongly curved (cyrtochoanitic) in early ontogenetic stages. *Chen & Teichert (1983*, p. 89) noted that *Sinoeremoceras* differed in that "the septal necks gradually are shortened adorally", but this contradicts their own diagnosis of *Sinoeremoceras*, where they stated that the septal necks are "much longer in adoral part of siphuncle" (*Chen & Teichert, 1983*, p. 86). There is no difference between *Wanwanoceras* and *Sinoeremoceras* even if those definitions are followed. As in *Sinoeremoceras* and *Multicameroceras*, the apparent ontogenetic changes in *Wanwanoceras* can be better explained by a slight y-rotation of the plane of section, potentially in combination with a small x-translation.

*Protactinoceras* Chen & Qi *in Chen et al., 1979a* (type species: *P. magnitubulum* Chen & Qi *in Chen et al., 1979a*, original designation), is perhaps the most extreme example of a misaligned section. Although *Mutvei, Zhang & Dunca (2007)* recognised the three-dimensional structure of the plectronoceratid siphuncle and synonymy of the Protactinoceratida with the Plectronoceratida, they did not mention *Protactinoceras* specifically, and in fact, *Mutvei (2020)* later regarded it as distinct from plectronoceratids. Sections of *Protactinoceras* may appear very different from sections of *Sinoeremoceras*. However, a first hint on the suboptimal alignment of the plane of section can be seen in the holotype of the type species (*Chen et al., 1979a*, pl. 3, fig. 12-13; text-fig. 10), in which the siphuncle contracts rapidly adaperturally and eventually even disappears. Aperturally, the septal necks are longer than apically and tilted inwards, providing evidence of the dorsal septal flap. A very similar outline can be produced in *S. marywadeae* sp. nov. by polishing a specimen directly from the venter (Fig. 7L). Thus, *Protactinoceras* is a section that very strongly deviates from the median plane by being rotated nearly 90° along the z-axis, probably with some additional x-translation (in order to go through the siphuncle) and y-rotation (in order to be more or less parallel to the siphuncle). Because of the very strong misalignment, it is almost impossible to compare these specimens with species of *Sinoeremoceras*, although they fit within the concept of regional species employed here.

*Paraplectronoceras* Chen, Qi & Chen *in Chen et al., 1979a* (type species: *P. pyriforme* Chen, Qi & Chen *in Chen et al., 1979a*, original designation) was originally described without differential diagnosis, but *Chen & Teichert (1983*, p. 41) mentioned that it differs from *Plectronoceras* by its slightly larger size and an ontogenetic change from orthochoanitic towards cyrtochoanitic septal necks, while the segments are less expanded. As before, this "ontogenetic" change is interpreted as indication for a septal flap. Thus, *Paraplectronoceras* represents a slight y-rotation of the plane of section, possibly with additional x-translation or z-rotation so that the segments appear less strongly expanded.

*Jiagouceras* Chen & Zou *in Chen et al., 1979a* (type species: *J. cordatum* Chen & Zou *in Chen et al., 1979a*, original designation) shows obvious signs of a misaligned section in the very flat appearance of the septa. While flat septa cannot be excluded *per se*, in the absence of any indication of alignment of the plane of section, this strong deviation from the usually concave septa is better interpreted as a misaligned section, especially as the siphuncle appears removed from the ventral wall by some distance, which is unknown in any cross-section of a Cambrian cephalopod. The suborthochoanitic septal necks and expanded siphuncular segments prevent its interpretation as an ellesmeroceratid. By contrast, *Jiagouceras* fits perfectly in the interpretation of a *Sinoeremoceras* with a plane of section moderately rotated in the z-axis (though not as strong as in "*Protactinoceras*"), probably combined with a slight x-translation, which would explain the less strongly expanded siphuncle.

*Lunanoceras* Chen & Qi *in Chen et al., 1979a* (type species: *L. precordium* Chen & Qi *in Chen et al., 1979a*, original designation) was considered very similar to *Eodiaphragmoceras* except for the shape of its septal necks and diaphragms. Five species have been ascribed to the genus, namely the type, *L. changshanense* Chen & Qi *in Chen et al., 1979a*; *L. compressum* Chen & Qi *in Chen et al., 1979a*; *L. densum Chen & Qi, 1982* and

*L. longatum Chen & Qi, 1982* (also erroneously spelled *L. elongatum* in the same publication). *Chen & Teichert (1983)* considered *Lunanoceras* to be very similar to *Sinoeremoceras* (but classified the two genera in different families) though differing mainly in the shape of the diaphragms. We do not consider diaphragm shape (at the current state of knowledge) to be a generic discriminator and assign *Lunanoceras* as another synonym of *Sinoeremoceras*.

*Eodiaphragmoceras* Chen & Qi *in Chen et al., 1979a* (type species: *E. sinense* Chen & Qi *in Chen et al., 1979a*, original designation) is a monospecific genus with a laterally expanded siphuncle and partly holochoanitic and cyrtochoanitic septal necks on the dorsal side, strongly suggesting a septal flap as in *Sinoeremoceras*. The shape of the diaphragms and the sometimes slightly S-shaped septal necks are difficult to explain by the plane of section alone, but we think that these differences do not justify generic separation, because variation in the shape of the septal flap and the diaphragms is insufficiently known. We consider *Eodiaphragmoceras* a synonym of *Sinoeremoceras* with a relatively centrally aligned plane of section but regard *Sinoeremoceras sinense* (Chen & Qi *in Chen et al., 1979a*) comb. nov. as a valid species.

*Physalactinoceras* Chen & Qi *in Chen et al., 1979a* (type species: *P. bullatum* Chen & Qi *in Chen et al., 1979a*, original designation) is the genus with the highest number of originally attributed species and was differentiated from *Sinoeremoceras* mainly by the more strongly expanded and peculiarly shaped siphuncular segments. The species are separated on the various extents of ontogenetic changes in the septal necks (*e.g.*, shortening in *P. changshanense* Chen & Qi *in Chen et al., 1979a* or lengthening in *P. planoconvexum Chen & Teichert, 1983*) or segment shape (*e.g.*, circular in *P. globosum* Chen & Qi *in Chen et al., 1979a*; oval in *P. bullatum* Chen & Qi *in Chen et al., 1979a*; pear-shaped in *P. breviconicum* Chen & Qi *in Chen et al., 1979a*; balloon-shaped in *P. papilla Chen & Teichert, 1983* or narrowly expanded in *P. confusum Chen & Teichert, 1983*). The frequent changes in ontogeny and seemingly large differences can be explained by planes of section with various slight deviations from the median plane, but in all species the plane of section is aligned through the septal flap, at least to some extent. The different shapes of the segments are likely caused by different positions of the section relative to the ends of the bulges, which end relatively abruptly dorsal to the septal flap. *Physalactinoceras* has a plane of section with slight x-translation (so that the dorsal ends of the siphuncular bulges are visible) and/or slight y-rotation (for ontogenetic changes) with the possibility of slight z-rotation as well.

*Rectseptoceras* Zou & Chen *in Chen et al., 1979a* (type species: *Rectseptoceras eccentricum* Zou & Chen *in Chen et al., 1979a*, original designation) is perhaps not even a plectronoceratid, but instead an ellesmeroceratid, as the septal necks appear to be only slightly curved. However, the connection ring is not preserved (thus making it unclear whether the segments are really expanded or not) and the plane of section is clearly misaligned as indicated by the flat apical septa, the ontogenetic change in siphuncle size (including its apical "disappearance") and position and the erratic expansion rate that seemingly decreases during the earliest stages (which is likely caused by conch curvature).

Because of its short septal spacing and slightly suborthochoanitic septal necks, we only tentatively place this genus in synonymy with *Sinoeremoceras*.

*Benxioceras Chen & Teichert, 1983* (type species: *B. rapidum Chen & Teichert, 1983*, original designation) was another supposedly monospecific genus, differentiated from *Physalactinoceras* by the less expanded segments that do not protrude into the chambers on the dorsal side. However, cross-sections of *B. rapidum* (*Chen & Teichert, 1983*, pl. 17, fig. 8, pl. 19, fig. 5, text-fig. 23, 24) show the same outline of the siphuncle as specimens attributed to *Physalactinoceras* (*e.g.*, *P. qiushugouense Chen & Teichert, 1983*, text-fig. 20) and *Sinoeremoceras marywadeae* sp. nov. (Figs. 7G, 7J), demonstrating the septal flap. Thus, *Benxioceras* represents a section of *Sinoeremoceras* that is close to the median plane, although probably with a slight rotation of the z-axis, which would explain the short distance of the septal necks from the ventral wall (while being contiguous in cross-section), the relatively small siphuncle and rapid expansion rate. In terms of cameral length, siphuncle size and conch compression the species does not differ markedly from other species and cannot be separated from *Sinoeremoceras*. We therefore consider *B. rapidum* as a synonym of *S. wanwanense*.

*Mastoceras Chen & Teichert, 1983* (type species: *M. qiushugouense Chen & Teichert, 1983*, original designation) demonstrates some of the difficulties in applying the original definitions to differentiating between Plectronoceratida and Protactinoceratida when sections are assumed to be correctly placed in the median plane. The only genus to which *Mastoceras* was compared was the plectronoceratid *Theskeloceras*, from which it was said to differ by the straight, more rapidly expanding conch and shape of the siphuncular segments. The latter was also the reason why *Mastoceras* was placed in the Protactinoceratida. The inferred changes in ontogeny of the dorsal septal necks from holochoanitic to hemichoanitic can again be explained by a slight y-rotation of the plane of section. The holotype of the type species represents one of the rare cases, where longitudinal and cross-section of the same specimen are available, indicating that the section is misaligned with a slight z-rotation. Moreover, this cross-section demonstrates that the siphuncle is expanded laterally with cyrtochoanitic septal necks (*Chen & Teichert, 1983*, pl. 9, fig. 4, text-fig. 25) while the adapertural necks of the longitudinal section of the same specimen are hemichoanitic (*Chen & Teichert, 1983*, pl. 9, fig. 1-2, 7, text-fig. 25), implying a septal flap.

*Theskeloceras Chen & Teichert, 1983* (type species: *T. benxiense Chen & Teichert, 1983*, original designation) was described as exogastric and therefore assigned to the Balkoceratidae *Flower (1964)*, but the only specimen (*Chen & Teichert, 1983*, pl. 4, fig. 1) has a weathered dorsal side, which was interpreted as originally concave by the authors based on the convex ventral outline. However, this outline can also result from sectioning an endogastric shell with y-rotation, the convex outline rather reflecting the rounded cross-section. The other species, *T. subrectum Chen & Teichert, 1983* (pl. 3, fig. 1) is essentially straight, with the ventral side slightly more convex. Having examined the original specimens, we conclude that there is no evidence for a marked exogastric curvature of *Theskeloceras* and the genus is more similar to *Sinoeremoceras* than to *Balkoceras* (or rather *Palaeoceras gracile*) because of the typical "protactinoceratid" outline

of the siphuncle, the more rapid expansion rate and the short cameral length. Slight variations in the ventral and dorsal conch outline (*e.g.*, between straight or slightly convex or concave) may also be a result of the misalignment of the plane of section and are only reliable when it is perfectly aligned (which is usually difficult to tell) or in three-dimensionally preserved specimens. The plane of section appears to exhibit slight y-rotation (due to ontogenetic changes) and possibly slight z-rotation and x-translation.

*Parapalaeoceras Li, 1984* (type species: *P. endogastrum Li, 1984*, original designation) has suborthochoanitic septal necks and expanded siphuncular segments throughout, indicating that the section does not pass through the septal flap, but is positioned close to it. It thus represents a section parallel to the median plane with slight x-translation, though it is possible that it is accompanied by some z-rotation. This genus is the only report of a Cambrian cephalopod with cyrtochoanitic septal necks from Zhejiang, which makes it interesting from a palaeobiogeographical perspective. Although there is no unambiguous evidence of a septal flap in the type species, the very close similarity of these specimens to some smaller species of *Sinoeremoceras* and the overlapping morphospace occupation supports its assignment to this genus.

*Hunyuanoceras Chen & Teichert, 1983* (type species: *H. shanxiense Chen & Teichert, 1983*, original designation) was described as one of the oldest ellesmeroceratids, differing from *Plectronoceras* in its tubular siphuncle and the development of diaphragms. The holotype of the type species is poorly preserved and based on a misaligned section, but according to the drawing by *Chen & Teichert (1983*, text-fig. 13), the specimen contains both cyrtochoanitic and orthochoanitic septal necks, thus providing evidence of the presence of a septal flap. Correspondingly, the species more closely resembles a plectronoceratid than an ellesmeroceratid. The lack of diaphragms in *Plectronoceras* may not be primary and their presence possibly represents the ancestral state in cephalopods (*Flower, 1964*), so they cannot serve as a distinguishing character. As explained further below, the distinction between *Plectronoceras* and *Sinoeremoceras* mainly relies on stratigraphic distribution and body size, and since "*Hunyuanoceras*" corresponds to *Sinoeremoceras* in both regards, we tentatively assign its only species to this genus.

### Geographic and stratigraphic occurrences
Anhui, Inner Mongolia, Liaoning, Shandong and Shanxi provinces of North China; Zhejiang province of South China; Krasnoyarsk, Siberia, Russia; Queensland, Australia; Jiangshanian/Stage 10, Furongian, Cambrian.

### *Sinoeremoceras marywadeae* sp. nov.
urn:lsid:zoobank.org:act:581A8E45-9064-4209-B256-5BCACB199967
Figure 4–9, 15; Tables 4, 5

*Protactinoceras*—*Wade, 1988*: 17.
*Protactinoceras*—*Grégoire, 1988*: 76.
*Protactinoceras*—*Wade & Stait, 1998*: 489; fig. 12.8 A, B, E.
*Sinoeremoceras* sp.—*Pohle et al., 2022*: 6; fig. 2; Supplemental Material.

**Table 5 Linear regression of conch parameters, representing ontogenetic trajectories in *Sinoeremoceras marywadeae* sp. nov.**

|  | df | Intercept | Slope | $R^2$ | σ | *p*-value |
|---|---|---|---|---|---|---|
| cw | 176 | 0.045 | +0.77 | 0.96 | 0.55 | <0.001 |
| cw* | 72 | −0.16 | +0.79 | 0.97 | 0.50 | <0.001 |
| CWI | 176 | 0.77 | −0.0006 | 0.003 | 0.040 | 0.5 |
| CWI* | 72 | 0.76 | +0.0017 | 0.026 | 0.039 | 0.17 |
| cl | 185 | 0.45 | +0.0094 | 0.054 | 0.13 | 0.001 |
| cl* | 70 | 0.37 | +0.016 | 0.17 | 0.13 | <0.001 |
| RCL | 185 | 0.092 | −0.0035 | 0.51 | 0.011 | <0.001 |
| RCL* | 70 | 0.088 | −0.0031 | 0.47 | 0.012 | <0.001 |
| sd | 171 | 0.56 | +0.15 | 0.70 | 0.32 | <0.001 |
| sd* | 71 | 0.50 | +0.15 | 0.74 | 0.33 | <0.001 |
| RSD | 171 | 0.24 | −0.0037 | 0.18 | 0.025 | <0.001 |
| RSD* | 71 | 0.24 | −0.0035 | 0.17 | 0.028 | <0.001 |
| $ER_h$ | 128 | 28.2 | −0.96 | 0.27 | 4.7 | <0.001 |
| $ER_h^*$ | 62 | 26.4 | −0.65 | 0.17 | 4.7 | <0.001 |
| $ER_w$ | 96 | 24.1 | −0.89 | 0.24 | 5.1 | <0.001 |
| $ER_h^*$ | 59 | 23.9 | −0.76 | 0.20 | 5.0 | <0.001 |

**Note:**
Measurements were taken from multiple ontogenetic points per specimen, rows with asterisk (*) denote measurements and calculations for only a single ontogenetic point per specimen, *i.e.*, the adapicalmost available chamber.

## Diagnosis

Slightly curved endogastric conch, tending to become straighter in adult stages but some intraspecific variation of curvature is evident across ontogenetic stages. Cross-section compressed but constant throughout ontogeny, with CWI ≅ 0.8. Early ontogenetic stages with more narrowly rounded venter, with a tendency towards equally rounded venter and dorsum in later stages. Expansion rate variable, higher dorsoventrally than laterally but generally decreasing during ontogeny, $ER_h$ ≅ 23° and $ER_w$ ≅ 20° at a conch height of 5 mm and $ER_h$ ≅ 9° and $ER_w$ ≅ 6° at a conch height of 20 mm. Adult body chamber diameter at least 22 mm. Chambers very closely spaced, between 0.3–0.9 mm with a tendency to increase during ontogeny while RCL decreases from around 0.07 in juvenile stages of 5 mm conch height to 0.02 at 20 mm conch height. No abrupt septal crowding near (assumed) maturity but nearly linear decrease in RCL. Sutures with distinct lateral lobes and ventral and dorsal saddles somewhat variable in shape but correlated to conch curvature, *i.e.*, more strongly curved specimens or earlier ontogenetic stages usually have higher dorsal than ventral saddles, while the saddles of nearly straight or mature specimens are roughly equal in height. Siphuncle consisting of strongly oblique segments but siphuncle cross-section (perpendicular to growth axis) roughly circular. Relative size of the siphuncle variable with an apparent tendency to decrease during ontogeny, RSD approximately 0.15–0.25 (mean RSD = 0.2). Septal necks cyrtochoanitic laterally and ventrally, folding adapically and becoming holochoanitic on the dorsum, forming a triangular septal flap that becomes longer during ontogeny until it slightly overlaps the flap

of the preceding segment. Connecting ring strongly expanded, laterally attached to septal flap and bulging dorsally, directed towards the median plane but without touching each other, with a kidney-shaped cross-section. Diaphragms present, with central dorsoventral ridge that is adapically convex and crosses the siphuncle nearly transversely, while lateral parts of the diaphragm slope ventraperturally.

### Etymology

We name the species for Mary Wade, in honour of her pioneering work in collecting, preparing, describing, and interpreting the cephalopods from Black Mountain, with most of her prior work relating to this species.

### Holotype

QMF 39529

### Paratypes

QMF 13332, QMF 13992, QMF 39533, QMF 39542, QMF 40855–40856, QMF 61277, QMF 61634

### Other material

200 specimens from the type locality and horizon (QMF 13314, QMF 13334, QMF 13337, QMF 13340, QMF 13988–13991, QMF 13993–14003, QMF 14005–14006, QMF 39522–39528, QMF 39530–39532, QMF 39534–39541, QMF 39543–39547, QMF 39549, QMF 40852, QMF 40853, QMF 40857–40859, QMF 40861–40862, QMF 40865, QMF 61276, QMF 61278–61281, QMF 61284, QMF 61288–61294, QMF 61315–61316, QMF 61320, QMF 61327–61340, QMF 61343, QMF 61346–61347, QMF 61349, QMF 61351, QMF 61355, QMF 61358–61360, QMF 61362–61363, QMF 61368–61370, QMF 61372, QMF 61375, QMF 61378–61380, QMF 61382–61384, QMF 61386–61388, QMF 61393–61397, QMF 61468–61519, QMF 61523–61540, QMF 61565–61569, QMF 61622–61626, QMF 61632–61633, QMF 61635–61638).

### Type locality and horizon

Black Mountain, near Boulia, western Queensland, Australia; Assemblage Zone 3 (= *Hirsutodontus appressus* Zone), Unbunmaroo Member (BMT 1), lower Ninmaroo Formation, Stage 10, Furongian, late Cambrian.

### Description

The holotype, QMF 39529 (Figs. 5A–5D, 7A) is a slightly endogastrically curved phragmocone fragment 6.3 mm long with a compressed cross-section, the dorsum slightly narrower than the venter and expanding from a width of 12.4 mm and a height of 15.3 mm (CWI = 0.81) to 13.5 and 17.6 mm (CWI = 0.77), respectively (*i.e.*, $ER_w = 9.7°$ and $ER_h = 20.8°$). Septa are extremely closely spaced, < 1mm throughout the specimen, leading to RCL < 0.05. Sutures are incompletely visible in the holotype, apparently slightly inclined dorsaperturally and consisting of slight lateral lobes and pronounced dorsal and ventral saddles. The siphuncle has a diameter between 3.1 mm at the apical end and 3.8 mm at the apertural end (RSD = 0.2). Adaperturally, the siphuncle is preserved as a partial external

mould, revealing the strongly elongated septal necks on the dorsal side of the siphuncle (septal flap). In the holotype, five consecutive septal flaps are preserved, the adapertural ones are broken but the adapical two septal flaps exceed 1 mm in length, almost twice the lengths of the corresponding chambers, although the siphuncular segments are steeply inclined and thus, the septal flaps only slightly overlap the preceding segment. The septal flaps are very narrow, about 1.5 mm at their widest (*i.e.*, adaperturalmost), but quickly decrease in width to about 0.5 mm adapically. Laterally, the septal foramina indicate cyrtochoanitic septal necks by their rounded outline and the smooth transition to the septal flap indicates that the flap's lateral margins represent essentially 90° tilted cyrtochoanitic septal necks as well.

One of the paratypes, QMF 40856 (Figs. 6H, 6I), had been designated by Mary Wade, in her working notes, as potential holotype. However, the specimen is partially embedded in matrix, making exact measurements of width, height, and siphuncle at different ontogenetic positions difficult. The specimen is 30.9 mm long and has an adapertural diameter of 14.1 and 18.3 mm (CWI = 0.77) with closely spaced chambers at the adapertural end (RCL = 0.03) and a siphuncle diameter of 3.2 mm (RSD = 0.18). The siphuncle is exposed adaperturally as an external mould, but the septal flaps are only partly preserved. At the apical end, the siphuncle is preserved as an internal mould, revealing the laterally and ventrally expanded, strongly oblique siphuncular segments. The shell surface is recrystallized, but it is apparently smooth.

Another paratype, QMF 13332 (Fig. 5I), was deliberately polished by Mary Wade with approximately 90° of z-rotation from the ventral side, exposing a "*Protactinoceras*"-like outline of the siphuncle (Fig. 7L). However, as the polished surface passes towards the dorsal margin of the siphuncle, the septal necks become longer and straight, thus also exposing the septal flap. The adapical surface of the specimen shows that it belongs to the same species as the other types, because the septal flap is visible as a rounded dorsal inflexion of the siphuncle (Fig. 7I). The adapical end of the siphuncle is sealed, and its symmetrical structure likely represents a diaphragm. The diaphragm has a broad surface that slopes ventraperturally and has a more directly transverse, dorsoventral central ridge about the same width as the septal flap. The central ridge does not appear to touch the ventral shell wall, but this may be caused by preservation. Two diaphragms are visible in the polished part of the section, revealing the same structure with a central ridge.

An earlier ontogenetic stage is represented by the paratype QMF 39533 (Figs. 5E–5H). This specimen is more strongly curved endogastrically, has a relatively rapid expansion rate and a more narrowly rounded dorsum. Because transitional forms between this morphology and larger specimens exist, these differences are attributed to ontogeny. The sutures of the specimens are distinct in the dorsal half, having a narrow dorsal saddle and slight lateral lobes. The siphuncle is only exposed adapically, but no details of its structure are visible.

In many specimens, the septal flap is not preserved, leaving only the imprint of the two siphuncular bulges on the dorsal side of the siphuncle, which results in an hourglass shaped elevation mid-dorsally. This structure is best seen in paratype QMF 39542 (Fig. 5J). The bulges themselves, or rather the internal mould of the siphuncle, are visible in

paratype QMF 40855, where the bulges almost touch each other but leave a small gap adaperturally. This gap causes the hourglass-shaped ridge in its negative and represents the position of the septal flap. The latter is hidden behind (ventral to) the bulges. That the specimens exposing imprints of siphuncular bulges are conspecific with those exposing septal flaps is shown by paratype QMF 61277, which combines both preservation types. The adapical part of the specimen preserves the imprints of the bulges, while adaperturally, the (somewhat corroded) septal flaps are evident. The septal flaps are very slightly displaced towards the venter compared to the imprints of the bulges, indicating that the bulges reach behind (dorsal) the flap.

The rest of the material consists of fragments of phragmocones and siphuncles of variable preservation. The siphuncle morphology was already described in detail in the Result section. Conch height 4–23 mm, measured in 77 specimens at 212 individual ontogenetic stages. Conch is compressed regardless of ontogenetic stage, with CWI $\cong$ 0.78. Cameral length is always very short, below 1 mm, and although there is a significant ontogenetic decrease with cl = 0.45 + 0.00094 $^*$ ch, this includes considerable variation, and as $R^2$ = 0.054, ontogeny only explains a negligible part of this variation. There is a stronger relationship with relative septal spacing, as RCL = 0.092–0.0035 $^*$ ch and $R^2$ = 0.51. The size of the siphuncle linearly increases during ontogeny, but its relative size decreases, with RSD = 0.24–0.0037 $^*$ ch, corresponding to RSD $\cong$ 0.22 at a conch height of 5 mm and RSD $\cong$ 0.17 at a conch height of 20 mm. It is possible that this decrease is at least partially caused by taphonomic effects, as larger specimens tend to have more poorly preserved siphuncles due to weathering, which makes exact measurements challenging. The decrease is relatively minor, and the linear regression of RSD results in $R^2$ = 0.18 and $\sigma$ = 0.025, meaning that there is considerable variation in RSD that is not explained by ontogeny. Expansion rate decreases during ontogeny, with $ER_h$ = 28.2°–0.96° $^*$ ch and $ER_w$ = 24.1°–0.89° $^*$ ch, corresponding to $ER_h$ $\cong$ 23.4° and $ER_w$ $\cong$ 19.7° at a conch height of 5 mm and $ER_h$ $\cong$ 9.0° and $ER_w$ $\cong$ 6.3° at a conch height of 20 mm. However, there is considerable variation in expansion rate at all ontogenetic stages, as illustrated by $R^2$ = 0.27 and $\sigma$ = 4.7° for $ER_h$ and $R^2$ = 0.24 and $\sigma$ = 5.1° for $ER_w$.

**Remarks**

Comparison with other species from China and Siberia is hampered by the fact that the latter are almost exclusively known from longitudinal sections, thus the three-dimensional outline of the conch and particularly the siphuncle is unknown in all other species of *Sinoeremoceras*. The diagnoses of many proposed plectronoceratid and "protactinoceratid" species are doubtful because they are based on misaligned sections not aligned with the median section, which potentially has a large effect on diagnostic characters such as expansion rate and relative size of the siphuncle. Importantly, this also applies to the length and shape of the septal necks and the shape of the siphuncular segments because of the peculiar plectronoceratid siphuncle morphology. Another character that has been frequently used to differentiate between species, is the seemingly large ontogenetic change within the siphuncle, which is shown above to be a result of the misalignment of the plane of section. Without new material from the original Chinese

localities, it is impossible to evaluate the number and morphology of the involved species. Therefore, the specimens described here cannot be referred to any other species. The material—and also any future material—can only be referred to other species of *Sinoeremoceras* if new material is collected from the original localities and 3D-reconstructions become available. The only described species of plectronoceratids that are known from the three-dimensional outline of the conch are *S. wanwanense* (*Kobayashi, 1931*); *Multicameroceras multicameratum* (*Kobayashi, 1931*); *M. cylindricum Kobayashi, 1933* and *Wanwanoceras peculiare Kobayashi, 1933*, all of which are here synonymised with the type species. *S. marywadeae* sp. nov. differs from *S. wanwanense* by its comparatively deeper lateral lobes. Measurements of *Physalactinoceras qiushugouense Chen & Teichert, 1983* (NIGP 73801, pl. 14, fig. 1, 3) and *Sinoeremoceras foliosum* Chen & Qi in *Chen et al., 1979a* (NIGP 46128, pl. 1, fig. 3, 4) suggest that they may be distinguished by their slightly higher expansion rate, although this is probably strongly influenced by alignment of the plane of section and thus somewhat questionable. Otherwise, in terms of siphuncle size, cameral length and curvature, the species seem to be close.

**Geographic and stratigraphic occurrences**
Type locality and horizon only.

**Sinoeremoceras wanwanense** (*Kobayashi, 1931*)
Figure 9, 16, 17; Table 6

*Eremoceras wanwanense Kobayashi, 1931*: 164; pl. 16, fig. 4a, b.
*Ellesmeroceras* (?) *multicameratum Kobayashi, 1931*: 163; pl. 16, fig. 7; pl. 19, fig. 2a, b, 3.
*Wanwanoceras peculiare Kobayashi, 1933*: 271; pl. 1, fig. 6, 10; pl. 2, fig. 12; pl. 4, fig. 9.
*Multicameroceras cylindricum Kobayashi, 1933*: 274; pl. 2, fig. 14; pl. 4, fig. 5.
*Theskeloceras benxiense Chen & Teichert, 1983*: 53; pl. 4, fig. 1, 2, 4.
*Theskeloceras subrectum Chen & Teichert, 1983*: 53; pl. 3, fig. 1, 3; pl. 10, fig. 3, 4; pl. 12, fig. 3; text-fig. 12.
*Physalactinoceras benxiense Chen & Teichert, 1983*: 76; pl. 5, fig. 4, 5.
*Physalactinoceras confusum Chen & Teichert, 1983*: 78; pl. 17, fig. 12; pl. 19, fig. 1.
*Physalactinoceras cornutum Chen & Teichert, 1983*: 79; pl. 14, fig. 5.
*Physalactinoceras niuxintaiense Chen & Teichert, 1983*: 81; pl. 1, fig. 8; pl. 19, fig. 11.
*Physalactinoceras papilla Chen & Teichert, 1983*: 81; pl. 1, fig. 2, 7; pl. 7, fig. 5, 8; pl. 17, fig. 7, 11; text-fig. 19.
*Physalactinoceras planoconvexum Chen & Teichert, 1983*: 83; pl. 7, fig. 6, 7; pl. 9, fig. 5; pl. 16, fig. 1, 3.
*Physalactinoceras qiushugouense Chen & Teichert, 1983*: 83; pl. 14, fig. 1, 3; text-fig. 20.
*Physalactinoceras rarum Chen & Teichert, 1983*: 85; pl. 3, fig. 5.
*Physalactinoceras speciosum Chen & Teichert, 1983*: 85; pl. 14, fig. 2; pl. 18, fig. 5.
*Sinoeremoceras magnum Chen & Teichert, 1983*: 87; pl. 12, fig. 1, 2, 5.
*Sinoeremoceras taiziheense Chen & Teichert, 1983*: 87; pl. 2, fig. 1, 4.
*Wanwanoceras peculiare curtum Chen & Teichert, 1983*: 89; pl. 10, fig. 1.

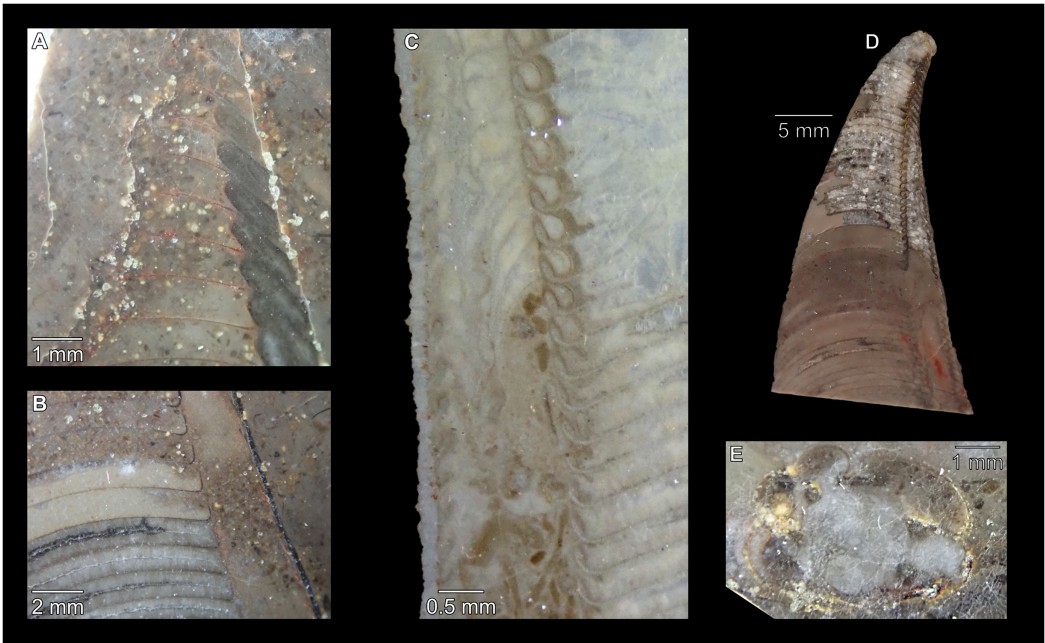

**Figure 16** *Sinoeremoceras wanwanense* (*Kobayashi, 1931*) **from the Wanwankou Member, Fengshan Formation of Liaoning, North China.** (A) NIGP 46148, originally designated as holotype of *Wanwanoceras lunanense* Chen & Qi *in Chen et al., 1979a*. Section through apical phragmocone, revealing the expanded part of the siphuncle and misalignment of the plane of section due to septa between the segments. (B) Same specimen, section through apertural phragmocone, long septal necks indicate septal flap. (C) NIGP 73805, originally designated as holotype of *Physalactinoceras benxiense Chen & Teichert, 1983*. Misaligned longitudinal section of siphuncle. (C) NIGP 73799, originally designated as holotype of *Physalactinoceras speciosum Chen & Teichert, 1983*. Misaligned longitudinal section. (D) NIGP 46123, originally designated as paratype of *Sinoeremoceras foliosum* Chen & Qi *in Chen et al., 1979a*. Cross-section of siphuncle, exposing two successive segments, dorsally folding behind the septal flap, and potentially part of the diaphragm.

**Table 6** Conch parameters and ontogenetic trajectories of *Sinoeremoceras wanwanense* (*Kobayashi, 1931*).

|  | *n* | Mean | Min | Max | Intercept | Slope | *p*-value |
|---|---|---|---|---|---|---|---|
| CWI | 23 | 0.77 | 0.60 | 0.9 | 0.76 | +0.0005 | 0.89 |
| RCL | 31 | 0.053 | 0.023 | 0.13 | 0.11 | −0.0040 | <0.001 |
| RSD | 33 | 0.18 | 0.10 | 0.34 | 0.16 | +0.0022 | 0.25 |
| $ER_h$ | 29 | 24.2° | 6° | 54° | 36.1° | −0.87 | 0.008 |

*Benxioceras rapidum Chen & Teichert, 1983*: 89; pl. 1, fig. 6; pl. 8, fig. 1–3; pl. 17, fig. 2, 8; pl. 19, fig. 5; text-fig. 21–24.

*Mastoceras qiushugouense Chen & Teichert, 1983*: 92; pl. 9, fig. 1–4, 7; pl. 13, fig. 1, 3; text-fig. 25, 26.

*Mastoceras? obliquum Chen & Teichert, 1983*: 94; pl. 2, fig. 2, 3.

*Multicameroceras multicameratum—Kobayashi, 1933*: 274; pl. 2, fig. 8; pl. 3, fig. 1, 3; pl. 4, fig. 1.

*Sinoeremoceras wanwanense—Kobayashi, 1933*: 305; pl. 2, fig. 6, 7, 9, 10; pl. 3, fig. 2, 5.
*Multicameroceras multicameratum—Flower, 1954*: 17; fig. 4A-D.
*Sinoeremoceras wanwanense—Flower, 1954*: 19; fig. 4E-H.
*Multicameroceras multicameratum—Furnish & Glenister, 1964*: K146; fig. 82,1a–c.
*Sinoeremoceras wanwanense—Furnish & Glenister, 1964*: K147; fig. 82,2a, b
*Wanwanoceras peculiare—Furnish & Glenister, 1964*: K147; fig. 82,3.
*Protactinoceras magnitubulatum* [*sic.*]*—Chen et al., 1979a*: 23; pl. 1, fig. 6.
*Protactinoceras magnitubulum—Chen & Teichert, 1983*: 75; pl. 7, fig. 9
*Protactinoceras magnitubulatum* [*sic.*]*—Chen & Teichert, 1983*: 98; pl. 3, fig. 4.
*Physalactinoceras breviconicum—Chen & Teichert, 1983*: 76; pl. 13, fig. 6; pl. 19, fig. 6.
*Physalactinoceras changshanense—Chen & Teichert, 1983*: 77; pl. 12, fig. 6.
*Physalactinoceras globosum—Chen & Teichert, 1983*: 79; pl. 5, fig. 3; pl. 12, fig. 4, 7; pl. 15, fig. 4, 5; pl. 19, fig. 9, 10.
*Sinoeremoceras wanwanense—Kobayashi, 1989*: 372; text-fig. 5.
*Theskeloceras subrectum—Mutvei, Zhang & Dunca, 2007*: 1328; text-fig. 1B.
*Theskeloceras benxiense—Mutvei, Zhang & Dunca, 2007*: 1328; text-fig. 1C, 2B.
*Physalactinoceras speciosum—Mutvei, Zhang & Dunca, 2007*: 1328; text-fig. 4A–D.
*Physalactinoceras globosum—Mutvei, Zhang & Dunca, 2007*: 1328.
*Physalactinoceras* cf. *globosum—Mutvei, Zhang & Dunca, 2007*: 1328; text-fig. 3.
*Sinoeremoceras taiziheense—Mutvei, Zhang & Dunca, 2007*: 1328.
*Physalactinoceras globosum—Mutvei, 2020*: 119; fig. 3B.
*Theskeloceras benxiense—Mutvei, 2020*: 119: fig. 3C, 4, 5.
*Physalactinoceras* cf. *globosum—Mutvei, 2020*: 122; fig. 6A, B.

non *Sinoeremoceras taiziheense—Chen & Teichert, 1983*: 88; pl. 13, fig. 4.

**Emended diagnosis**
Large species with adult conch height of more than 20 mm. Conch distinctly curved endogastrically during early ontogenetic stages, almost orthoconic when approaching maturity, conch cross-section compressed (CWI $\cong$ 0.8). Expansion rate variable (6°–41°), but with tendency to decrease during ontogeny, with $ER_h \cong 29.5°$ at 5 mm conch height and $ER_h \cong 17.5°$ at 20 mm conch height. Septal spacing displays a similar decreasing, but less variable trend during ontogeny with RCL = 0.10 − 0.004 $^*$ ch ($n$ = 32, $p < 0.001$), corresponding to RCL = 0.08 at 5 mm conch height and RCL = 0.02 at 20 mm conch height. The ontogenetic increase in siphuncle size is not statistically significant (RSD = 0.16 + 0.002 $^*$ ch, $n$ = 33), and may be caused by the placement of the plane of section. The size of the siphuncle varies between RSD = 0.11 and RSD = 0.25, with mean RSD = 0.19. Long septal flap, reaching holochoanitic condition mid-dorsally. Siphuncular segments with pronounced dorsal siphuncular bulge, moderately expanded laterally. Diaphragms generally sloping ventraperturally.

**Holotype**
UMUT PM 00115

**Type locality and horizon**

Near Qiushugou Village, Taizihe Valley, Liaoning Province, North China; *Sinoeremoceras* Zone (= upper *Proconodontus muelleri* Zone to lower *Eoconodontus notchpeakensis* Subzone), upper Wanwankou Member, Fengshan Formation, late Furongian, late Cambrian.

**Remarks**

Together with *Multicameroceras multicameratum* (*Kobayashi, 1931*), this was the first description of the typical "protactinoceratid" siphuncle. Although details of the structure of the siphuncle cannot be recognised in the original publication, they are clear in a paratype described by *Kobayashi (1933)*. *Sinoeremoceras* was distinguished from *Multicameroceras* by the general outline of the conch (*Kobayashi, 1933*) but we regard the variation to be intraspecific or ontogenetic, especially considering the similarly wide and continuous variation in the Australian material of *S. marywadeae* sp. nov. The types of *S. wanwanense* and *M. multicameratum* differ in preservation, which may have misled *Kobayashi (1931*, *1933)*. *Chen & Teichert (1983)* regarded the two type species as congeneric. We cannot separate them at species level either and consider *M. multicameroceratum* and *M. cylindricum Kobayashi, 1931* (originally separated on its lower expansion rate) to be synonyms of *S. wanwanense*. Many species established by *Chen et al. (1979a*, *1979b)*, *Chen & Qi (1982)*, *Chen & Teichert (1983)* were based on characters that can be explained by different alignments of the plane of section; since most specimens are only available as thin sections, no indication of their relation to the median plane is available. For example, *S. magnum Chen & Teichert, 1983* was distinguished from *S. multicameratum* solely by its larger siphuncle, but this difference may easily be caused by x-translation and/or z-rotation. Although the observed RSD varies considerably between the holotypes of those two species (0.16 *vs.* 0.25), it falls well within the limits of intraspecific variation seen in *S. marywadeae* sp. nov. and there are other specimens (which were again designated as separate species) with intermediate siphuncle sizes, *e.g.*, *S. taiziheense Chen & Teichert, 1983* (RSD = 0.23); *Physalactinoceras niuxintaiense Chen & Teichert, 1983* (RSD = 0.2); *S. wanwanense* (RSD = 0.19 in holotype) or *P. qiushugouense* (RSD = 0.17). As explained above, we rely mostly on ontogenetic trajectories of conch parameters and adult size under consideration of their stratigraphic and geographic distribution for species assignment. Because intra-population variability is lower than between different populations, we group all species previously described from the Wanwankou Member of the Fengshan Formation of Liaoning under this species, although there is a strong overlap with *S. bullatum* from equivalent horizons of Shandong. Within Liaoning, *Sinoeremoceras* has been documented from three different localities within the Taizihe Valley (Wangouli and Qiushukou, both Niuxintai area and Yingzi, Huolianzhai area) and at Fuzhouwan, Liaodong Peninsula (*Chen & Teichert, 1983*). In summary, we consider 20 previously established species and one subspecies as subjective synonyms of *S. wanwanense* (*Kobayashi, 1931*). Specimens described from the Wanwankou Member of Liaoning reach sizes of more than 20 mm in diameter, which is larger than any of the known specimens from Anhui. We therefore consider adult size as another distinguishing

character between *S. wanwanense* and other species of *Sinoeremoceras*. The diaphragms of *S. wanwanense* are steeply sloping adaperturally towards the venter, which may distinguish it from *S. bullatum* (Chen & Qi *in Chen et al., 1979a*) with regularly concave transverse diaphragms and *S. sinense* (Chen & Qi *in Chen et al., 1979a*) with almost straight diaphragms that slope in the opposite direction. However, the slight ω-shape of the diaphragms in the co-occurring *"Physalactinoceras compressum"* Chen & Qi *in Chen et al., 1979a* (here referred to *Sinoeremoceras bullatum*), and supposed species of *Protactinoceras* Chen & Qi *in Chen et al., 1979a* suggests that the diaphragms may not simply traverse the siphuncle but show a more complex three-dimensional shape, as also shown in the Australian material (Figs. 7H, 7I). There appears to be a central siphuncular ridge that may be more steeply sloping, while the diaphragms are more or less transversely concave laterally. The strong bilateral symmetry would thus also extend to the diaphragms. Consequently, distinguishing species within *Sinoeremoceras* based on longitudinal sections of diaphragms is also questionable. Another likely synonym is *Benxioceras rapidum Chen & Teichert, 1983*, which was distinguished by its large expansion rate and the shape of the septal necks. As shown above, the latter cannot serve as a diagnostic character. The expansion rate is extremely high in the holotype; however, after examination of the specimen we conclude that the expansion rate was probably overestimated, since the dorsum of the specimen appears to be strongly weathered and the plane of section was misaligned. The other specimens fit well into the variation of other Chinese *Sinoeremoceras* species and since all specimens are rather small, it fits into the ontogenetic trajectory of the decreasing apical angle. Unfortunately, no diaphragms were described from *B. rapidum*, and the diaphragms indicated by *Chen & Teichert (1983)* in their text-fig. 21 are difficult to recognise in the original specimen. The conch is more strongly curved than in most other specimens of *S. wanwanense* because it is an early ontogenetic stage, which show higher curvatures in *S. marywadeae* sp. nov. as well. *Mastoceras qiushugouense Chen & Teichert, 1983* is interpreted as another synonym of *S. wanwanense*. It was established based on apparently conical diaphragms; however, they already indicated in their description that the diaphragm itself is not preserved and only indicated by the conical boundary between the calcite spar filling and the matrix. From direct observation, we conclude that this conical shape is diagenetic in origin and cannot be used in taxonomy. The supposed slight exogastric curvature of *M. qiushugouense* is barely recognisable and we consider the conch to be more or less straight, which is also in line with the variation and ontogenetic decrease in curvature seen in other *Sinoeremoceras* representatives. Similar minor differences exist between other synonymised species that can be explained by misalignment of the plane of section. Variation in conch parameters in specimens from the Wanwankou Member of Liaoning shows a continuous variation, thus not allowing the separation of multiple species (Fig. 17).

*Sinoeremoceras wanwanense* differs from *S. inflatum* and *S. (?) shanxiense* in its larger adult size and less strongly curved conch particularly during late ontogeny. Similarly, *S. wanwanense* is larger than *S. magicum* and *S. endogastrum*, but those species are characterised by a very low expansion rate. In comparison with these smaller species, the siphuncular segments appear to be more strongly expanded and the septal flaps longer in

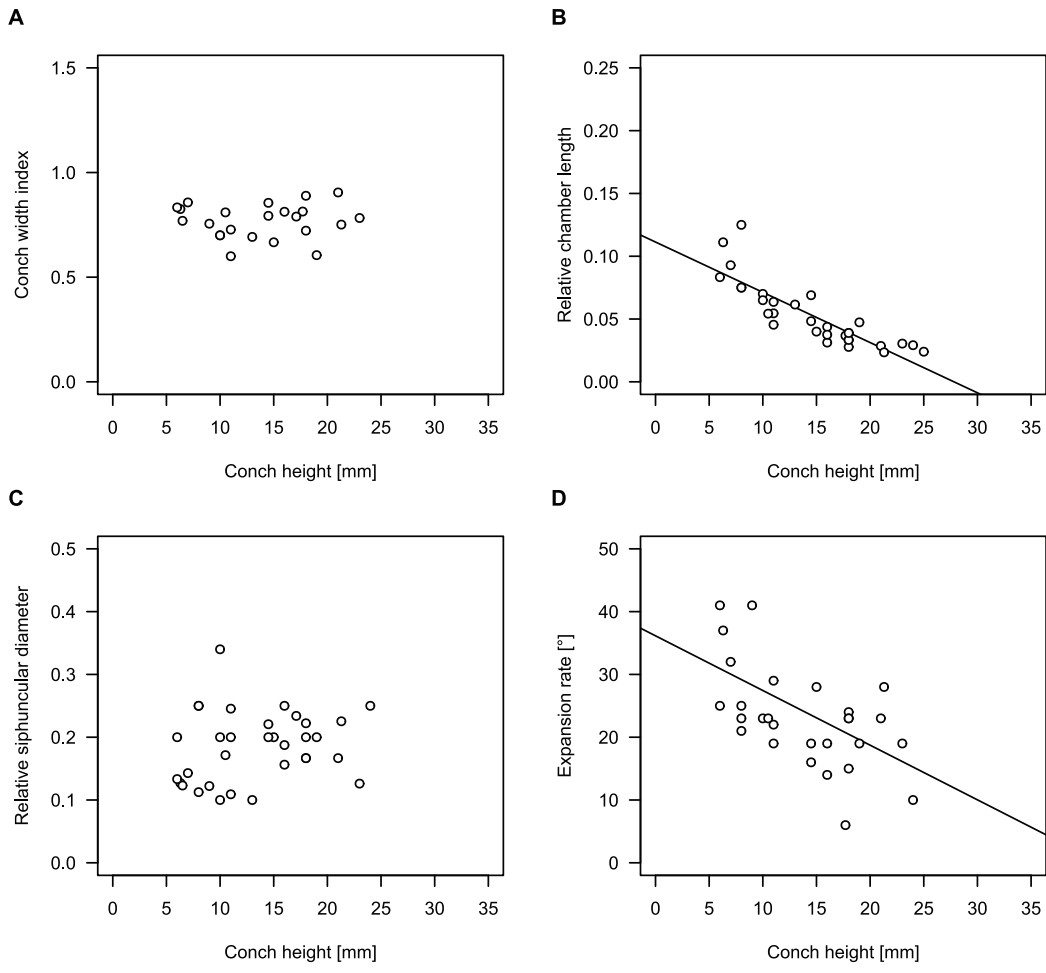

**Figure 17 Conch parameters of *Sinoeremoceras wanwanense* (*Kobayashi, 1931*) throughout ontogeny (represented by conch height).** (A) Conch width index (CWI). (B) Relative cameral length (RCL). (C) Relative siphuncular diameter. (D) Height expansion rate (ER$_h$).

*S. wanwanense*, although the possible misalignment of the plane of section and the ontogenetic trajectory of the siphuncle need further studies. *S. bullatum* is close to *S. wanwanense*, but it differs by its larger conch size. This size difference also applies to the large species *S. sibiriense*, which is distinguished by its nearly orthoconic conch, and *S. sinense* that differs by its peculiarly shaped septal necks. The similarly sized *S. marywadeae* sp. nov. can be distinguished by its deeper lateral lobes and lower expansion rate.

**Geographic and stratigraphic occurrences**
Liaoning Province, North China; close to Jiangshanian/Stage 10 boundary, Furongian, late Cambrian.

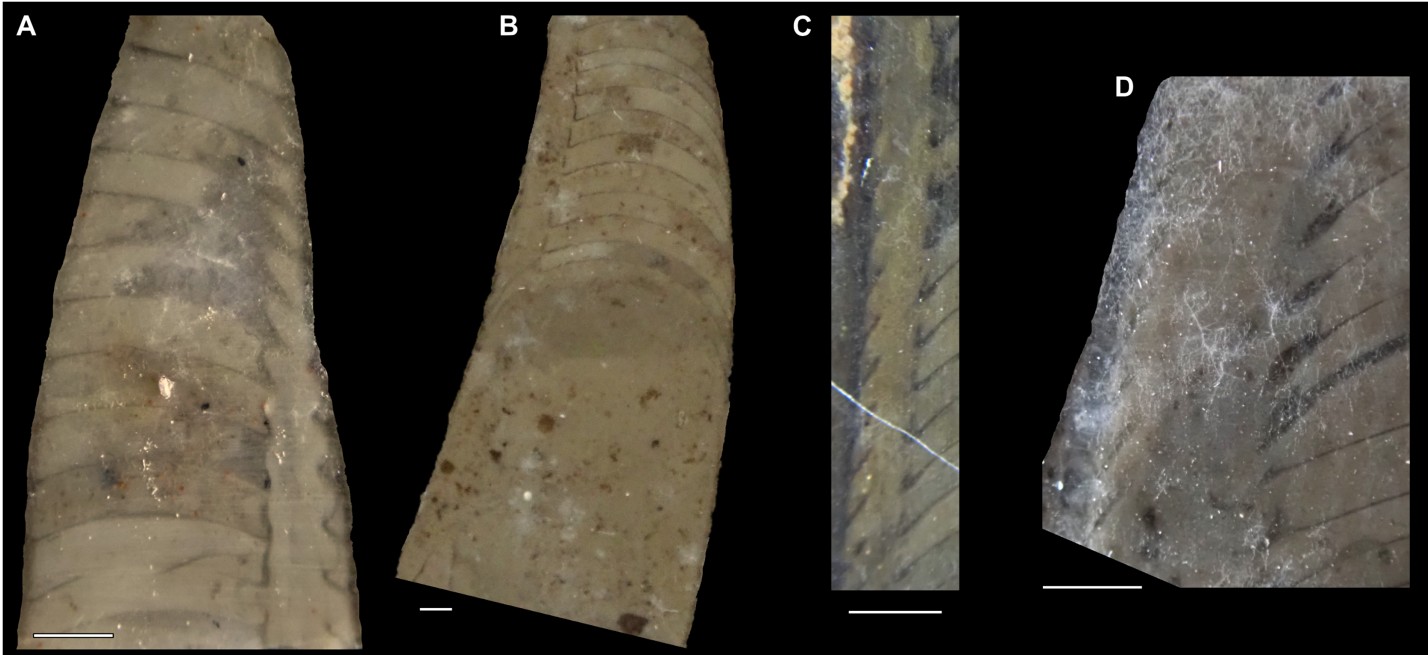

**Figure 18** *Sinoeremoceras inflatum* (Chen & Zou *in Chen et al., 1979a*) from the Jiagou Member, Fengshan Formation of Anhui, North China.
(A) NIGP 46194, holotype, originally attributed to *Paraplectronoceras* Chen, Qi & Chen *in Chen et al., 1979a*. Misaligned longitudinal section through expanded siphuncular segments apically and septal flap adaperturally. (B) NIGP 46741, originally designated as holotype of *Paraplectronoceras suxianense* Chen, Qi & Chen *in Chen et al., 1979a*. Longitudinal, slightly misaligned section through siphuncle, showing apparent change from long orthochoanitic septal necks to cyrtochoanitic septal necks in adapertural direction. (C) NIGP 46201, originally designated as holotype of *Paraplectronoceras impromptum* Chen, Qi & Chen *in Chen et al., 1979a*. More or less central longitudinal section, exposing elongated septal necks that indicate septal flap. (D) NIGP 46193, originally designated as holotype of *Lunanoceras compressum* Chen & Qi *in Chen et al., 1979a*. Misaligned longitudinal section through siphuncle.

**Sinoeremoceras inflatum** (Chen & Zou *in Chen et al., 1979a*) comb. nov.
Figures 18, 19; Table 7

*Paraplectronoceras inflatum* Chen & Zou *in Chen et al., 1979a*: 8; pl. 3, fig. 6.
*Paraplectronoceras pyriforme* Chen, Qi & Chen *in Chen et al., 1979a*: 7; pl. 4, fig. 12.
*Paraplectronoceras suxianense* Chen, Qi & Chen *in Chen et al., 1979a*: 7; pl. 1, fig. 5; pl. 3, fig. 14.
*Paraplectronoceras impromptum* Chen & Zou *in Chen et al., 1979a*: 7; pl. 3, fig. 2
*Jiagouceras cordatum* Chen & Zou *in Chen et al., 1979a*: 8; pl. 4, fig. 17; text-fig. 5.
*Lunanoceras compressum* Chen & Qi *in Chen et al., 1979a*: 10; pl. 3, fig. 3, 4; pl. 4, fig. 4, 5.
? *Rectseptoceras eccentricum* Zou & Chen *in Chen et al., 1979a*: 11; pl. 3, fig. 1; text-fig. 9.
*Sinoeremoceras anhuiense* Zou & Chen *in Chen et al., 1979b*: 118; pl. 3, fig. 5, 6.
*Wanwanoceras multiseptum* Zou & Chen *in Chen et al., 1979b*: 118; pl. 1, fig. 14, 15; pl. 2, fig. 13.
*Paraplectronoceras curvatum* Chen & Qi *in Chen et al., 1980*: 167; pl. 1, fig. 12.
*Paraplectronoceras abruptum* Chen & Qi *in Chen et al., 1980*: 168; pl. 1, fig. 15.
*Paraplectronoceras vescum* Chen & Qi *in Chen et al., 1980*: 168; pl. 1, fig. 20.
*Parapectronoceras parvum* Chen & Qi *in Chen et al., 1980*: 168; pl. 2, fig. 3.
*Sinoeremoceras pisinum* Chen, Zou & Qi *in Chen et al., 1980*: 179; pl. 1, fig. 18.

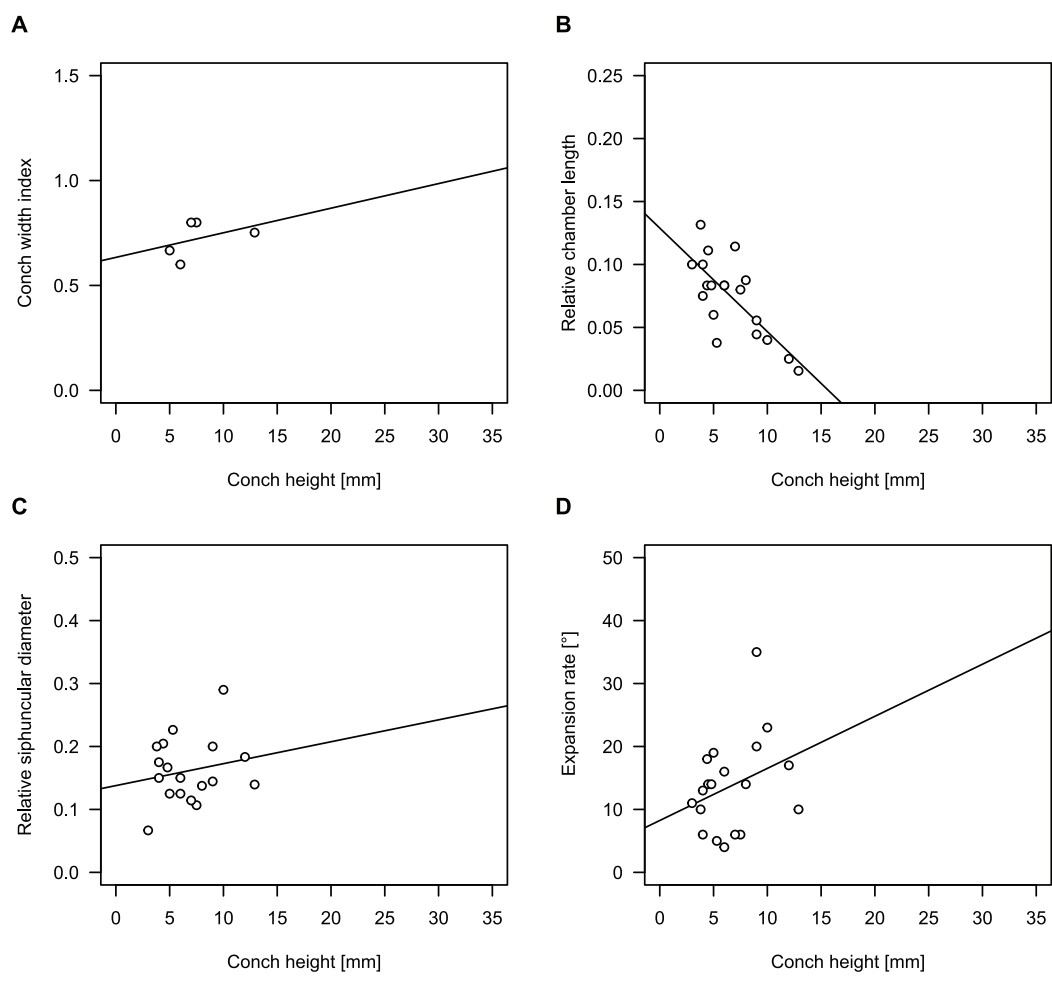

**Figure 19 Conch parameters of *Sinoeremoceras inflatum* (Chen & Zou in *Chen et al., 1979a*) throughout ontogeny (represented by conch height).** (A) Conch width index (CWI). (B) Relative cameral length (RCL). (C) Relative siphuncular diameter. (D) Height expansion rate ($ER_h$).

**Table 7 Conch parameters and ontogenetic trajectories of *Sinoeremoceras inflatum* (Chen & Zou in *Chen et al., 1979a*).**

|        | *n* | Mean  | Min   | Max  | Intercept | Slope    | *p*-value |
|--------|-----|-------|-------|------|-----------|----------|-----------|
| CWI    | 6   | 0.69  | 0.50  | 0.80 | 0.52      | +0.023   | 0.18      |
| RCL    | 20  | 0.08  | 0.016 | 0.24 | 0.15      | −0.011   | 0.002     |
| RSD    | 19  | 0.17  | 0.067 | 0.32 | 0.17      | −0.0001  | 0.98      |
| $ER_h$ | 20  | 13.6° | 4°    | 35°  | 8.05°     | +0.85°   | 0.16      |

*Paraplectronoceras pandum Chen & Qi, 1982*: 396; pl. 2, fig. 11, text-fig. 4.
*Paraplectronoceras longicollum Chen & Qi, 1982*: 396; pl. 1, fig. 6; pl. 2, fig. 12.
*Lunanoceras longatum Chen & Qi, 1982*: 397; pl. 1, fig. 15.
*Lunanoceras densum Chen & Qi, 1982*: 397; pl. 2, fig. 10.
*Acaroceras primordium Chen & Qi, 1982*: 399; pl. 1, fig. 11; text-fig. 8.

*Wanwanoceras exiguum Chen & Qi, 1982*: 400; pl. 1, fig. 1, 2; text-fig. 11.

*Paraplectronoceras pyriforme—Chen et al., 1980*: 167; pl. 3, fig. 12, 13; text-fig. 2.
*Paraplectronoceras suxianense—Chen et al., 1980*: 168; pl. 1, fig. 9, 10.
*Plectronoceras* cf. *huaibeiense—Chen & Qi, 1982*: 396; pl. 1, fig. 3.
? *Paraplectronoceras* sp.—*Chen & Qi, 1982*: 397; pl. 1, fig. 10.

**Emended diagnosis**
Small conch with a maximum height of 13 mm, with compressed cross-section
(CWI = 0.69, $n$ = 6; potentially biased), slightly cyrtoconic and endogastric, expansion rate
variable between 4° and 35° (mean = 13.6°, $n$ = 20; potentially biased), without significant
ontogenetic trend ($ER_h$ = 8.1 + 0.85 * ch; $p$ = 0.16). Septal spacing decreasing during
ontogeny with RCL = 0.15 – 0.011 * ch ($n$ = 19, $p$ = 0.0016), corresponding to RCL = 0.1 at
5 mm conch height and RCL = 0.04 at 10 mm conch height. Siphuncle size varies strongly
(likely due to misaligned planes of section) with RSD between 0.07 and 0.29 (mean = 0.16,
$n$ = 18; potentially biased). Expansion of the siphuncular segments apparently weaker than
in larger species. Diaphragms poorly known, but apparently transverse, concave.

**Holotype**
NIGP 46198

**Type locality and horizon**
Hanjia, near Jiagou Town, Suxian County, Anhui Provine, North China; *Acaroceras-
Eburoceras* Zone, upper Jiagou Member, Fengshan Formation.

**Remarks**
Seven species that we consider to be synonymous were described simultaneously *in Chen
et al. (1979a)*, which was published 2 months earlier than *Chen et al. (1979b)*, containing
two additional new species. We choose *Paraplectronoceras inflatum* Chen & Zou *in Chen
et al., 1979a* as senior synonym, despite its holotype displaying strong signals of a
misaligned section with y-rotation. The reason for this decision is that due to the
misaligned section (y-rotation), the septal flap and the expanded part of the segment are
clearly visible (Fig. 18A), while the specimen is otherwise average for the species in its
revised scope. The holotype comes from the *Acaroceras-Eburoceras* Zone of the upper
Jiagou Member of the Fengshan Formation of northern Anhui, which is equivalent to the
*Sinoeremoceras* Zone (Wanwankou Member) in Shangdong and Liaoning (*Chen &
Teichert, 1983*). Further specimens considered by us to be conspecific have been reported
from the lower and upper *Quadraticephalus* Zone of the middle Jiagou Member (*Chen
et al., 1979a*, *1980*; *Chen & Qi, 1982*). *S. inflatum* thus has potentially the longest
stratigraphic range within the genus. *Lunanoceras densum Chen & Qi, 1982* and
*L. longatum Chen & Qi, 1982* (which was also referred to as *L. elongatum Chen & Qi, 1982*
in the same publication) from the lower *Quadraticephalus* Zone at Huaibei do not differ
significantly from the slightly younger *S. inflatum* in any character that cannot be
attributed to misalignment in the plane of section. *Chen et al. (1979b)* reported described
and illustrated *Sinoeremoceras anhuiense* Zou & Chen *in Chen et al., 1979b* and

*Wanwanoceras multiseptum* Zou & Chen *in Chen et al., 1979b* from the *Acaroceras-Eburoceras* Zone, which are further relatively small sized, rapidly expanding *Sinoeremoceras*, their siphuncles including the typical expanded segments and we see no reason to separate these species. We regard *L. compressum* Chen & Qi *in Chen et al., 1979a* as another subjective synonym of *S. inflatum*, because it is similar in size despite its lower expansion rate (which might result from sectioning). Small species such as *S. exiguum* (*Chen & Qi, 1982*) may be regarded as a juvenile of *S. inflatum* with a more slender conch. We also synonymise all previously described species of *Paraplectronoceras* with *S. inflatum*, as all of them come from the Jiagou Member of Anhui and correspond well in terms of their size and conch parameters, while supposed differences are attributable either to intraspecific or ontogenetic variation or to variation in alignment in the plane of section (Fig. 19). This includes nine additional species, *i.e.*, *P. abruptum* Chen & Qi *in Chen et al., 1980*; *P. curvatum* Chen & Qi *in Chen et al., 1980*; *P. impromptum* Chen & Zou *in Chen et al., 1979a*; *P. longicollum Chen & Qi, 1982*; *P. pandum Chen & Qi, 1982*; *P. parvum* Chen & Qi *in Chen et al., 1980*; *P. pyriforme* Chen & Qi *in Chen et al., 1979a*; *P. suxianense* Chen & Qi *in Chen et al., 1979a* and *P. vescum* Chen & Qi *in Chen et al., 1980*. We include in *S. inflatum* one species previously described as an ellesmeroceratid, *Acaroceras primordium Chen & Qi, 1982*, due to its narrow septal spacing and slightly expanded segments with the section showing at least suborthochoanitic septal necks. This is significant insofar as all ellesmeroceratids older than the upper *Quadraticephalus* Zone are now re-identified as plectronoceratids, pushing the origin of the Ellesmeroceratida to slightly younger strata (see also *S.* (?) *shanxiense* (*Chen & Teichert, 1983*) below). The fact that no *Sinoeremoceras* from the *Quadraticephalus* or *Acaroceras-Eburoceras* zones of Anhui is known with diameters above 13 mm (with several species known from body chambers), supports the retention of *S. inflatum* as a valid separate species.

*S. inflatum* differs from *S. marywadeae*, *S. wanwanense*, *S.bullatum*, *S. sibiriense* and *S. sinense* by its small conch size. It differs from *S. magicum* and *S. endogastrum* by its more rapid expansion rate and from the latter also by its more strongly curved conch. *S.* (?) *shanxiense* is generally poorly known but appears to be slightly smaller and might be distinguished by the more compressed cross-section and a smaller siphuncle.

**Geographic and stratigraphic occurrences**
Northern Anhui, North China; *Quadraticephalus* to *Acaroceras-Eburoceras* Zone (= *Proconodontus posterocostatus* Zone to lower *Eoconodontus notchpeakensis* Subzone), middle to upper Jiagou Member, Fengshan Formation, late Jiangshanian to early Stage 10, Furongian, late Cambrian.

***Sinoeremoceras bullatum*** (Chen & Qi *in Chen et al., 1979a*) comb. nov.
Figure 9, 20A–20C, 21; Table 8

*Physalactinoceras bullatum* Chen & Qi *in Chen et al., 1979a*: 13; pl. 3, fig. 10, 11; text-fig. 11.
*Lunanoceras precordium* Chen & Qi *in Chen et al., 1979a*: 9; pl. 2, fig. 12, 13; pl. 4, fig. 16; text-fig. 6.
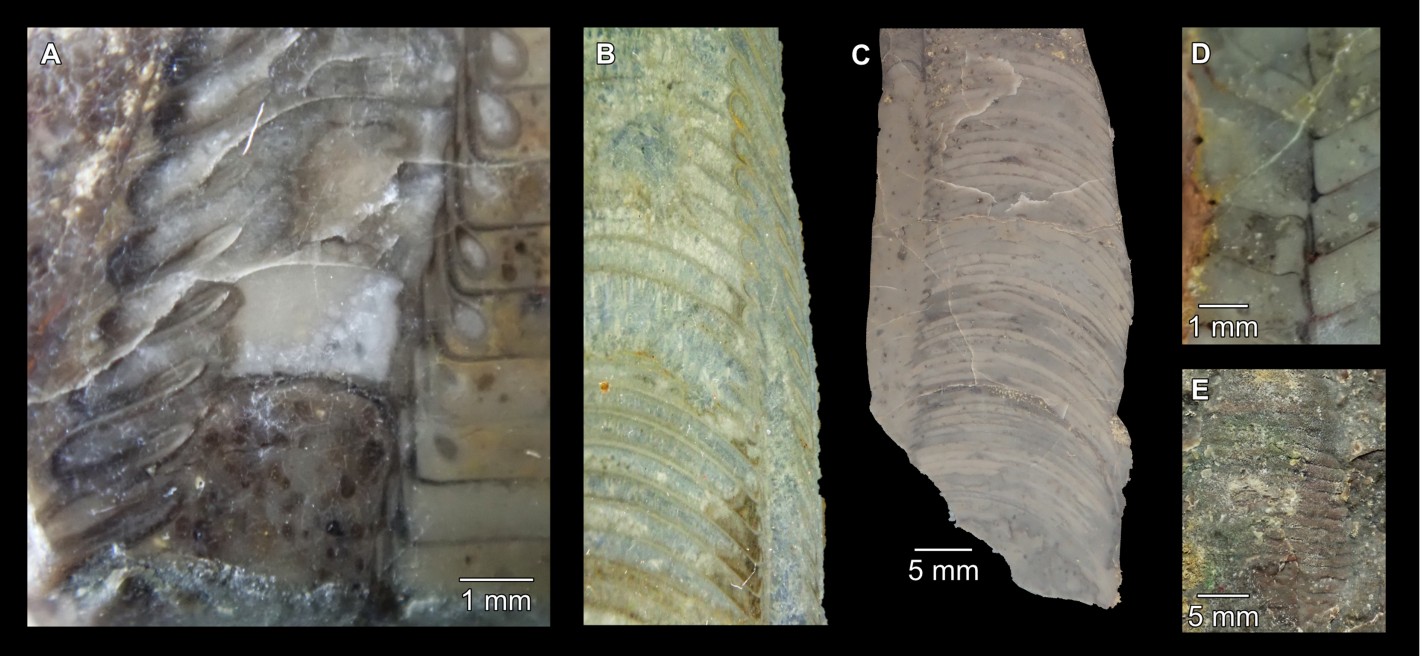

**Figure 20 *Sinoeremoceras* species from the Wanwankou Member, Fengshan Formation of Shandong, North China.** (A–C) *Sinoeremoceras bullatum* (Chen & Qi *in Chen et al., 1979a*). (A) NIGP 46150, holotype, originally attributed to *Physalactinoceras* Chen & Qi *in Chen et al., 1979a*. Detail of misaligned longitudinal section of siphuncle, seemingly showing rapid ontogenetic changes. (B) NIGP 73775, originally designated as holotype of *Mastoceras qiushugouense Chen & Teichert, 1983*. Misaligned longitudinal section of siphuncle, note (separated) expanded apical segments and convex conch margin. (C) NIGP 46155, originally designated as holotype of *Lunanoceras precordium* Chen & Qi *in Chen et al., 1979a*. Misaligned longitudinal section. (D and E) *Sinoeremoceras sinense* (Chen & Qi *in Chen et al., 1979a*). (D) NIGP 46127, holotype, originally attributed to *Eodiaphragmoceras*. Longitudinal section of siphuncle. Note ventraperturally sloping diaphragms. (E) NIGP 52551, paratype. Partial exposure of lateral sutures.

*Lunanoceras changshanense* Chen & Qi *in Chen et al., 1979a*: 9; pl. 3, fig. 5; text-fig. 7.
? *Protactinoceras magnitubulum* Chen & Qi *in Chen et al., 1979a*: 12; pl. 2, fig. 5; pl. 3, fig. 12, 13; text-fig. 10.
? *Protactinoceras lunanense* Chen & Qi *in Chen et al., 1979a*: 12; pl. 2, fig. 8, 9; pl. 4, fig. 1, 2.
*Physalactinoceras globosum* Chen & Qi *in Chen et al., 1979a*: 14; pl. 2, fig. 6, 7; text-fig. 12.
*Physalactinoceras changshanense* Chen & Qi *in Chen et al., 1979a*: 14; pl. 3, fig. 7, 8.
*Physalactinoceras breviconicum* Chen & Qi *in Chen et al., 1979a*: 14; pl. 1, fig. 7; text-fig. 13.
*Physalactinoceras subcirculum* Chen & Qi *in Chen et al., 1979a*: 15; pl. 1, fig. 12, 13.
*Sinoeremoceras foliosum* Chen & Qi *in Chen et al., 1979a*: 15; pl. 1, fig. 3, 4; pl. 2, fig. 10; pl 4, fig. 13; text-fig. 14.
*Sinoeremoceras zaozhuangense* Chen & Qi *in Chen et al., 1979a*: 16; pl. 2, fig. 1, 2; text-fig. 15.
*Wanwanoceras lunanense* Chen & Qi *in Chen et al., 1979a*: 16; pl. 4, fig. 3; text-fig. 16.
*Physalactinoceras compressum Chen & Teichert, 1983*: 77; pl. 16, fig. 2, 4; text-fig. 18.
*Physalactinoceras longiconum Chen & Teichert, 1983*: 80; pl. 10, fig. 5.

*Lunanoceras precordium—Chen & Teichert, 1983*: 52; pl. 1, fig. 1.
*Physalactinoceras* cf. *globosum—Chen & Teichert, 1983*: 80; pl. 15, fig. 2, 3.

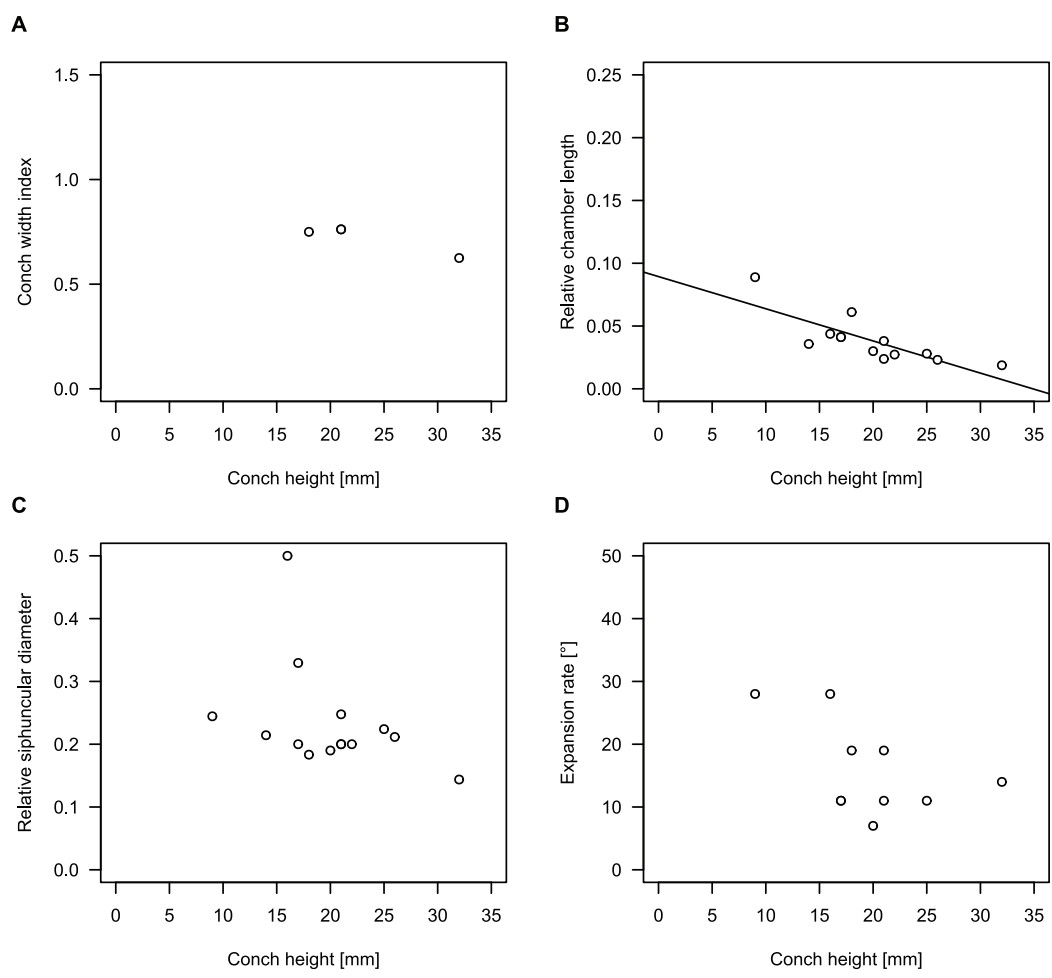

**Figure 21 Conch parameters of *Sinoeremoceras bullatum* (Chen & Qi *in Chen et al., 1979a*) throughout ontogeny (represented by conch height).** (A) Conch width index (CWI). (B) Relative cameral length (RCL). (C) Relative siphuncular diameter. (D) Height expansion rate (ER_h).

**Table 8 Conch parameters and ontogenetic trajectories of *Sinoeremoceras bullatum* (Chen & Qi *in Chen et al., 1979a*).**

|  | *n* | Mean | Min | Max | Intercept | Slope | *p*-value |
|---|---|---|---|---|---|---|---|
| CWI | 4 | 0.72 | 0.63 | 0.76 | 0.96 | −0.010 | 0.049 |
| RCL | 13 | 0.039 | 0.019 | 0.089 | 0.089 | −0.0026 | 0.001 |
| RSD | 14 | 0.23 | 0.14 | 0.5 | 0.36 | −0.006 | 0.16 |
| ER_h | 10 | 15.9° | 7° | 28° | 28.3° | −0.63° | 0.13 |

*Sinoeremoceras foliosum*—*Chen & Teichert, 1983*: 86; pl. 4, fig. 3.
*Sinoeremoceras taiziheense*—*Chen & Teichert, 1983*: 88; pl. 13, fig. 4.
*Sinoeremoceras zaozhuangense*—*Chen & Teichert, 1983*: 88; pl. 15, fig. 1; pl. 17, fig. 6; pl. 19, fig. 2.
*Mastoceras qiushugouense*—*Mutvei, Zhang & Dunca, 2007*: 1328; text-fig. 1A, 2A.

*Mastoceras qiushugouese* [*sic.*]—*Mutvei, 2020*: 119; fig. 3A.
*Physalactinoceras bullatum*—*Pohle et al., 2022*: 6; fig. 2; Supplementary Material.

non *Protactinoceras magnitubulum*—*Chen et al., 1979a*: 12; pl. 1, fig. 6.
non *Physalactinoceras breviconicum*—*Chen & Teichert, 1983*: 76; pl. 13, fig. 6; pl. 19, fig. 6.
non *Physalactinoceras changshanense*—*Chen & Teichert, 1983*: 77; pl. 12, fig. 6.
non *Physalactinoceras globosum*—*Chen & Teichert, 1983*: 79; pl. 5, fig. 3; pl. 12, fig. 4, 7; pl. 15, fig. 4, 5; pl. 19, fig. 9, 10.

### Emended diagnosis

This species has a maximum known conch height of 32 mm and is thus the largest known plectronoceratid. It has a compressed cross-section with CWI = 0.75 ($n$ = 7; potentially biased). Conch shape is slightly cyrtoconic and endogastric, with initially rapid expansion rate that decreases during ontogeny with ER = 47.2 – 1.2 * ch. ($n$ = 7, $p$ = 0.07; potentially biased). This results approximately in ER = 41.2° at 5 mm conch height, 29.2° at 15 mm conch height and 11.2° at 30 mm conch height. Septal spacing decreases slowly during ontogeny with RCL = 0.12–0.0035 * ch ($n$ = 7, $p$ = 0.002), corresponding to RCL = 0.1 at 5 mm conch height, RCL = 0.07 at 15 mm conch height and RCL = 0.02 at 30 mm conch height. Siphuncle size has been reported between RSD = 0.1 and RSD = 0.24 (mean = 0.16, $n$ = 9; potentially biased). Siphuncular segments bulging dorsal to septal flap. Diaphragms directly transverse, concave.

### Holotype

NIGP 46150

### Type locality and horizon

Near Changshan Village, Taozhuang Town, Zaozhuang area, Shandong Province, North China; *Sinoeremoceras* Zone, Upper Middle Algal Limestone (= Wanwankou Member, upper *Procondodontus muelleri* Zone and lower *Eoconodontus notchpeakensis* Subzone), Fengshan Formation, close to Jiangshanian/Stage 10 boundary, Furongian, late Cambrian.

### Remarks

A few plectronoceratid species from Shandong are characterised by concave, transverse diaphragms, such as *Sinoeremoceras bullatum* (Chen & Qi *in Chen et al., 1979a*) comb. nov., which shows the shape of the diaphragms and the transition of the septal flap and the siphuncular bulges quite well. Although seemingly different inclinations of the diaphragms may result from differently aligned sections of the siphuncle with regards to the central ridge of the diaphragm, but it is a conspicuous pattern of directly transverse diaphragms that is not known in the otherwise very similar *S. wanwanense* from Liaoning. Additional indications that these species are separate are given by the larger size of the specimens from Shandong and by the tendency for more strongly endogastrically curved conchs in *S. bullatum*, in contrast to the nearly orthoconic conch of most specimens from Liaoning. Conch parameters display continuous variation among specimens here assigned to *S. bullatum*. According to *Chen et al. (1979a)*, the difference between *Lunanoceras precordium* and *L. changshanense* Chen & Qi *in Chen et al., 1979a* are more sharply bent
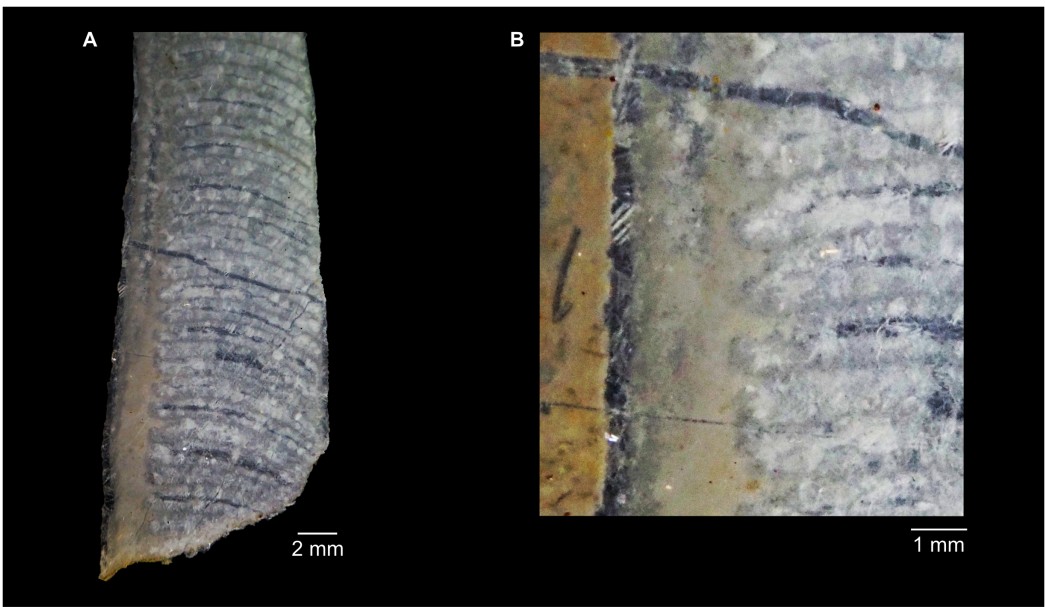

**Figure 22** *Sinoeremoceras endogastrum* (*Li, 1984*) **from the** *Lotagnostus americanus* **Zone, Siyangshan Formation of Zhejiang, South China.** (A) NIGP 79778, originally designated as paratype of *Parapalaeoceras sinense* Li, 1984. Misaligned longitudinal section, probably slightly parallel to the median plane. (B) Same specimen, detail of siphuncle. Note the expanded siphuncular segments and cyrtochoanitic septal necks. 

septal necks and a rounder cross-section, both of which can easily be explained by a misaligned plane of section or slight intraspecific variation. *Physalactinoceras compressum Chen & Teichert, 1983* has transverse diaphragms, which are somewhat ω-shaped. We interpret this as further evidence for more complex diaphragm morphology than might be anticipated (compare diaphragms in *S. marywadeae* sp. nov.) but treat the species as a synonym of *S. bullatum* until they are better understood. Using the siphuncle of *S. marywadeae* sp. nov. as a reference, we cannot find indications for distinct species within the Wanwankou Member of Shandong, with the possible exception of *S. sinense*, which has peculiarly S-shaped septal necks that can at the moment not be explained by misalignment of the plane of section. Thus, we keep both *S. bullatum* and *S. sinense* as the only sympatric species pair but note that *S. sinense* falls otherwise within the range of variation in *S. bullatum*. Several species from Shandong are only known from sections outside the septal flap; thus, synonymisation with *S. bullatum* is mainly for convenience and to avoid declaring nomina dubia. These synonyms are marked as tentative above.

*S. bullatum* differs from *S. wanwanense* in its more strongly curved conch and its more regularly concave and transverse diaphragms. *S. bullatum* is larger than most other species of *Sinoeremoceras* except for *S. sibiriense*, which differs by having a straight conch.

**Geographic and stratigraphic occurrences**
Type locality and horizon only.

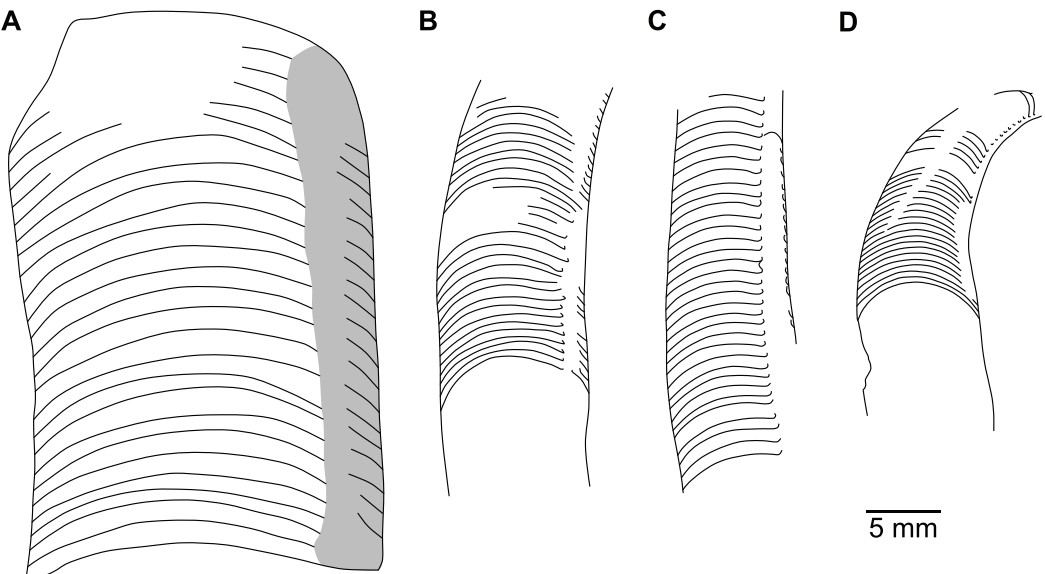

**Figure 23 Drawings of longitudinal sections of lesser known *Sinoeremoceras* species.** (A) *S. sibiriense* (*Balashov, 1959*) (ibid., pl. 5, fig. 12a); holotype, SPB 13/426; Ust-Kut Formation of Krasnojarsk, Siberia, Russia. Originally tentatively attributed to *Multicameroceras* Kobayashi, 1933. (B) *S. magicum* Chen *in Lu, Zhou & Zhou, 1984* (ibid., pl. 1, fig. 14); holotype, NIGP 65428; *Sinoeremoceras* Zone of Inner Mongolia, North China. (C) *S. endogastrum* (*Li, 1984*); holotype, NIGP 79753; Siyangshan Formation of Zhejiang, South China. Originally attributed to *Parapalaeoceras* Li, 1984. Redrawn after Fig 18 in *Li, 1984*. (D) *S. (?) shanxiense* (*Chen & Teichert, 1983*); holotype, NIGP 73768; upper Yenchou Member, Fengshan Formation of Shanxi, North China. Redrawn after text-fig. 13 in *Chen & Teichert, 1983*, to reflect the interpretation of the authors. Originally attributed to *Hunyuanoceras Chen & Teichert, 1983*. (A and B) Traced from originally published photographs with poor resolution, details visible in the original material may be missing here.

*Sinoeremoceras endogastrum* (*Li, 1984*) comb. nov.
Figures 22, 23C, 24; Table 9

*Parapalaeoceras endogastrum Li, 1984*: 230; pl. 7, fig. 12, 13; text-fig. 18.
*Parapalaeoceras sinense Li, 1984*: 232; pl. 7, fig. 14, 15; text-fig. 19.

**Emended diagnosis**
Relatively small, up to 9 mm phragmocone diameter; body chamber unknown. Conch nearly orthoconic, with low expansion rate, mean $ER_h$ = 6.5° (*n* = 2) and apparently relatively broad cross-section (CWI = 0.87) and more narrowly rounded ventral side (single specimen with probably misaligned plane of section). Septal spacing relatively wide, RCL = 0.09 at a conch height of approximately 8 mm, though wider adapically. Siphuncle with mean RSD = 0.16 (*n* = 2). The expansion of the segments is slight.

**Holotype**
NIGP 79753

**Type locality and horizon**
Duibian Village, near Jiangshan City, western Zhejiang Province, South China; Cephalopod Bed DD-14 (= KD-60), 1–3 m below the *Lotagnostus hedini* Zone, in

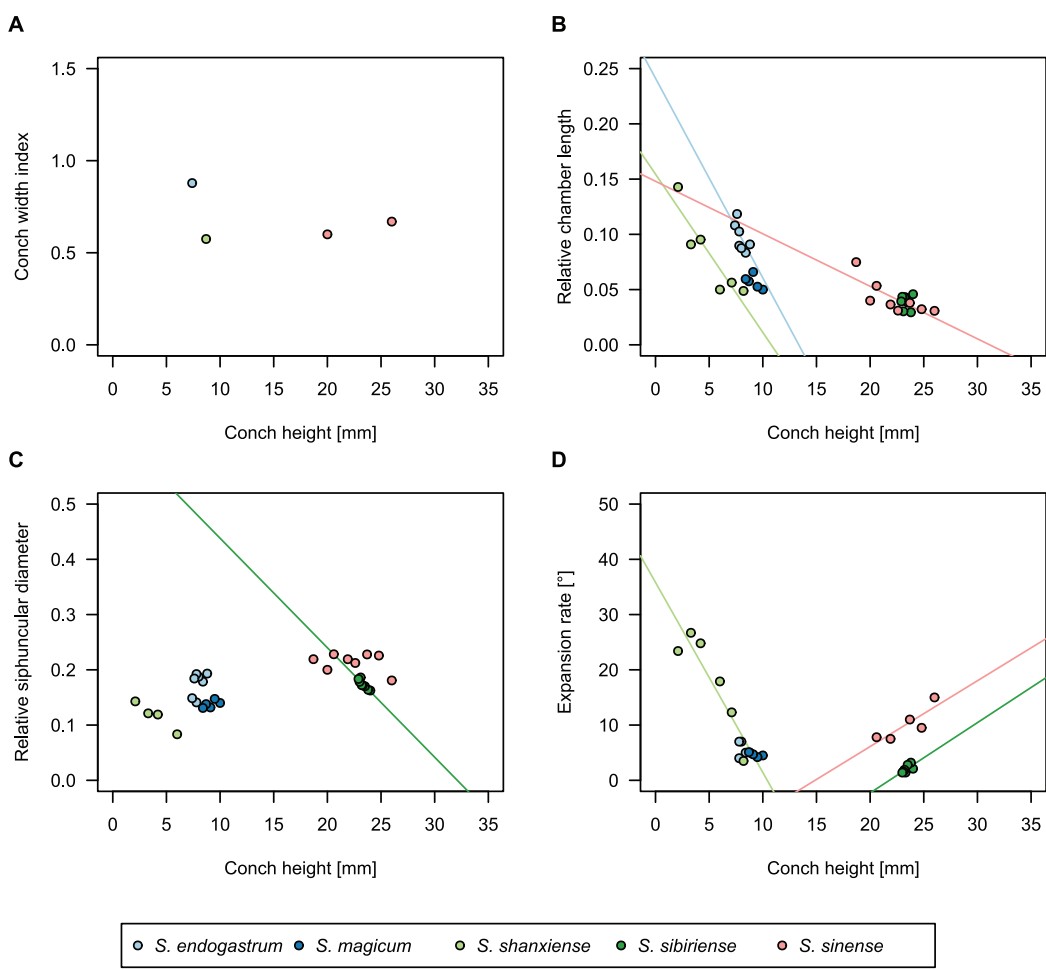

**Figure 24 Conch parameters throughout ontogeny (represented by conch height) of *Sinoeremoceras* species that are only known from few isolated specimens.** *S. endogastrum* (*Li, 1984*), *S. magicum* Chen in *Lu, Zhou & Zhou (1984)*, *S. shanxiense* (*Chen & Teichert, 1983*), *S. sibiriense* (*Balashov, 1959*) and *S. sinense* (Chen & Qi *in Chen et al., 1979a*). (A) Conch width index (CWI). (B) Relative cameral length (RCL). (C) Relative siphuncular diameter. (D) Height expansion rate (ER_h).

association with *L. punctatus* (= *L. americanus*), *Acaroceras-Antacaroceras* Zone (= *L. americanus* Zone, corresponding to the *Proconodontus posterocostatus* Zone in Laurentia), lower Siyangshan Formation (stratotype section of the Jiangshanian Stage, see *Peng et al., 2012*).

**Remarks**

This species is closely similar to *S. inflatum* from the Jiagou Member of the Fengshan Formation of northern Anhui. It differs by the less strongly curved, almost orthoconic conch and lower expansion rate, although the latter is difficult to judge from the few available specimens. The chambers are relatively long compared to other plectronoceratids, though they hardly exceed 1 mm. Since there are only two reported specimens, data on conch parameters and their intraspecific and ontogenetic variation is

**Table 9 Conch parameters and ontogenetic trajectories of *Sinoeremoceras endogastrum* (*Li, 1984*).**

|  | n | Mean | Min | Max | Intercept | Slope | p-value |
|---|---|---|---|---|---|---|---|
| CWI | 1 (1) | 0.88 | 0.88 | 0.88 | – | – | – |
| RCL | 7 (2) | 0.097 | 0.08 | 0.12 | 0.24 | −0.018 | 0.09 |
| RSD | 7 (2) | 0.18 | 0.14 | 0.19 | −0.0047 | +0.023 | 0.24 |
| ER$_h$ | 5 (2) | 5.6° | 4° | 7° | 13.0° | −0.9° | 0.63 |
| ch | 7 (2) | 8.0 mm | 7.4 mm | 8.8 mm | – | – | – |

Note:
Measurements were taken from multiple ontogenetic points per specimen, values in brackets denote number of individual specimens.

**Table 10 Conch parameters and ontogenetic trajectories of *Sinoeremoceras magicum* Chen *in Lu, Zhou & Zhou (1984)*.**

|  | n | Mean | Min | Max | Intercept | Slope | p-value |
|---|---|---|---|---|---|---|---|
| RCL | 5 | 0.057 | 0.05 | 0.066 | 0.11 | −0.0062 | 0.25 |
| RSD | 5 | 0.14 | 0.13 | 0.147 | 0.78 | +0.0066 | 0.26 |
| ER$_h$ | 4 | 4.6° | 4.2° | 5.1 | 9.4° | −0.51° | 0.25 |
| ch | 5 | 9.1 mm | 8.4 mm | 10 mm | – | – | – |

Note:
All measurements come from a single specimen, the holotype NIGP 65428.

scarce. The maximum extent of the expanded siphuncular segments and the detailed morphology of the septal flap are unknown. We consider *Parapalaeoceras sinense Li, 1984* a subjective junior synonym because the supposed differences, namely the larger expansion rate and siphuncle size hardly go beyond a level that cannot be attributed to intraspecific variation or misaligned planes of section (Fig. 24). *S. endogastrum* is chosen as the senior synonym because *S. sinense* Chen & Qi *in Chen et al., 1979a* is preoccupied. The species is the only plectronoceratid from Zhejiang and South China in general. It is possible that several other species described as ellesmeroceratids or yanheceratids represent further misaligned sections of *S. endogastrum*, though none shows clearly expanded siphuncular segments. For example, *Yanheceras shanbeilingense Li, 1984* shows similarly short septal spacing, but long straight septal necks. Without having access to 3D reconstructions, it is thus impossible to tell, whether these straight septal necks were straight around the entire septal foramen, or whether they represent a section exactly midway through the septal flap.

**Geographic and stratigraphic occurrences**
Type locality and horizon only.

***Sinoeremoceras magicum* Chen *in Lu, Zhou & Zhou, 1984***
Figures 23B, 24; Table 10

*Sinoeremoceras magicum* Chen *in Lu, Zhou & Zhou, 1984*: 123; pl. 1, fig. 13, 14.

**Emended diagnosis**
Medium-sized (body chamber at~10 mm conch height), with endogastric curvature, short septal spacing (RCL = 0.05, $n$ = 1) and low expansion rate (ER$_h$ = 7°, $n$ = 1). Cross-section

compressed, elliptical (CWI unknown, no measurements given, and cross-section not figured). Siphuncle ventral marginal, with expanded segments; septal necks cyrtochoanitic, sharply bent outwards. Suture with broad and shallow lateral lobes and narrowly rounded dorsal and ventral saddles.

### Holotype
NIGP 65428

### Type locality and horizon
Zhusilenghaierhan Hill, Ejin Banner, Alxa League, Inner Mongolia, North China; *Sinoeremoceras* Zone (bed h13), (corresponding to the late Pre-Payntonian in Australia according to *Lu, Zhou & Zhou (1984)*, *i.e.*, *Proconodontus posterocostatus* to *P. muelleri* zones), Jiangshanian/Stage 10, Furongian, late Cambrian.

### Remarks
This species was only very briefly described in the figure captions of *Lu, Zhou & Zhou (1984)*; its main characteristics are translated above, adjusted in accordance with our revision of *Sinoeremoceras* and with the addition of the corresponding conch parameters. The cross-section was described as elliptical, and it appears probable that CWI would fall within a similar range as other species. This species is one of the few described with visible sutures, making comparisons with the Australian species possible. In contrast to the latter, *S. magicum* has very shallow lateral lobes. It is similar in size to *S. inflatum* and distinctly smaller than *S. wanwanense* and *S. bullatum*. However, the expansion rate is relatively slow, similar to *S. endogastrum*, from which it differs in its cyrtoconic conch. Another argument for the provisional retention of this species is its isolated geographic position, being the only report of a plectronoceratid from Inner Mongolia. Because only a single specimen is known, nothing can be said about its intraspecific variation or detailed siphuncular structure.

### Geographic and stratigraphic occurrences
Type locality and horizon only.

***Sinoeremoceras* (?) *shanxiense* (*Chen & Teichert, 1983*) comb. nov.**
Figures 23D, 24, 25; Table 11

*Hunyuanoceras shanxiense* *Chen & Teichert, 1983*: 59; pl. 5, fig. 1; Text-fig. 13.

### Emended diagnosis
Adult body chamber height slightly below 10 mm; endogastric; septal spacing short (RCL = 0.05 at conch height of 8.7 mm); expansion rate moderate ($ER_h$ = 19°). Cross-section apparently very strongly compressed (CWI = 0.57, possibly biased). Siphuncle marginal ventrally, RSD = 0.14, structure poorly known, but seems to consist of both cyrtochoanitic and orthochoanitic septal necks, the latter of which are tentatively interpreted as evidence for a septal flap. Diaphragms apparently present but poorly preserved.

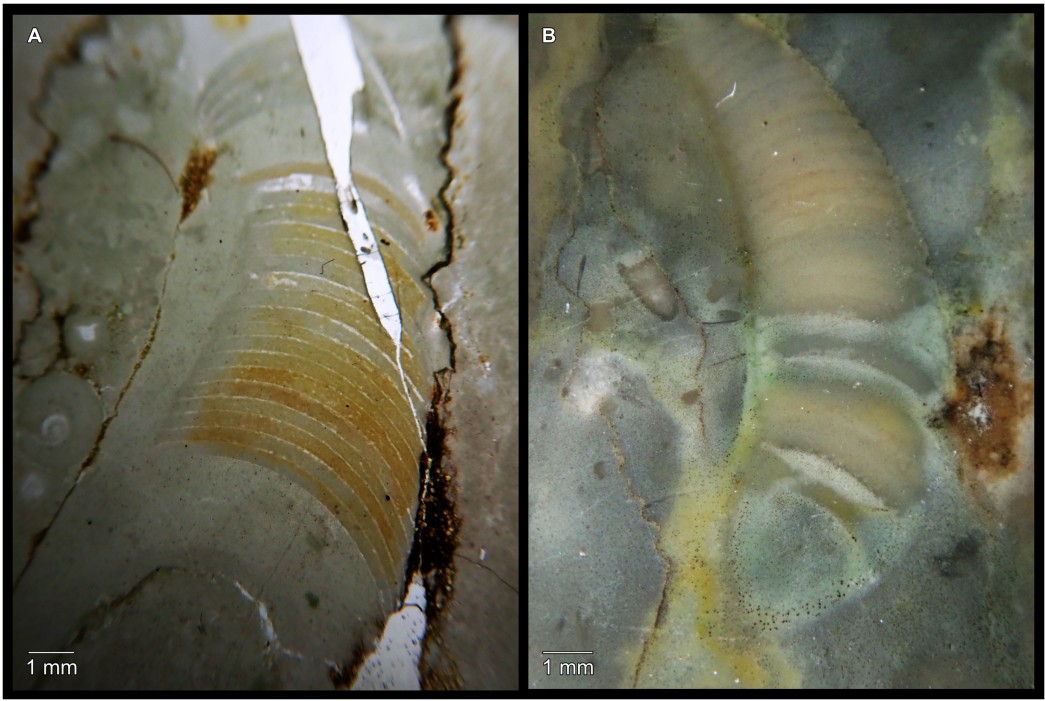

**Figure 25** *Sinoeremoceras* (**?**) *shanxiense* (*Chen & Teichert, 1983*) **from the lower** *Quadraticephalus* **Zone, upper Yenchou Member, Fengshan Formation of Shanxi, North China.** (A) NIGP 73768, holotype, originally attributed to *Hunyuanoceras Chen & Teichert, 1983*. Misaligned longitudinal section, note the barely visible siphuncle with poorly preserved septal necks. Same specimen as in Fig. 23D (photographed from the back side of the thin section, thus mirrored). (B) NIGP 73769, paratype, misaligned longitudinal section, as indicated by the rounded adapertural end.

**Table 11  Conch parameters and ontogenetic trajectories of *Sinoeremoceras* (?) *shanxiense* (*Chen & Teichert, 1983*).**

|  | $n$ | Mean | Min | Max | Intercept | Slope | $p$-value |
|---|---|---|---|---|---|---|---|
| CWI | 1 | 0.57 | 0.57 | 0.57 | – | – | – |
| RCL | 6 | 0.081 | 0.049 | 0.14 | 0.15 | −0.014 | 0.01 |
| RSD | 4 | 0.12 | 0.083 | 0.14 | 0.17 | −0.015 | 0.02 |
| $ER_h$ | 6 | 18.1° | 3.5° | 26.7° | 35.8° | −3.4° | 0.01 |
| ch | 7 | 5.7 mm | 2.1 mm | 8.7 mm | – | – | – |

**Note:**
All measurements come from a single specimen, the holotype NIGP 73768.

## Holotype
NIGP 73768

## Type locality and horizon
Xuankong Temple, Hunyuan County, Datong City, Shanxi Province, North China; lower *Quadraticephalus* Zone (= *Proconodontus posterocostatus* Zone), Yenchou Member, Fengshan Formation, Jiangshanian/Stage 10, Furongian, late Cambrian.

**Table 12 Conch parameters and ontogenetic trajectories of *Sinoeremoceras sibiriense* (*Balashov, 1959*).**

|  | n | Mean | Min | Max | Intercept | Slope | p-value |
|---|---|---|---|---|---|---|---|
| RCL | 8 | 0.039 | 0.029 | 0.046 | 0.046 | −0.0003 | 0.97 |
| RSD | 8 | 0.17 | 0.16 | 0.19 | 0.64 | −0.020 | 0.002 |
| $ER_h$ | 7 | 2.1° | 1.4° | 3.2° | −27.7 | 1.27 | 0.10 |
| ch | 8 | 23.4 mm | 22.9 mm | 24.0 | – | – | – |

**Note:**
All measurements from the holotype, SPB 13/426.

## Remarks
This species was originally described as an ellesmeroceratid. Although the holotype is poorly preserved, *Chen & Teichert (1983*, text-fig. 13) illustrated a transition from apparently orthochoanitic to cyrtochoanitic septal necks, which suggests tentative assignment to *Sinoeremoceras*. However, the septal necks are difficult to differentiate in the holotype itself. The strongly compressed cross-section and correspondingly low CWI is possibly underestimated due to misalignment of the section; it also appears that part of the dorsum is missing in the holotype. The paratype represents a strongly displaced section that does not expose the siphuncle but can be tentatively assigned to the same species based on the very dense septal spacing.

## Geographic and stratigraphic occurrences
Type locality and horizon only.

***Sinoeremoceras sibiriense*** (*Balashov, 1959*) comb. nov.
Figures 23A, 24; Table 12

*Multicameroceras sibiriense Balashov, 1959*: 37; pl. 5, fig. 12.
*Multicameroceras* (?) *sibiricum* [*sic.*]—*Balashov, 1962b*: 8; pl. 2, fig. 10a-c.
*Multicameroceras siberiense* [*sic.*]—*Flower, 1964*: 161.

## Emended diagnosis
Conch orthoconic, slowly expanding, with smooth shell. Cross-section oval (CWI unknown), slightly compressed laterally. Septal spacing very short, 17 chambers within the same distance as the conch diameter (RCL ≅ 0.06). Septal concavity equals the length of almost two chambers (SCI ≅ 0.12). Suture with weak lateral lobes. Siphuncle ventral, slightly laterally compressed. Septal necks short, slightly bent outwards (= suborthochoanitic or cyrtochoanitic?). Siphuncular segments slightly expanded. Connecting rings poorly visible due to recrystallisation. Diaphragms not known.

## Holotype
SPB 13/426 (SPB 880/9135 according to *Balashov, 1962b*)

## Type locality and horizon
Chunya River, Krasnojarsk, Siberia, Russia, specific locality unknown; Chunya Regional Stage.

**Table 13 Conch parameters and ontogenetic trajectories of *Sinoeremoceras sinense* (Chen & Qi *in Chen et al., 1979a*).**

|  | n | Mean | Min | Max | Intercept | Slope | p-value |
|------|-------|---------|---------|--------|-----------|---------|---------|
| CWI | 2 (2) | 0.63 | 0.6 | 0.67 | 0.37 | +0.012 | – |
| RCL | 8 (4) | 0.04 | 0.031 | 0.075 | 0.15 | −0.0048 | 0.02 |
| RSD | 8 (4) | 0.21 | 0.18 | 0.23 | 0.26 | −0.002 | 0.46 |
| $ER_h$ | 5 (4) | 10.2° | 7.5° | 15° | −17.7° | +1.2° | 0.07 |
| ch | 8 (4) | 22.3 mm | 18.7 mm | 26 mm | – | – | – |

Note:
Measurements were taken from multiple ontogenetic points per specimen, values in brackets denote number of individual specimens.

## Remarks

The main characteristics of this species are translated above from *Balashov (1959)* and adjusted in accordance with the current revision of *Sinoeremoceras*. This species was first mentioned in English language literature in the addendum to *Flower (1964)*. *Dzik (2020)* presented rich material from the Ust-Kut Formation, of the Angara River in southern Krasnojarsk, Siberia. The holotype of *S. sibiriense* comes from the probably coeval or slightly younger Chunya Regional Stage on the Chunya River, which is a few hundred kilometres to the north of *Dzik*'s *(2020)* locality, although the exact type locality and horizon of *S. sibiriense* are not well defined. Nevertheless, *Balashov (1962b)* reported the same species from the Angara River and the similarity of the ellesmeroceratid specimens from these two localities, reported by *Balashov (1962b)* and *Dzik (2020)*, suggest that they are roughly of the same age. Furthermore, the cephalopods of the Ust-Kut Formation generally resemble other Cambrian faunas (*e.g.*, that of the Wanwankou Member) and thus, it appears plausible that *S. sibiriense* is of similar or slightly younger age as other plectronoceratids. *Balashov (1959)* distinguished the species from *M. multicameratum* by the presence of diaphragms and absence of connecting rings in the latter species; however, these can hardly serve as diagnostic characters, since diaphragms are restricted to juvenile portions of the conch and both structures may be susceptible to diagenesis. Nevertheless, we retain this species because it is the only report of *Sinoeremoceras* from Siberia and more material is needed in order to properly assess its similarity to the Australian and Chinese material. The only described specimen is in the upper size range but comparable to *Sinoeremoceras* species from the Wanwankou Member, although it does not preserve the body chamber. *Balashov (1962b)* mentioned two additional specimens but did not provide measurements or images. The species may be distinguished from other *Sinoeremoceras* species by its relatively large size in combination with the very low expansion rate and an essentially orthoconic conch.

## Geographic and stratigraphic occurrences

Ust-Kut and Chunya Regional Stages (poorly dated; according to *Dzik (2020)* not older than *Cordylodus proavus* Zone, Stage 10, Furongian, but older than *Cordylodus angulatus* Zone, earliest Tremadocian, Early Ordovician); Chunya and Angara Rivers, Krasnoyarsk, Siberia, Russia.

***Sinoeremoceras sinense*** (Chen & Qi *in Chen et al., 1979a*) comb. nov.
Figures 20D, 20E, 24; Table 13

*Eodiaphragmoceras sinense* Chen & Qi *in Chen et al., 1979a*: 10; pl. 4, fig. 6–10, 15.
*Eodiaphragmoceras sinense* Chen & Qi, 1979—*Chen & Teichert, 1983*: 51; pl. 2, fig. 5.

**Emended diagnosis**
Conch almost orthoconic, only very slightly endogastric, with maximum reported height of 26 mm, though no body chambers are known. Cross-section strongly compressed, CWI = 0.67 ($n$ = 1), though it is possible that this an underestimation, partially caused by an misaligned section. Expansion rate is moderate, with 15°. There is not enough data on the ontogenetic trajectory of septal spacing, but at 26 mm conch height, RCL = 0.03. Sutures partially known, apparently almost straight transverse, with only very slight lateral lobes. The siphuncle has RSD = 0.18. In the known sections, the siphuncle appears to be only slightly expanded and the septal necks somewhat S-shaped. Diaphragms slope towards the dorsum adaperturally.

**Holotype**
NIGP 46127

**Type locality and horizon**
Near Changshan Village, Taozhuang Town, Zaozhuang area, Shandong Province, North China; *Sinoeremoceras* Zone, Upper Middle Algal Limestone (= Wanwankou Member, upper *Procondodontus muelleri* Zone and lower *Eoconodontus notchpeakensis* Subzone), Fengshan Formation, close to Jiangshanian/Stage 10 boundary, Furongian, late Cambrian.

**Remarks**
The structure of the connecting ring of *S. sinense*, which was originally proposed as the type species of *Eodiaphragmoceras* Chen & Qi *in Chen et al., 1979a* is not well known. However, the apparent variation between cyrto-, ortho- and holochoanitic septal necks indicates a septal flap and therefore, *Eodiaphragmoceras* cannot be separated from *Sinoeremoceras*. The slight S-shape of the septal necks and the slope direction of the diaphragms are nevertheless difficult to explain by reference to the effect of a misaligned plane of section and they differentiate this species within the genus. Because *S. sinense* and *S. bullatum* co-occur, it is difficult to assign some specimens to either species, especially if the plane of section does not pass through the septal flap as in *Protactinoceras magnitubulum* Chen & Qi *in Chen et al., 1979a*. Because we only assign specimens to *S. sinense* if S-shaped septal necks are clearly present, it is possible that some specimens currently assigned to *S. bullatum* must be transferred to *S. sinense* in the future.

Other than septal neck shape and size, this species is distinguished by its sutures, which appear to be almost straight, lacking the lateral lobes of, for example, *S. marywadeae* sp. nov.

**Geographic and stratigraphic occurrences**
Type locality and horizon only.

Genus *Palaeoceras* *Flower, 1954*

**Type species**
*Palaeoceras mutabile* *Flower, 1954*

**Included species**
Only the type species.

**Emended diagnosis**
Conch slender (ER$_h$ ≅ 3–8°), compressed (CWI ≅ 0.7–0.9) orthoconic to very slightly cyrtoconic (exogastric). Septal spacing decreasing strongly during ontogeny, with relatively long chambers compared to other plectronoceratids at the earliest known ontogenetic stages (up to RCL = 0.2), but slightly below RCL = 0.1 in the largest known specimens, *i.e.*, at about 8 mm diameter. Septal necks orthochoanitic or hemichoanitic. Expanded, straight or concave parts of siphuncular segments occur.

**Remarks**
*Palaeoceras* *Flower, 1954* differs from *Sinoeremoceras* by its essentially orthoconic conch. *Flower (1954*, *1964)* described marked changes of the siphuncle during ontogeny, from expanded segments with hemichoanitic septal necks to straight segments and macrochoanitic septal necks. These apparent ontogenetic changes are reminiscent of the misinterpretations in other plectronoceratids, where similar ontogenetic changes seemed to be present, but were, in fact, due to misaligned planes of section. From the available specimens, it is impossible to interpret the three-dimensional shape of the siphuncle. Presumably, *Flower (1954*, *1964)*— like other cephalopod researchers at that time— assumed that the siphuncle was more or less radially symmetric (as it is the case in all non-plectronoceratid cephalopods), which led to his interpretation of the siphuncle. *Flower (1954*, *1964)* described the septal necks as longer on the ventral side than on the dorsal side, which together with the lack of cyrtochoanitic septal necks does not correspond to the septal flap seen in *Sinoeremoceras marywadeae* sp. nov. It is possible that he was misguided by the extremely oblique siphuncular segments, which cause the ventral edge of the septum near the septal foramen to be almost parallel to the siphuncle, making the boundary to the septal necks difficult to discern. Future studies will need to establish, whether the three-dimensional interpretations by *Flower (1954*, *1964)* were accurate, ideally using 3D imaging techniques. At the very least, the published sectioned specimens reveal strongly inclined and expanded siphuncular segments reminiscent of *Sinoeremoceras* but differ in their lack of cyrtochoanitic septal necks. The shape of the septal flap or if there is one at all is unclear. Accordingly, *Palaeoceras* is transitional between plectronoceratids and ellesmeroceratids, which have straight to concave siphuncular segments. Because of the shape of these segments, we retain the genus within the Plectronoceratidae.

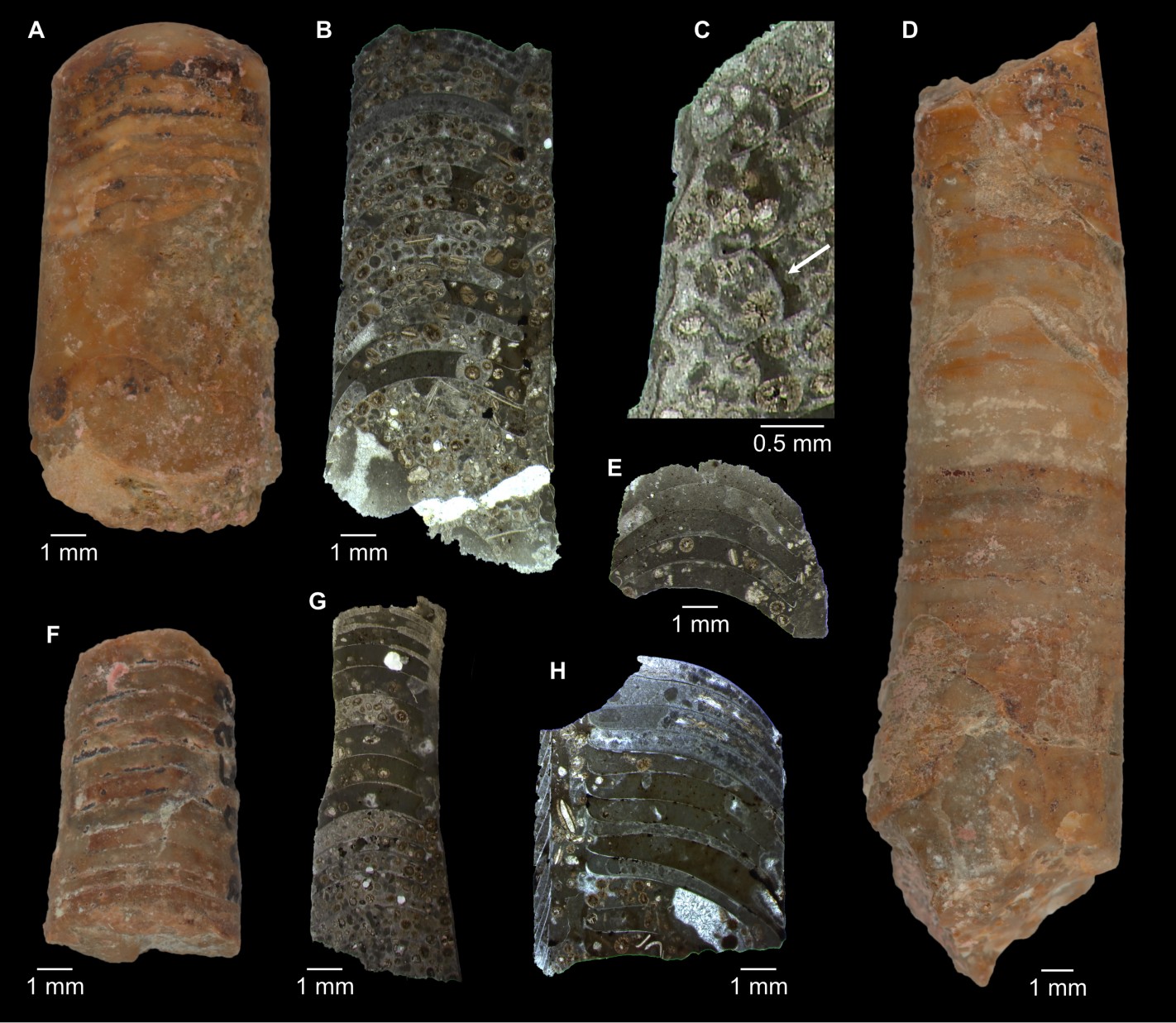

**Figure 26** *Palaeoceras mutabile* **Flower, 1954 from the *Cambrooistodus minutus* Subzone, San Saba Member, Wilberns Formation of Gillespie County, Texas, USA.** Image courtesy of the Smithsonian Institution (photos: Nicholas Drew). (A) USNM PAL 302534, lateral view of body chamber and phragmocone, venter right. (B) USNM PAL 302535, longitudinal section. (C) Same specimen, enlargement of sectioned siphuncle further apicad. Arrow indicates expanded connecting ring as interpreted by *Flower (1964)*. (D) USNM PAL 302540, originally designated as holotype of *Balkoceras gracile* Flower, 1964. Lateral view of phragmocone, venter right. Note that apparent exogastric curvature may partially be caused by polishing ventral side and breakage of specimen apically. (E) USNM PAL 30539, originally designated as paratype of *B. gracile*, longitudinal section. (F) USNM PAL 302532, originally designated as holotype of *Plectronoceras exile Flower, 1964*. Lateral view of phragmocone, venter left. (G) Same specimen, longitudinal section further apicad. (H) USNM PAL 302529, longitudinal section, originally designated as holotype of *Palaeoceras undulatum* Flower, 1964.

*Balkoceras Flower, 1964* is considered a junior subjective synonym of *Palaeoceras Flower, 1954*, because its type species *Balkoceras gracile Flower, 1964* is essentially identical with *Palaeoceras mutabile Flower, 1964*, except for the very slight exogastric curvature.

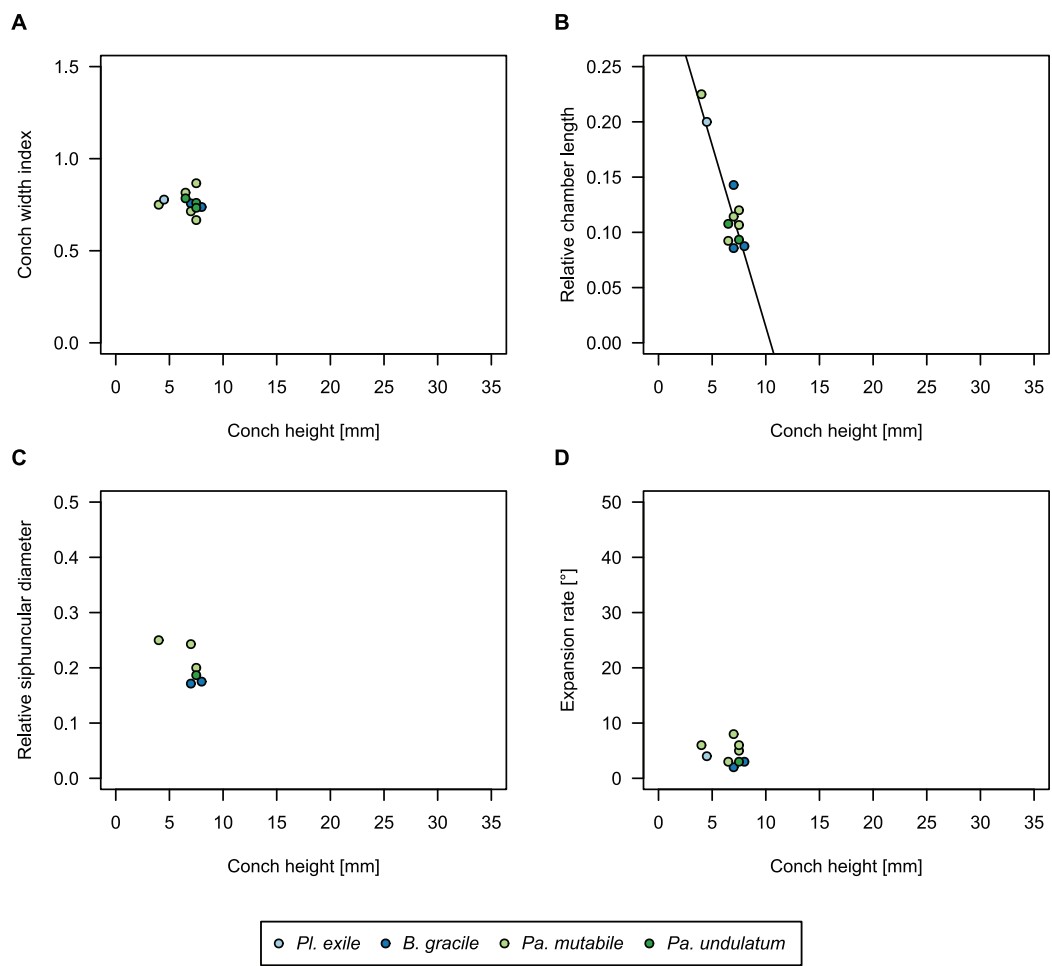

**Figure 27 Conch parameters of *Palaeoceras mutabile Flower, 1954* throughout ontogeny, (represented by conch height).** (A) Conch width index (CWI). (B) Relative cameral length (RCL). (C) Relative siphuncular diameter. (D) Height expansion rate (ER$_h$).

**Table 14 Conch parameters and ontogenetic trajectories of *Palaeoceras mutabile Flower, 1954*.**

|       | *n* | Mean   | Min   | Max  | Intercept | Slope   | *p*-value |
|-------|-----|--------|-------|------|-----------|---------|-----------|
| CWI   | 11  | 0.76   | 0.67  | 0.87 | 0.79      | −0.0037 | 0.72      |
| RCL   | 11  | 0.13   | 0.086 | 0.23 | 0.34      | −0.033  | 0.0002    |
| RSD   | 6   | 0.20   | 0.17  | 0.25 | 0.32      | −0.017  | 0.11      |
| ER$_h$ | 9  | 4.4°   | 2°    | 8°   | 6.1°      | −0.25   | 0.64      |
| ch    | 12  | 6.7 mm | 4 mm  | 8 mm | –         | –       | –         |

*Flower (1964*, p. 34) wrote "Indeed, I had at first assigned the present species to *Palaeoceras*, and surely the affinities are close enough that this course would have been eminently justifiable, but so great has the importance of curvature become that such a course would only result in the proposal of a new genus for this form by the next person to

give these older cephalopods any attention". Comparing the proportions of the various specimens of *Palaeoceras* and *Balkoceras* described by *Flower (1954, 1964)* reveals that they are so close to each other that even the separation at species level is untenable, especially as the exogastric curvature is very subtle.

### *Palaeoceras mutabile* Flower, 1954
Figures 26, 27; Table 14

*Palaeoceras mutabile Flower, 1954*: 10; pl. 1, fig. 5, 9, 10; pl. 2, fig. 11; pl. 3.
*Plectronoceras exile Flower, 1964*: 30; pl. 4, fig. 13–16; pl. 5, fig. 1.
*Palaeoceras undulatum Flower, 1964*: 32; pl. 1, fig. 1–10; pl. 2, fig. 8–10.
*Balkoceras gracile Flower, 1964*: 34; pl. 2, fig. 1–3; pl. 3, fig. 10–15.

*Palaeoceras mutabile*—*Flower, 1964*: 31; pl. 2, fig. 4–7, 11–18; pl. 3, fig. 1–9.
*Palaeoceras mutabile*—*Furnish & Glenister, 1964*: K146; fig. 83,1a-b.

**Emended diagnosis**
As for genus, by monotypy.

**Holotype**
BEG 34757

**Type locality and horizon**
San Saba Member, 67 feet below the top (= *Cambrooistodus minutus* Subzone, *Miller, Loch & Taylor, 2012*, fig. 5), Wilberns Formation; Threadgill Creek section, northern Gillespie County, Texas, USA.

**Remarks**
The three-dimensional structure of the siphuncle is unknown. Furthermore, the sections are difficult to compare to those of the Asian taxa because those showing variation between hemichoanitic to cyrtochoanitic septal necks were made by polishing the specimens from the ventral side, but the illustrations and descriptions by *Flower (1954, 1964)* give little indication on the position of the xz-plane of the section relative to the siphuncle. Specimens sectioned longitudinally invariably have long orthochoanitic septal necks. The connecting ring is mostly missing, though the segments appear to be expanded apically (Fig. 26C).

The specimens described as *Balkoceras gracile Flower, 1964* are unique among plectronoceratids in their exogastric curvature. For this reason, *Flower (1964)* established a new family, while noting the close similarity to *Palaeoceras mutabile*. However, his own measurements show that the dimensions of all specimens described as either *P. mutabile Flower, 1954*; *P. undulatum Flower, 1964*; *Plectronoceras exile Flower, 1964* (already suggested as referrable to *Palaeoceras* by *Chen & Teichert, 1983*) or *B. gracile Flower, 1964* fall within a very narrow range, making even a distinction at species level questionable. Thus, curvature is the only character separating *P. mutabile* from *P. gracile*, but note that the difference is very slight and we tentatively synonymise those species as they overlap in all other conch parameters (Fig. 27). The combination of an essentially straight conch with

low expansion rate, lack of cyrtochoanitic septal necks and small size separate *Palaeoceras* from species of *Sinoeremoceras*.

### Geographic and stratigraphic occurrences
Type locality and horizon only.

Genus **Plectronoceras** *Ulrich & Foerste, 1933*

### Type species
*Plectronoceras cambria* (*Walcott, 1905*)

### Included species
Only the type species.

### Emended diagnosis
Small plectronoceratid with adult size below 5 mm, cyrtoconic conch with moderate expansion rate ($ER_h \cong 15°$) and compressed cross-section ($CWI \cong 0.8$). Chambers short with RCL between about 0.05 and 0.1 with a tendency to decrease during ontogeny.

### Remarks
*Plectronoceras* *Ulrich & Foerste, 1933* is the type genus of the Plectronoceratida, and due to its stratigraphic position the most often cited Cambrian cephalopod. Unfortunately, its internal characters are poorly known. However, if the single connecting ring known from *Plectronoceras* and the variation between orthochoanitic to cyrtochoanitic septal necks is taken as evidence for a septal flap, then there is little that distinguishes this genus from *Sinoeremoceras* apart from size. Indeed, small species of *Sinoeremoceras*, such as *S. inflatum* are so similar that they could also be assigned to *Plectronoceras*. Nevertheless, this needs to be confirmed, ideally using three-dimensional reconstructions of the siphuncle. We regard the genera as separate but note that a future suppression of *Sinoeremoceras* in favour of *Plectronoceras* appears likely to us. As *Plectronoceras exile Flower, 1964* is considered by us as a synonym of *Palaeoceras mutabile Flower, 1954* (see also *Chen & Teichert, 1983*), the genus is at present exclusively known from the *Ptychaspis-Tsinania* Zone of North China.

**Plectronoceras cambria** (*Walcott, 1905*)
Figure 28, 29; Table 15

*Cyrtoceras cambria Walcott, 1905*: 22.
*Plectronoceras liaotungense Kobayashi, 1935*: 17; pl. 4, fig. 1–3.
*Plectronoceras huaibeiense* Chen & Qi *in Chen et al., 1979a*: 6; pl. 3, fig. 9.

*Cyrtoceras cambria—Walcott, 1913*: 98; pl. 6, fig. 4, 4a-c.
*Plectronoceras cambria—Miller, 1943*: 99; fig. 1A–D.
*Plectronoceras cambria—Ulrich et al., 1944*: 133; pl. 68, fig. 4–11.
*Plectronoceras cambria—Flower, 1954*: 15; fig. 3C–F.
*Plectronoceras liaotungense—Flower, 1954*: 16; fig. 3A-B.

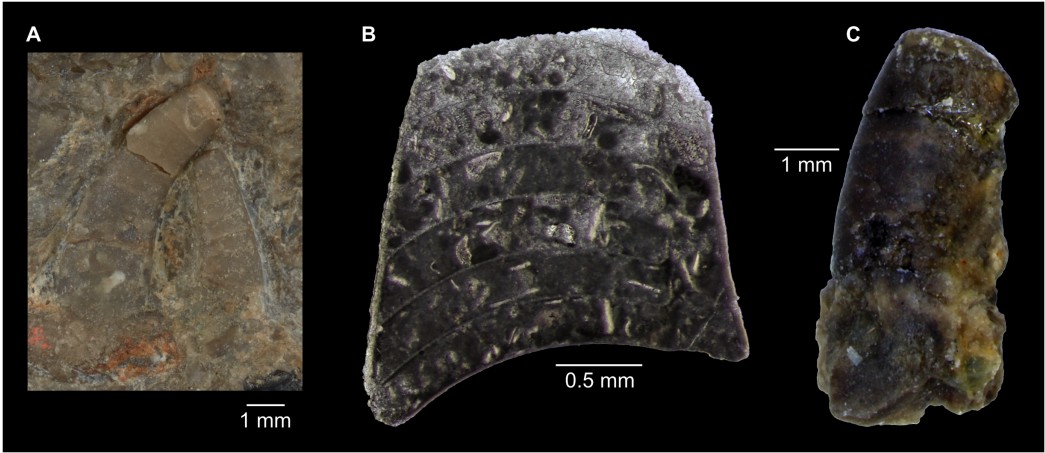

**Figure 28** *Plectronoceras cambria* (*Walcott, 1905*) **from the *Ptychaspis-Tsinania* Zone, lower Yenchou Member of Anhui, Liaoning and Shandong, China.** Image courtesy of the Smithsonian Institution (photos: Nicholas Drew). (A) USNM PAL 57819, holotype, lateral view, left specimen with venter right, right specimen with venter left. (B) USNM PAL 57820, paratype, longitudinal section. Note the long straight septal necks, likely indicating an exact median section through the septal flap. (C) USNM PAL 57820, paratype, lateral view, venter right.

*Plectronoceras cambria*—*Balashov, 1962a*: 73; pl. 5, fig. 6.
*Plectronoceras cambria*—*Furnish & Glenister, 1964*: K146; fig. 81a, b.
*Plectronoceras liaotungense*—*Furnish & Glenister, 1964*: K146; fig. 81c.
*Plectronoceras cambria*—*Yochelson, Flower & Webers, 1973*: 277; fig. 2.
*Plectronoceras* cf. *cambria*—*Chen et al., 1979a*: 6; pl. 4, fig. 11, 14; text-fig. 4.
*Plectronoceras cambria*—*Dzik, 1984*: 14; fig. 2.
*Plectronoceras liaotungense*—*Dzik, 1984*: 14; fig. 2.
*Plectronoceras huaibeiense*—*Dzik, 1984*: 14; fig. 2.
*Plectronoceras liaotungense*—*Kobayashi, 1989*: 370; text-fig. 1.
*Plectronoceras liaotungense*—*Webers, Yochelson & Kase, 1991*: 347; fig. 1.
*Plectronoceras cambria*—*Kröger, 2003*: 43; text-fig. 4, 11.
*Plectronoceras cambria*—*Kröger, Vinther & Fuchs, 2011*: 604; fig. 3C, D.
*Plectronoceras cambria*—*Vinther, 2015*: 23; fig. 2.
*Plectronoceras* cf. *cambria*—*Fang et al., 2019*: fig. 2.

non *Plectronoceras* cf. *huaibeiense* Chen & Qi—*Chen & Qi, 1982*: 396; pl. 1, fig. 3.

**Emended diagnosis**
As for genus.

**Holotype**
USNM 57819

**Type locality and horizon**
Kaolishan, Tai'ain City, Shandong Province; *Ptychaspis-Tsinania* Zone (= *Proconodontus tenuiserratus* Zone, compare *Bagnoli et al., 2017*), lower Yenchou Member, Fengshan Formation.

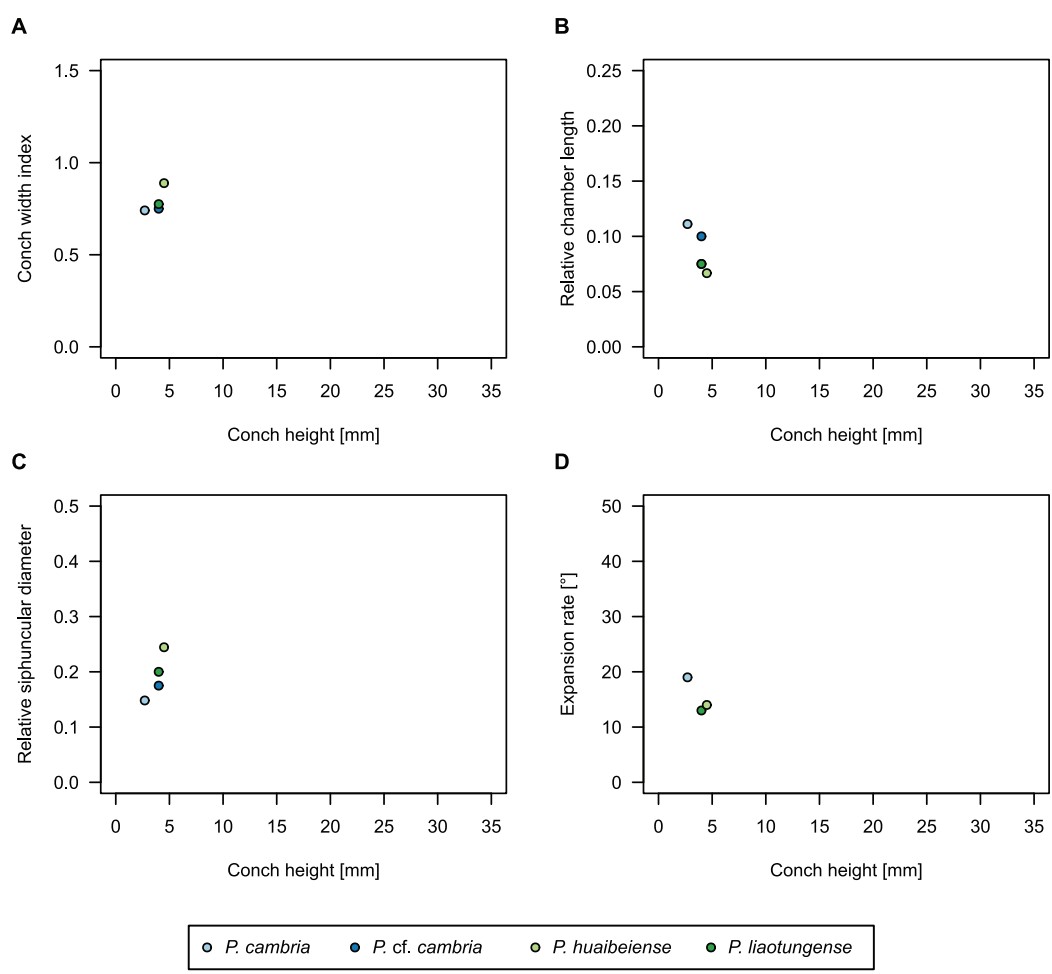

**Figure 29 Conch parameters of *Plectronoceras cambria* (*Walcott, 1905*) throughout ontogeny (represented by conch height).** (A) Conch width index (CWI). (B) Relative cameral length (RCL). (C) Relative siphuncular diameter. (D) Height expansion rate (ER$_h$).

**Table 15 Conch parameters and ontogenetic trajectories of *Plectronoceras cambria* (*Walcott, 1905*).**

|  | *n* | Mean | Min | Max | Intercept | Slope | *p*-value |
|---|---|---|---|---|---|---|---|
| CWI | 4 | 0.79 | 0.74 | 0.89 | 0.55 | +0.063 | 0.29 |
| RCL | 5 | 0.86 | 0.067 | 0.11 | 0.18 | −0.024 | 0.079 |
| RSD | 4 | 0.19 | 0.15 | 0.24 | 0.047 | +0.018 | 0.12 |
| ER$_h$ | 3 | 15.3° | 13° | 19° | 27.1° | −3.15° | 0.27 |
| ch | 5 | 3.8 mm | 2.7 mm | 4.5 mm | – | – | – |

## Remarks

In terms of proportions and adult size, the three described species of *Plectronoceras* are indistinguishable. *Kobayashi (1935)* credited *P. cambria Walcott, 1905* with a more slender conch, a more ovate cross-section, longer chambers and a less curved suture to distinguish

it from *P. liaotungense Kobayashi, 1935*. *P. huaibeiense* Chen & Qi *in Chen et al., 1979a* was described without differential diagnosis. We find no justification for separation of different *Plectronoceras* species, as the variability among different specimens assigned to *Plectronoceras* is generally small (Fig. 29) and we attribute those differences to ontogenetic or intraspecific variation. Variation between different *Plectronoceras* specimens is smaller than the variation within a single species of *Sinoeremoceras*. Remarkably, *P. cambria* appears to have a wider geographic distribution than later plectronoceratids, as contemporaneous specimens from Anhui, Liaoning and Shandong have almost identical proportions, while there are considerable size differences indicating separate populations or species of *Sinoeremoceras*. The specimen described by *Chen & Qi (1982)* as *Plectronoceras* cf. *huaibeiense* Chen & Qi *in Chen et al., 1979a* from the lower *Quadraticephalus* Zone of Suxian, Anhui likely does not belong to *Plectronoceras* and is better referred to the co-occurring *Sinoeremoceras inflatum* (Chen & Zou *in Chen et al., 1979a*).

**Geographic and stratigraphic occurrences**
Liaoning, Shandong and Anhui Provinces, North China; *Ptychaspis-Tsinania* Zone (= *Proconodontus tenuiserratus* Zone), lower Yenchou and lower Jiagou members, respectively, Fengshan Formation, Jiangshanian, Furongian, late Cambrian.

# CONCLUSIONS

*Sinoeremoceras marywadeae* sp. nov. from the Ninmaroo Formation (Stage 10, Furongian, Cambrian) at Black Mountain, Queensland, Australia, collected 40–50 years ago by Mary Wade and her team, is described from more than 200 specimens, exceeding the entire previously known published record of plectronoceratids. Importantly, these specimens reveal the complex three-dimensional structure of the plectronoceratid siphuncle for the first time. Considerations of the plane of section relative to the median plane enable in-depth revision of plectronoceratid cephalopods, revealing that many taxa mainly based on longitudinal sections, including the Protactinoceratida, are synonyms due to misalignment of the plane of section with the median plane. The main cause for these misinterpretations was the bilateral symmetry of the plectronoceratid siphuncle. The septal flap is a strongly adapically elongated part of the septal necks on the dorsal side of the siphuncle. Laterally and ventrally, the siphuncular segment was expanded, with cyrtochoanitic septal necks. We further document the variability of conch parameters and their ontogenetic trajectories in Cambrian cephalopods, which provides additional arguments to significantly reduce the number of taxa, but also allows to recognise distinct spatio-temporal populations of plectronoceratids. In our revised taxonomy, the Plectronoceratida consists of one family, three genera and eleven species, out of which nine are assigned to *Sinoeremoceras*. The specimens from Black Mountain provide an excellent opportunity to investigate the early evolution of cephalopods and the still poorly understood origin of the siphuncle. For future research, we encourage obtaining 3D-reconstructions (*e.g.*, μCT-scans or serial grinding tomography) of plectronoceratids from the original localities.

## INSTITUTIONAL ABBREVIATIONS

**BEG**    Bureau of Economic Geology, now housed in the Non-vertebrate Paleontology Laboratory, Jackson School Museum of Earth History, University of Texas at Austin, USA.

**NIGP**   Nanjing Institute of Geology and Palaeontology, Nanjing, China.

**QMF**    Queensland Museum, Brisbane, Australia.

**SPB**    St. Petersburg collection, formerly Department of Palaeontology, Leningrad State University, Russia (current repository not traced).

**UMUT**   University Museum, University of Tokyo, Japan.

**USNM**   National Museum of Natural History, Smithsonian Institution, Washington D.C., USA.

## ACKNOWLEDGEMENTS

We dedicate this article to Mary Wade, who collected the material and spent a lot of time in the lab preparing and sectioning specimens. Would it not be for her, these faunas would still largely be unknown. We hope that we did her legacy justice, although we will never know, whether she would have agreed with all our conclusions. We thank the Queensland Museum and in particular Kristen Spring and Andrew Rozefelds for making the loan of the specimens possible and facilitating the transfer. We thank Björn Kröger (Helsinki) and Marcela Cichowolski (Buenos Aires) for helpful advice and discussions, and Evelyn Friesenbichler (Zurich) for drawing reconstructions. AP is indebted to Fang Xiang (Nanjing) for his hospitality and access to the large Cambrian cephalopod collection in Nanjing. AP also thanks Annette Jell for the hospitality during his stay in Brisbane. Nicholas Drew (Washington) provided photographs of the specimens housed in the collections of the Smithsonian Institution. The reviews by David Peterman (Pennsylvania), Vojtěch Turek (Prague) and Ed Landing (New York) provided valuable suggestions to improve the final version of the manuscript.

### Funding

This study was supported by the Swiss National Foundation (project nr. 200020_169627) and the Deutsche Forschungsgemeinschaft (project nr. 507867999). The funders had no role in study design, data collection and analysis, decision to publish, or preparation of the manuscript.

### Grant Disclosures

The following grant information was disclosed by the authors:
Swiss National Foundation: 200020_169627.
Deutsche Forschungsgemeinschaft: 507867999.

### Competing Interests

The authors declare that they have no competing interests.

## Author Contributions

- Alexander Pohle conceived and designed the experiments, performed the experiments, analyzed the data, prepared figures and/or tables, authored or reviewed drafts of the article, and approved the final draft.
- Peter Jell conceived and designed the experiments, authored or reviewed drafts of the article, and approved the final draft.
- Christian Klug conceived and designed the experiments, prepared figures and/or tables, authored or reviewed drafts of the article, and approved the final draft.

## Data Availability

The measurements, calculated conch parameters and correlations of ontogenetic trajectories are available in the Supplemental Files.

## New Species Registration

The following information was supplied regarding the registration of a newly described species:

Publication LSID: urn:lsid:zoobank.org:pub:F1C67134-9A19-4D18-AB74-B984B5555D40.

*Sinoeremoceras marywadeae* sp. nov. LSID: urn:lsid:zoobank.org:act:581A8E45-9064-4209-B256-5BCACB199967.

## Supplemental Information

Supplemental information for this article can be found online at http://dx.doi.org/10.7717/peerj.17003#supplemental-information.

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
