# Peer review of "Plectronoceratids (Cephalopoda) from the latest Cambrian at Black Mountain, Queensland, reveal complex three-dimensional siphuncle morphology, with major taxonomic implications"

_PeerJ, doi:10.7717/peerj.17003_

## Round 0.1 · original submission · Minor Revisions

Dear authors,

Based on the three reviewers' unanimous decision of 'minor revisions', I have accepted their recommendations.

All three are very complimentary about your manuscript and have offered constructive comments that will improve the quality of the final text.

I look forward to receiving your revised manuscript.

·

Basic reporting

Reviewer report for MS #94261 “Plectronoceratids (Cephalopoda) from the latest Cambrian at Black Mountain, Queensland reveal complex three-dimensional siphuncle morphology”.

The authors present a comprehensive description of some of the earliest known cephalopods (Plectronocerida), using new specimens from Australia. This material adds a considerable number of specimens to the existing records of this order. Furthermore, the authors address systemic problems with previously published descriptions based on oblique sections, prompting a revision of the order.

Most of my comments concern minor clarifications. My most substantial comment involves using statistical tests to determine differences between fitted slopes (see line 466 comment). I also think the manuscript could be made more concise, but I recognize this task would be difficult based on the sheer number of new descriptions and comparisons.

Line 55: This first sentence could be split up for readability. The reader might be inclined to think the numbers of genera and species refer to “unequivocal cephalopods”, and not only cephalopods from this time. For clarity, I suggest changing to “….genera having been described from this time. Most descriptions are from….”.

Line 133: Please add comma after “thought”.

Line 147: Please briefly elaborate on the details that were excluded.

Line 158: instead of “failed to mention” it would be softer wording to say “he did not mention”. The voice in this text sounds frustrated at times, which is understandable considering the challenges with the material in these groups. I suggest keeping this in mind to not sound too critical of earlier workers.

For indices depending on the conch diameter – how is diameter chosen for bent cone shapes? From apical end to the ventral-most point? This might be worth clarifying.

Line 335: Please reread sentence. I think the word “intended” should be deleted.

Line 365: What hypotheses in particular? That Protactinoceratida have morphological similarities to Plectronocerida? Please briefly state.

Line 466: Please state p-values to back up this statement that the slopes are distinct. I suggest doing this wherever possible. I see p-values are reported in tables below that must be testing whether or not a slope differs from zero. But it would also strengthen the manuscript if you include statistical tests to determine if slopes differ from each other (e.g., with a t-test).

Line 483: Missing parentheses around citation?

End of discussion: Your reconstructions in Figure 15 range from limpet-like soft body to features expected of a crown group cephalopod. However, these early cephalopods may have had constraints on buoyancy imposed by their relatively large soft body volumes and close septal spacing. In our paper referenced below, we found body chamber ratios of 40% (curvilinear body chamber length to curvilinear total length) imparted moderate negative buoyancy. Ratios of <35% could have achieved neutral buoyancy (with our modeled proportions and assumptions). Would it be possible to estimate minimum body chamber ratios from your material? If so, you could weigh in on this argument. While these models are not as sophisticated as my later ones, they can still serve as a first-order estimate for buoyancy. Even if these cephalopods were not capable of neutral buoyancy, they still might have been able to saltate off the bottom (perhaps like the flamboyant cuttlefish, which also has slight negative buoyancy). If you wanted to add a brief discussion of this point, it could back up your design choices for the free-swimming reconstruction in Figure 15.

See Peterman et al. (2019): https://doi.org/10.26879/884.

Experimental design

See comment for line 466 regarding an additional statistical test to distinguish between slopes (only where relevant to back up statements in the text).

Validity of the findings

no comment

Additional comments

no comment

·

Basic reporting

The main contribution of the paper is a critical taxonomic revision of the late Cambrian plectronocerids. The study is based mainly on material obtained during field research in the Black Mountains (Queensland) in the 1970s and 1980s by M. Wade and her co-workers. However, the results of this research were left by M. Wade only in notes and an elaborated MS.
The paper offers new insights into the sudden appearance and diversification of cephalopods in the late Cambrian. Also, key to the study is the first description of three-dimensional structures within the siphonal tube of plectronocerids. Based on the wealth of knowledge gained, with over 200 specimens exceeding all material recorded to date, taxa described from other regions - China, Kazakhstan, Siberia and North America - could also be included in this review. The critical approach leading to a major reduction in the genera (from 18 to three) and species (from 68 to 10) recognized by the authors is based on the fact that most taxa were established on the basis of morphologically poorly known material. These were mostly sections made outside the medial plane and rotated in various ways. The diagnostic characters reported were thus problematic in many cases. This led, according to the authors of the present study, to distortions of many basic parameters of the shells and false conclusions about the surprisingly high diversity of the oldest cephalopods.

The study is written in clear language and uses the established terminology in an understandable way. However, as English is not my first language, I cannot further assess grammatical correctness.
The introductory chapters are sufficiently comprehensive and well-conceived. They provide a detailed introduction to the history of the research, the nature of the material and the methodology used. The chapter on species delimitation is also detailed, which is necessary for such a comprehensive revision of all taxa described so far. Other chapters devoted to morphology and changing parameters of shells during ontogeny are also carefully treated. Literature is well referenced and relevant, and the structure of the paper conforms to disciplinary norms. Figures are relevant and have a high quality (with a few exceptions, see below).
The paper is richly supplemented with additional graphical appendices and statistical data demonstrating the validity of the taxonomic conclusions reached by the authors.
This is primary research within the scope of the journal with a clearly defined problem and approach to the solution with well stated conclusions that are convincingly supported. I consider the study to be very carefully crafted and useful.

Experimental design

No comment.

Validity of the findings

See "Basic reporting"

Additional comments

Given the fundamental importance of the taxonomic revision contained in the present work and the great relevance to the broader palaeontological community, I suggest a possible revision of the title to reflect those aspects. The following comes to my mind as an example:
Revision of plectronocerids (Cephalopoda, latest Cambrian), based on material from the Black Mountain, Queensland revealing complex three-dimensional siphuncle morphology.

The title of the paper says plectronoceratids but starting with the introduction authors talks primarily about the order, not the family. Because the authors prefer the abbreviated version published in Treatise of Invertebrate Paleontology, part K (1964) for the names of orders, which they also use in the abstract of the paper, i.e., Plectronocerida instead of Plectronoceratida. I therefore recommend changing the name of the order in the present study and replace plectronoceratids with plectronocerids.

Lines 364, 365 "....Mutvei (2020) came to similar conclusion ...". Australian specimens confirm these hypotheses ..." For a better understanding, I recommend repeating what conclusions and hypotheses are involved.

In Figure 7: It would be useful to clearly indicate by arrows or similar the different structures mentioned in the text. For a paleontologist not specializing in cephalopods this may not be entirely clear.

Significantly better quality of some photographs does not seem to be allowed by the preservation of material. However, in the case of Figs 7J-M showing very interesting internal structures, the low quality of pictures might be helped by increasing the contrast. Same in the case of Figs 22 and 25. If better quality cannot be achieved, I suggest adding a pen drawing.

·

Basic reporting

Review comments on Pohle et al. on Cambrian cephalopods

Overall: This is a superb, coherent, well written review that focuses on the oldest cephalopods, with a focus on important legacy collections from the Ninmaroo Formation. It would be accessible even to advanced undergraduate students and certainly to grad students. Its approach to taxonomy strips away the existing complexity in family- and higher-level nomenclature.

The “species delimitations” and “ontogenetic trajectories” provide a needed standard and "lesson" for specimen preparation and comparative taxonomy. Other than line editing, only a few modifications are needed:

Mandatory change: Throughout, use Cambrian paleocontinent names. “North America” is an amalgum of least four Phanerozoic continents, all the known Cambrian specimens are from “Laurentia.” As a Cambrian paleocontinent, it is “Sibiria” (this spelling) with Siberia and likely the Mongolian occurrences. South and North China are likely paleocontinents, Australia is the margin of East Gondwana.

Abstract: Great. Maybe add short comment on highest known diversity of Late Cambrian cephs in North China with this as a center of diversification.

Text length, figures, and tables all great.

References up to date.

Line edits—throughout use semicolon in double series, as “Taxon Taxonomic author 1, 19XX; Taxon T. a. 2, 19XX; etc.

Suggest:
Line 11, “colleagues” not “party
20, “Laurentia” not North America, (as elsewhere in text and Table 3)
36, “because”
37, awkward, ?”morphologic uncertainty in defining and recognizing species”?
38, 39, which are the junior synonyms, three families = two?
43: “assigned” not “distributed” add Landing et al. (2023) to list of Cambrian reports
58, 59: “Laurentia”
73: “the type”
78: ?”towards”?
79: “erecected” sp.
79, 80, reads awkwardly
86, An American “Treatise of Paleontology”
89: “1954,” “1964,” Commas
90: here and elsewhere, “in” in italics
97: “the order”
100: briefly explain “differences’
104–107: double series semicolons
144: again, what “difference” explain, vague
152: Problem: No, Ulrich’s (1911) Ozarkian is regularly not understood and is not simply Tremadocian! See Byers 2001, Geoscience Wisconsin. The Ozarkian defined in Missouri and Wisconsin absolutely included everything in New York and Pennsylvania from the upper Middle Cambrian through Floian.
163: “upper Cambrian part of the Ninmaroo Formation”
198: “lower Datsonian regional stage” Don’t mix/combine geochronologic (early, middle, late) with chronostratigraphic (lower, middle, upper) adjectives.
204: Chronic word misuse in paleontology: “FAD” is properly a horizon of phyletic origin, which will not be found in course of field work. Say “LO” or “lowest occurrence” which is what is found by the paleontologist as a result of taphonomic, collection, migration artifacts. See Landing et al. (2013, see researchgate for download) for problems with “FAD.”
208: No, base of Tremadocian is within (not base of) local C. lindstroemi Zone, as you correctly show in Fig. 2.214, 215: As correctly defined: the “lowest occurrence of E. notchpeakensis below the onset of the HERB carbon isotope excursion” defines the base of the Stage 10/Lawsonian Stage. Ahlberg and others simply could not understand a chronostrat unit base defined by a LO and onset of a chemozone and never could write it correctly. Cite Landing et al. (2010, 2011), check researchgate ed landing for 2010 abstract.
215: “zones 2–5” with longer n-dash, not hyphen, n-dash means “through”
219–223: “Cambrian–Ordovician” with n-dash
335: “herein”
593: : ?”misaligned”? better
797, 798, again “LO” not “FAD”
800: “members,” the plural of a capitalized noun is lower case
828, small addition, “based on their widespread occurrence and highest diversity”

Table 3, looks confusing as format does not indent species names. Also, keep regional distributions as a paleocontinental block, so North and South China follow Sibiria (correct spelling) with Siberia and Mongolia regions. Use “Laurentia”. Should Nevada and New York unidentified plectonoceroids be listed? Why not?

Experimental design

As stated above, directions taken for specimen preparation in this legacy collection from Australia and data collection fully outlined and provide a standard for other's work.

Validity of the findings

Simply, stated, lower (species) and higher taxonomic conclusions not assailable.

Additional comments

A list of comments on relatively minor changes in content and in line editing provided above.

---

## Round 0.2 · accepted · Accept

Dear authors,

I asked the reviewer who had the most extensive comments to re-review your manuscript. Based on their recommendation I have accepted it for publication.

The production team will contact you to take your case through the proofing stages.

Congratulations, and I hope you will use PeerJ as your publication venue again in the future.

·

Basic reporting

The authors have satisfactorily addressed all comments. I have no more suggestions.

Experimental design

The authors have satisfactorily addressed all comments. I have no more suggestions.

Validity of the findings

The authors have satisfactorily addressed all comments. I have no more suggestions.

Additional comments

I look forward to this paper being published.